# Distinct neurochemical influences on fMRI response polarity in the striatum

Domenic H. Cerri[1,2,3,16], Daniel L. Albaugh[1,2,3,4,16], Lindsay R. Walton [1,2,3,16], Brittany Katz[1,2,3], Tzu-Wen Wang [1,2,3], Tzu-Hao Harry Chao [1,2,3], Weiting Zhang[1,2,3], Randal J. Nonneman[1,2,3], Jing Jiang[5,6,7,8,9], Sung-Ho Lee [1,2,3], Amit Etkin[5,6,10], Catherine N. Hall [11,12], Garret D. Stuber [13,14,15] & Yen-Yu Ian Shih [1,2,3] ✉

The striatum, known as the input nucleus of the basal ganglia, is extensively studied for its diverse behavioral roles. However, the relationship between its neuronal and vascular activity, vital for interpreting functional magnetic resonance imaging (fMRI) signals, has not received comprehensive examination within the striatum. Here, we demonstrate that optogenetic stimulation of dorsal striatal neurons or their afferents from various cortical and subcortical regions induces negative striatal fMRI responses in rats, manifesting as vasoconstriction. These responses occur even with heightened striatal neuronal activity, confirmed by electrophysiology and fiber-photometry. In parallel, midbrain dopaminergic neuron optogenetic modulation, coupled with electrochemical measurements, establishes a link between striatal vasodilation and dopamine release. Intriguingly, in vivo intra-striatal pharmacological manipulations during optogenetic stimulation highlight a critical role of opioidergic signaling in generating striatal vasoconstriction. This observation is substantiated by detecting striatal vasoconstriction in brain slices after synthetic opioid application. In humans, manipulations aimed at increasing striatal neuronal activity likewise elicit negative striatal fMRI responses. Our results emphasize the necessity of considering vasoactive neurotransmission alongside neuronal activity when interpreting fMRI signal.

Functional magnetic resonance imaging (fMRI) is the most widely used technique to noninvasively map brain function[1]. Critical for the interpretation of fMRI data is the assumption that local hemodynamic fMRI signals reflect neuronal activity. This straightforward interpretation of fMRI signals is largely based on seminal studies with concurrent neuronal and hemodynamic recordings in the cerebral cortex[2-4]. However, there is growing evidence that hemodynamics are also subject to cell-type- and neurochemical-specific influences and could be differently regulated outside of cortex[5-13]. Although the cell types, circuit architecture and neurochemical milieu of subcortical regions may be highly distinct from cortex, the extent to which these factors can alter the

relationship between neuronal and hemodynamic activity in these regions has yet to be comprehensively determined.

The dorsal striatum (or caudate-putamen, CPu, in rodents) is the major input nucleus of the basal ganglia, and has been widely studied by fMRI for its diverse roles in learning[14], motivation[15-17], cognition[18], and motor function[19,20]. Striatal circuit dysfunction occurs in many neurodevelopmental, neurological and neuropsychiatric conditions (such as autism[21], Parkinson's disease[22,23] and substance use disorders[24,25]), with fMRI studies widely used to discern the nature of pathology-related striatal dysfunction and regional impact of therapeutic interventions[26-28]. The striatum is comprised primarily (over

---

90%) of GABAergic medium spiny neurons (MSNs)[29] and features relatively dense expression of markers related to vasoactive neurochemicals[30], including dopamine (DA)[31,32], acetylcholine (ACh)[33–35], adenosine[36], and opioid neuropeptides[37]. While the contribution of GABAergic neuron activity and neurochemical release on hemodynamic activity has been documented in cortex[12,38–40], these relationships are less well understood in striatum. Thus, applying the same assumptions to cortical and striatal fMRI data may lead to inaccurate inferences regarding neuronal activity or limit the interpretability of results from human subjects[41].

Previous studies have revealed that heightened neuronal activity does not consistently lead to positive hemodynamic responses in the striatum. In rodent experiments, both positive[42–50] and negative[51–57] hemodynamic responses in the CPu have been observed where increased neuronal activity was either directly measured or inferred by selective manipulations (i.e., optogenetics). Intriguingly, the occurrence of negative hemodynamic responses in the CPu cannot be easily attributed to the activation of inhibitory neurons, as several studies have reported positive local hemodynamic responses following the selective activation of cortical inhibitory neurons[12,30,38–40,49,58,59]. Consequently, it becomes apparent that the variable hemodynamic responses observed in the CPu may depend on several factors, including the choice of anesthetics, the physiological condition of the subject, the prevailing brain state, as well as the frequency and amplitude of the targeted modulations. In essence, these studies collectively underscore that factors beyond mere neuronal activity might exert a significant influence on striatal hemodynamic responses, which likely include vasoactive neurochemicals.

To better understand the neural circuit and neurochemical bases of striatal fMRI signals, we conducted a comprehensive assessment of CPu fMRI responses to optogenetic stimulation of CPu neurons or afferents in rats. We demonstrate that CPu neuronal activity increases are sufficient to evoke negative CBV responses, as is stimulation of numerous CPu input regions (e.g., motor cortex (M1), parafascicular thalamus (PfT), external globus pallidus (GPe)). Spontaneous peaks in CPu neuronal activity were also associated with negative hemodynamic signals in awake, behaving rats. Interestingly, negative CBV responses evoked by CPu neuronal stimulation were attenuated by local infusions of opioid receptor antagonists, whereas direct application of an opioid receptor agonist to brain slices induced CPu vasoconstriction. These findings suggest that striatal opioidergic transmission may play a critical role in generating fMRI signals in striatum. Dopaminergic transmission is not critically involved in CPu negative CBV responses, but is instead associated with positive CBV responses. Lastly, we provide evidence that negative fMRI responses may occur in human striatum under certain conditions in which striatal activity may increase. These results suggest that, beyond neuronal activity changes, neurochemical transmission in striatum can play an outsized role in striatal fMRI signal generation.

## Results

### Optogenetic stimulation evokes pathway-specific fMRI responses in CPu

To disentangle the contribution of individual neuronal pathways within the complex CPu macro-circuitry, we performed a series of optogenetic-fMRI experiments in lightly anesthetized rats, using ChannelRhodopsin-2 (ChR2) stimulation and cerebral blood volume (CBV) fMRI readouts. All fMRI scans were acquired with a single shot, gradient echo EPI sequence on a 9.4 T system. We first stimulated neurons in three extra-striatal target regions that densely innervate CPu (Fig. S1a–c): M1, PfT, and GPe. M1 and PfT output contribute to the glutamatergic corticostriatal and thalamostriatal pathways, respectively[29], while GPe provides reciprocal inhibitory input to CPu via arkypallidal cells[60]. Optogenetic stimulation was applied to each of these target regions using a 40 Hz pulse train, chosen not only for its

alignment with low gamma frequency and its established contribution to the BOLD signal[61] but also based on our prior studies showing pronounced modulation of fMRI signals within the CPu when employing this particular frequency[62,63]. This stimulation paradigm yielded local positive CBV changes at the stimulation site while consistently eliciting negative CBV responses within the CPu across all cases (Figs. 1a–c, S2a–c, S3a–c). Furthermore, we extended our findings by demonstrating that negative CBV responses within the CPu could also be evoked at lower stimulus frequencies in other cortical regions projecting to the CPu (Figs. S4, S5).

Stimulation of each of these target regions may have complex and varying effects on CPu neuronal firing based on factors such as neurotransmitter released (glutamate vs GABA), and projection patterns, including differential inputs to both MSNs and striatal interneurons[64,65]. Thus, we next sought to determine if direct optogenetic stimulation of MSNs is sufficient to elicit similar negative CBV responses. Using a CaMKIIα promoter to selectively target ChR2 to striatal MSNs (Fig. S1d–g), we indeed observed striatal CBV decreases in response to optogenetic stimulation of MSN cell bodies (Figs. 1d, S2d, S3d) or bilateral stimulation of the striatal direct pathway terminals within the substantia nigra pars reticulata (SNr) (Figs. 1e, S2e, S3e). Specifically, stimulation of CPu terminals at the SNr can evoke antidromic activities in direct pathway MSNs[66–68], while allowing the optical fiber to be placed farther away from the CPu region of interest. Notably, in each of these optogenetic fMRI experiments, the negative CBV response amplitude in CPu was anticorrelated with the positive responses reported outside CPu (Fig. 1a–e, right column). The observed negative correlations between CBV responses at various optogenetic stimulation targets and the CPu suggest a "negative coupling" relationship, indicating that the intensity of CPu vasoconstriction scales with the activation levels in its connected brain regions. This suggests that neurovascular coupling, manifested as a decrease in CBV when synaptic inputs to these regions or CPu cell bodies are activated, still maintains an activity-dependent relationship. These effects were not observed in corresponding control subjects that did not express ChR2 (Figs. 1, S2), ruling out the possibility that these experimental results were due to local heating or light-evoked responses[69–71].

We observed negative CBV responses in CPu to optogenetic stimulation of primarily glutamatergic (i.e., PfT, M1) and GABAergic (i.e., GPe, CPu cell bodies or terminals) targets; nevertheless, it has been established that optogenetic stimulation of midbrain dopaminergic neurons in the ventral tegmental area evokes positive hemodynamic responses in ventral striatum (or nucleus accumbens) that rely on DA receptor signaling[63,72,73], suggesting the potential for a different activity-dependent vascular signaling mechanism involving DA[74,75]. To address this possibility, we performed bilateral, DA-neuron selective, optogenetic stimulation of the substantia nigra pars compacta (SNc) during CBV-fMRI (Fig. S6). This stimulation paradigm produced robust positive CBV changes in CPu for ChR2 experimental subjects but not eYFP controls (Figs. 1f, S2f). However, a correlation between SNc and CPu peak-response amplitudes could not be calculated due to the lack of a significant local response in SNc (Figs. 1f, S3f).

### Optogenetic stimulation of PfT and CPu terminals in SNr evokes increased neuronal activity, but not high levels of phasic DA, in the CPu

For many of the optogenetic manipulations in which we observed negative CBV responses in the CPu, one may anticipate net increases in overall CPu neuronal activity. Increases in CPu neuronal activity have previously been observed under conditions that elicit negative CPu fMRI signals (e.g., noxious forepaw stimulation, seizures)[54,55,76], however our current observations provide more direct causal evidence that CPu activity increases can induce vasoconstriction. To characterize the neuronal responses related to negative CBV responses

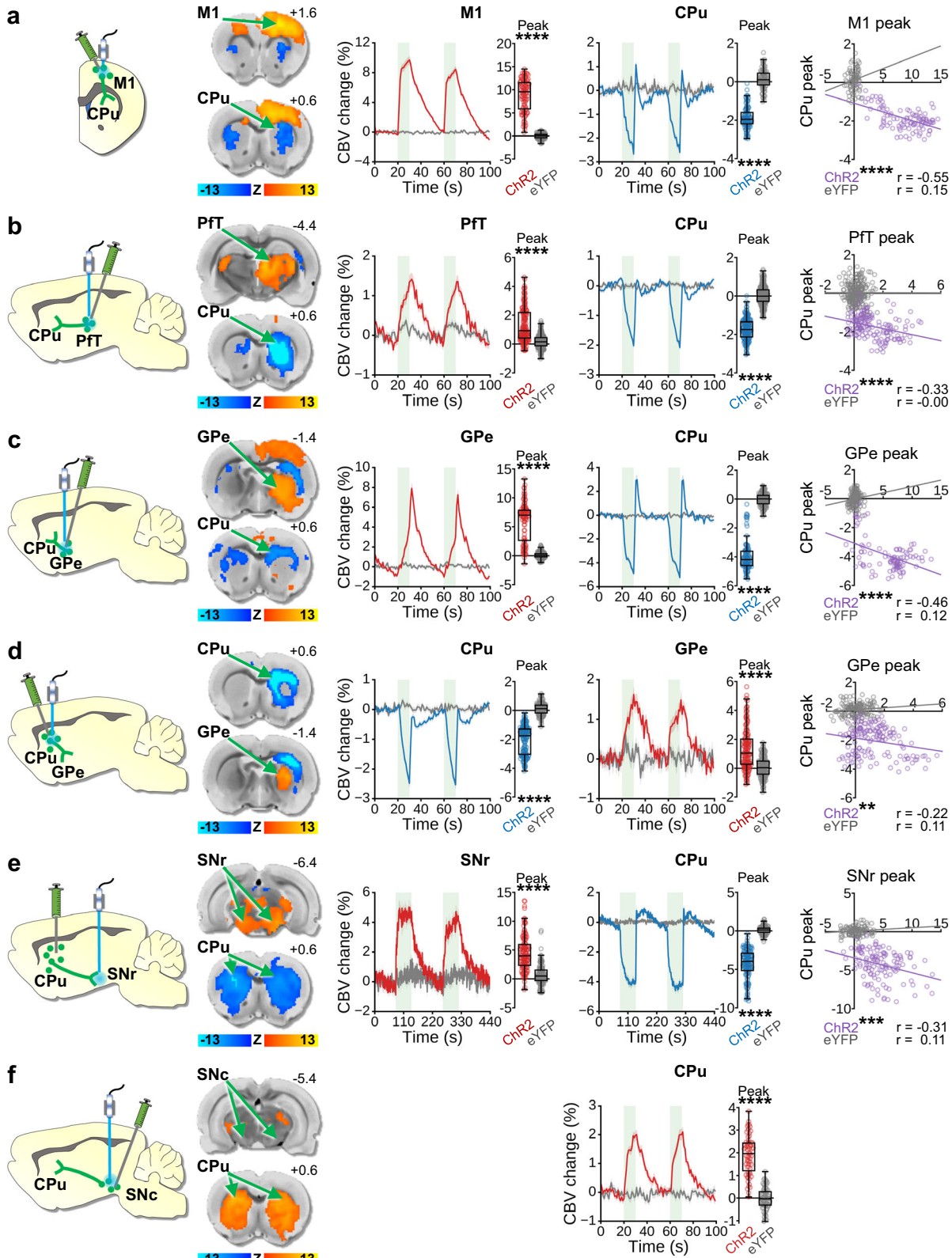

observed in Fig. 1, we reproduced a subset of the optogenetic stimulation paradigms while recording CPu neuronal activity with acute in vivo electrophysiology in lightly anesthetized rats.

Electrophysiological data were recorded using a 16-channel array acutely implanted across multiple CPu depths per subject, and optical fiber placements were confirmed via post-mortem histological examination (Fig. S7a, b). Local field potentials (LFPs) representing synaptic neuronal input, and multi-unit spiking activity (MUA) representing somatic neuronal output were collected from each recording. Stimulating PfT (Fig. 2a and Fig. S7a) or MSN terminals in SNr (Figs. 2b, S7b) at 40 Hz increased the 40 Hz LFP power and MUA in CPu during the stimulation period as compared to baseline, and these changes were significantly larger for the ChR2 group than the eYFP control group (Fig. 2a, b). Thus, it is evident that positive neuronal responses

**Fig. 1 | Optogenetic stimulation of CPu-related circuits during fMRI.** Left to right columns: Stimulation schematics, where viral expression and targeted projections are indicated in green (not all projections shown) and optogenetically stimulated areas are indicated in blue; CBV Z-score response maps from ChR2 subjects, with stimulation target response maps on top and stimulation projection response maps beneath; CBV time-courses from stimulation target response maps aligned to stimulation epochs (green bars indicate 40 Hz optogenetic stimulation blocks), with corresponding quantified peak amplitude changes; CBV time-courses from stimulated projection response maps with corresponding quantified peak amplitude changes (ChR2 vs. eYFP two-tailed Welch's t-test, ****$p < 0.0001$); Correlation plots between peak CBV changes from the stimulation targets and stimulation projections (linear regression two-tailed t-test, **$p < 0.01$, ***$p < 0.001$, ****$p < 0.0001$). **a** M1 cortical stimulation (ChR2 $n = 4$ rats, 50 epochs, 100 peaks; eYFP $n = 7$ rats, 40 epochs, 80 peaks). **b** PfT stimulation (ChR2 $n = 4$ rats, 80 epochs, 160 peaks; eYFP $n = 4$ rats, 115 epochs, 230 peaks). **c** GPe stimulation (ChR2 $n = 5$ rats, 50 epochs, 100 peaks; eYFP $n = 8$ rats, 68 epochs, 136 peaks). **d** Local CPu stimulation (ChR2 $n = 5$ rats, 75 epochs, 150 peaks; eYFP $n = 6$ rats, 69 epochs, 138 peaks). **e** Bilateral CPu-to-SNr stimulation (ChR2 $n = 18$ rats, 64 epochs, 128 peaks; eYFP $n = 5$ rats, 59 epochs, 118 peaks). **f** Bilateral SNc dopaminergic stimulation (ChR2 $n = 4$ rats, 32 epochs, 64 peaks; eYFP $n = 4$ rats, 35 epochs, 70 peaks). Note that multiple circuits show a post-stimulus "overshoot" or faster offset times than others. The result for each circuit likely reflects an accumulated confluence of vasoconstrictive and dilative forces operating at different time scales over the course of the stimuli, including metabolism, activated synapses, and vasoactive neurotransmission. Time-course data are presented as mean ± SEM. Box plots span IQR, with median line and whiskers within bounds ±1.5 IQR, using Tukey's method. Exact $p$-values and test statistics are in Source Data. Source data are provided as a Source Data file.

accompany the negative hemodynamic responses produced by these stimulation paradigms.

If CPu neuronal responses are sufficient to explain CPu fMRI responses, then one would expect the positive CPu fMRI response we observed with SNc dopaminergic stimulation to be concurrent with a net decrease in CPu neuronal activity. This is plausible given that DA can inhibit neuronal activity via D2 receptors. However, the relationship between DA and neuronal activity in the CPu may be complex as DA can also enhance activity via D1 receptors and multifaceted brainwide neuromodulatory effects of DA have been reported[77]. Indeed, studies in rats[78,79] and cats[80,81] have shown that the effects of exogenous DA on striatal neuronal activity can vary, with some studies showing excitation, inhibition, no effect, or mixed-effects. Nevertheless, replication of the 40 Hz SNc stimulation paradigm from Fig. 1 under the previously described in vivo electrophysiology recording setting revealed significant increases in 40 Hz LFP power and MUA in CPu for the ChR2 group as compared to eYFP controls (Figs. 2c, S7c). Collectively, we observed increases in CPu neuronal activity for stimulation paradigms that produced both negative and positive CPu fMRI responses, thus CPu neuronal activity alone is insufficient to account for the observed fMRI responses.

To further probe the circuit mechanisms engaged by our optogenetic stimulation paradigms, we examined the change in MUA spike probability with each light pulse relative to baseline (Fig. S7, right column). Overall, spike probability changes observed in the eYFP control groups were smaller than those in the ChR2 groups by at least one order of magnitude. Interestingly, PfT stimulation (Fig. S7a) and MSN terminal stimulation at SNr (Fig. S7b) in the ChR2 group produced similar single-pulse response profiles, suggesting comparable recruitment of striatal neurons by both stimulation targets. In contrast, SNc dopaminergic stimulation primarily enhanced activity between stimulation pulses (Fig. S7c), more consistent with the delay of dopaminergic rather than glutamatergic or GABAergic signaling[79,82].

Finally, to assess the potential contribution of DA signaling to fMRI responses in CPu, we used high-resolution fast-scan cyclic voltammetry (FSCV) to obtain simultaneous local tissue-oxygen and DA concentration changes for the SNc DA neuron (Fig. S7d), PfT (Fig. S7e), and MSN terminals in SNr (Fig. S7f) stimulation paradigms. For each FSCV recording, a carbon-fiber microelectrode was lowered into the CPu of a lightly anesthetized rat (Fig. S7g), and a voltage waveform spanning the oxidation and/or reduction potentials of oxygen and DA was scanned at the electrode to detect concentration-dependent changes in current (Fig. 2d)[83]. As expected, stimulating SNc DAergic neurons evoked significant DA and tissue-oxygen increases in ChR2 over eYFP controls within CPu (Fig. 2e). In contrast, PfT stimulation produced a significant negative change in peak tissue-oxygen, but no significant change in peak DA concentrations compared to eYFP controls (Fig. 2f). Similarly, stimulation of MSN terminals in SNr produced a significant negative change in peak tissue-oxygen compared to eYFP controls, but also a small-but-significant increase in peak DA

concentration over an order-of-magnitude less than observed with SNc DA neuron stimulation (Fig. 2g). Collectively, our tissue-oxygen findings agree with the CBV-fMRI responses obtained from the same stimulations, while our DA measurements indicate that positive, but not negative, CBV/oxygen responses in CPu are strongly associated with micromolar concentrations of DA release.

## Spectral fiber-photometry in awake rat reveals simultaneous negative CBV and positive neuronal activity changes in CPu

While optogenetic stimulation enabled us to precisely control neural circuits capable of driving negative hemodynamic responses in CPu, the synchronized activity-induced by such artificial stimulations is unlikely to occur naturally, and at certain stimulation targets may drive neuronal spiking rates outside of physiological ranges[84]. Further, although the light anesthesia protocol employed for the aforementioned recordings is well-established in rodents to facilitate reproducible neuronal and vascular responses and stable physiology[85–89], it is widely known that anesthetics can alter these metrics[89–92]. To examine the relationship between CPu hemodynamics and neuronal firing under more naturalistic conditions, we employed spectral fiber-photometry to simultaneously measure neuronal and vascular activity[51,93–96] in the CPu of freely moving awake rats (Fig. 3a).

GCaMP6f was expressed in CPu under the CaMKIIα promoter, allowing us to selectively record MSN activity. Temporally interleaved 405 and 488 nm light pulses were delivered via a single optical fiber to excite GCaMP, and the emission signal from these excitation pulses was used to derive CBV from the total hemoglobin absorption and GCaMP activity, respectively (Fig. 3a)[51,93]. We recorded responses to 1 s footshocks in awake rats and observed concurrent increases in CPu GCaMP activity and decreases in CBV (Fig. 3b). In addition, we extracted significant spontaneous GCaMP signal peaks in the absence of any stimulation and found the same general relationship (Fig. 3c). Collectively, the correspondence between positive neuronal and negative vascular activity in the CPu observed with fiber-photometry validates the relationship inferred from our independent fMRI and electrophysiology results, and confirms that it also translates to awake, freely moving conditions. Importantly, our analysis of spontaneous GCaMP activity also suggests that the negative vascular relationship to CPu neuronal activity is not unique to the optogenetic stimulation parameters and paradigms chosen for the fMRI and electrophysiology studies.

## Negative CPu CBV responses in fMRI are attenuated by local pharmacological disruption of MSN activity and opioidergic neurotransmission

It is generally accepted that positive changes in neuronal activity and fMRI signal are coupled, as regional blood flow increases to provide sufficient local energy supplies to satisfy metabolic demand; this is known as neurovascular coupling[97]. However, this is insufficient to explain the neurovascular relationship in CPu, as shown under our

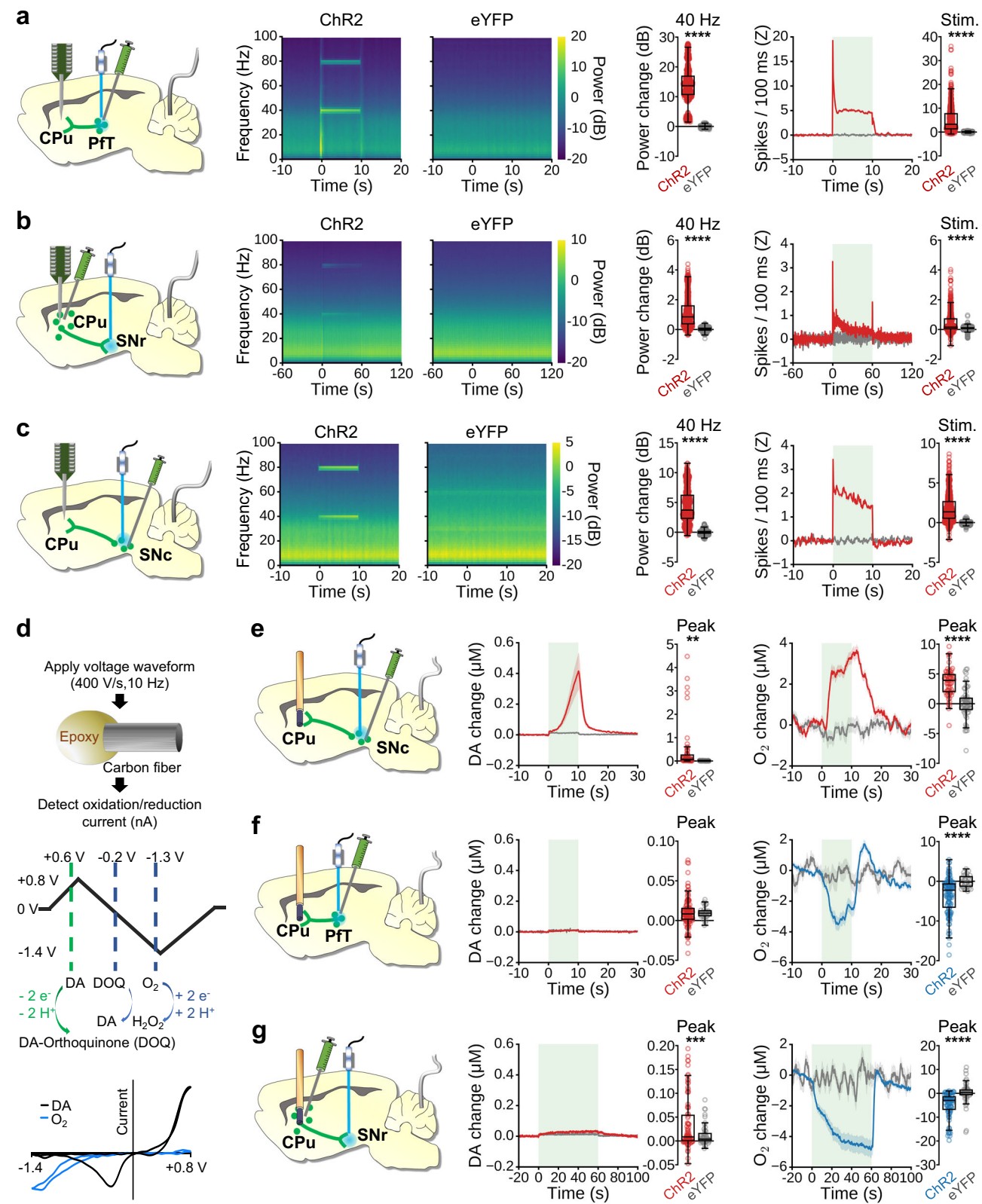

experimental conditions. Although it was originally believed that neurovascular coupling was dictated by metabolic feedback mechanisms, increasing experimental evidence illustrates the dominant role of activity-dependent, feed-forward mechanisms instead[98,99]. Neurochemicals released via neuronal activity can alter vascular tone either directly by interacting with vascular smooth muscle cells or pericytes or indirectly through astrocytes, endothelial cells, and/or

interneurons[98,100,101]. To evaluate whether activity-dependent, feedforward neurochemical release could be responsible for the vasoconstriction we have observed in CPu, we performed intra-CPu pharmacology experiments with optogenetic fMRI by stimulating CPu terminals in SNr.

Rats were implanted with MRI-compatible guide cannula above the CPu for acute drug delivery during fMRI, and prepared as

**Fig. 2 | Optogenetic stimulation of CPu-related circuits during acute in vivo electrophysiology and FSCV. a–c** Left to right: Experimental schematic indicating CPu electrode array, reference wire, and optogenetic viral expression (green) and stimulation site (blue); Local field potential (LFP) perievent spectrograms; Quantified LFP power changes at 40 Hz stimulation frequency (ChR2 vs. eYFP two-tailed Welch's *t* test, ****$p$ < 0.0001); Multiunit activity (MUA) peri-event time-courses (green box indicates the stimulation block), and average firing rate over the stimulation blocks (ChR2 vs. eYFP two-tailed Welch's t test, ****$p$ < 0.0001). **a** 10 s PfT stimulation (ChR2 $n$ = 4 rats, 576 LFP/575 MUA recordings; eYFP $n$ = 4 rats, 528 LFP/ 520 MUA recordings). **b** 60 s stimulation of MSN terminals in SNr (ChR2 $n$ = 4 rats, 528 LFP/528 MUA recordings; eYFP $n$ = 4 rats, 464 LFP/456 MUA recordings). **c** 10 s SNc dopaminergic neuron stimulation (ChR2 $n$ = 4 rats, 624 LFP/624 MUA recordings; eYFP $n$ = 4 rats, 656 LFP/596 MUA recordings). **d** FSCV waveform applied to simultaneously detect local DA and tissue-oxygen, with voltages relative

to Ag/AgCl reference. **e-f** Left to right: Schematic of optogenetic viral expression (green), stimulation site (blue), and acute CPu FSCV electrode; Time-courses for DA with peak values during optogenetic stimulation; Time-courses for oxygen changes in CPu with peak values during optogenetic stimulation (green box indicates stimulation block; ChR2 vs. eYFP two-tailed Welch's *t* test, **$p$ < 0.01, ***$p$ < 0.001, ****$p$ < 0.0001). **e** 10 s ipsilateral SNc dopaminergic neuron stimulation (ChR2 $n$ = 5 rats, 61 epochs/peaks; eYFP $n$ = 4 rats, 61 epochs/peaks). **f** 10 s ipsilateral PfT stimulation (ChR2 $n$ = 9 rats, 115 epochs/peaks; eYFP $n$ = 5 rats, 30 epochs/peaks). **g** 60 s ipsilateral stimulation of MSN terminals in SNr (ChR2 $n$ = 7 rats, 88 epochs/ peaks; eYFP $n$ = 5 rats, 50 epochs/peaks). Time-course data are presented as mean ± SEM. Box plots span IQR, with median line and whiskers within bounds ±1.5 IQR, using Tukey's method. Exact $p$-values and test statistics are in Source Data. Source data are provided as a Source Data file.

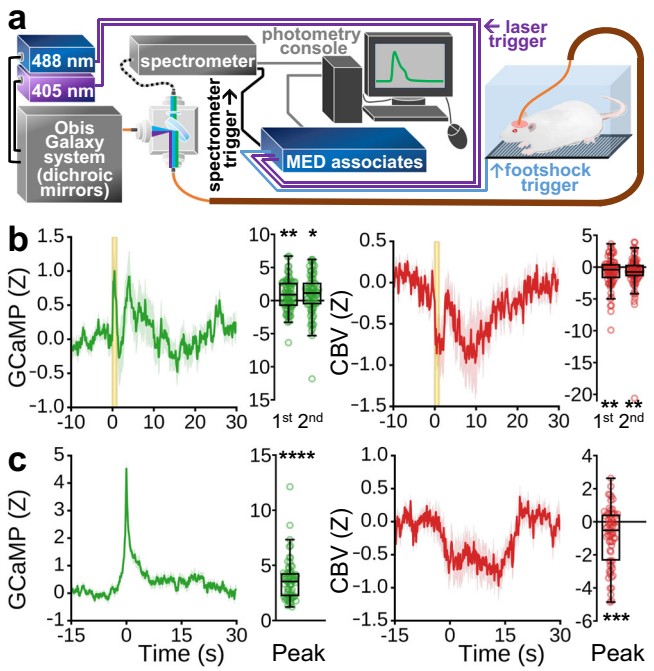

**Fig. 3 | Simultaneous calcium activity and CBV measurements in awake, freely moving rats using spectral fiber-photometry. a** Fiber-photometry instrumental setup. **b** Neuronal (left) and vascular (right) activity time-locked to pseudorandomly presented 1 s footshocks ($n$ = 9 rats, 79 footshocks total). Two prominent GCaMP and CBV peaks were visually identified and the response amplitude for each was quantified to the right of their respective time-courses (one-sample two-tailed t-tests: GCaMP 1st peak, GCAMP 2nd peak, CBV 1st peak, CBV 2nd peak; *$p$ < 0.05, **$p$ < 0.01). **c** Average of strong spontaneous GCaMP peaks exceeding 97.7th percentile amplitude, and at least 30 s from neighboring peaks, from animals at rest (left), and (right) average vascular response from the same time-windows ($n$ = 9 rats, 66 peaks), with respective peak amplitude quantification (one-sample two-tailed t-tests: GCAMP peak, CBV peak; ***$p$ < 0.001, ****$p$ < 0.0001). The footshock and spontaneous peak activation data, after correcting for sensor cross-talk[51,93], generally exhibit opposing response polarities in GCaMP (mostly above zero) and CBV (mostly below zero) signals, suggesting atypical neurovascular coupling. Nevertheless, there are also notable non-opposing features between footshock GCaMP and CBV responses (e.g., t = -5 s and -15–30 s). This observation underscores the intricate coupling relationship between neuronal and vascular activities in more naturalistic conditions. Additionally, while motion confounds may be present in our footshock data, motion typically induces same directional changes in both the green and red spectra[216], so the generally opposing polarity of GCaMP and CBV signals observed here, particularly at stimulation onset, argues against motion as a major contributing factor in these signals. Time-course data are presented as mean ± SEM. Box plots span IQR, with median line and whiskers within bounds ±1.5 IQR, using Tukey's method. Exact $p$-values and test statistics are in Source Data. Source data are provided as a Source Data file.

described above for bilateral optogenetic stimulation of CPu projection terminals in SNr (Fig. 4a). This stimulation strategy, as shown in Fig. 1e, was chosen to accommodate intra-CPu infusion cannula (i.e., far from stimulating fibers), and avoid potential stimulation artifacts in the CPu. Fiber and cannula placements were confirmed with high-resolution anatomical MRI (Fig. S8). Each scan consisted of 15 stimulation blocks. The first 5 stimulation blocks ("pre-drug") were followed by unilateral drug infusion, and pre-drug evoked response amplitudes were compared to those of the last 5 stimulation blocks ("post-drug") from within the pre-drug CPu response area (Fig. S9) to determine drug effect. Drug delivery was confirmed through changes in fMRI baseline signal intensity (Fig. S10)[51,96]. We observed consistent negative CBV responses in CPu to optogenetic stimulation (Fig. 4b). The significance of all drug effects were determined relative to saline vehicle (Fig. 4b). Though it is known that pH fluctuations can influence neuronal activation[102], no correlation was observed between any drug effect and its solution pH (Fig. S10k).

First, to corroborate our electrophysiology and photometry findings, we disrupted all local CPu neuronal activity by blocking voltage-gated sodium channels with lidocaine (Fig. 4c, l). Lidocaine significantly attenuated the negative CBV response, confirming that local neuronal activity is necessary. Next, we pharmacologically targeted several neurochemical pathways that are present in CPu, many of which with known vasomodulatory properties in CPu or elsewhere in brain. Targeting the cholinergic system, we observed that the muscarinic ACh receptor agonist oxotremorine-M significantly attenuated the negative CBV response (Fig. 4d, l), but infusion of ACh itself did not (Fig. 4e, l). Antagonism of somatostatin or neuropeptide-Y receptors with cyclo somatostatin (Fig. 4f, l) and BIBP-3226 (Fig. 4g, l), respectively, had no significant effect on the stimulus-evoked negative CBV response. A similar lack of effect was also observed with intra-CPu infusions of caffeine (Fig. 4h, l), an adenosine A2A receptor antagonist, or a cocktail of D1 and D2 DA receptor antagonists SCH-23390 and raclopride (Fig. 4i, l). Strikingly, we observed marked reduction of the negative CBV stimulation-response in CPu with intra-CPu infusions of opioid receptor antagonists, including nor-binaltorphimine, which preferentially targets κ (KORs) and μ opioid receptors (MORs) (Fig. 4j, l)[103], and by naltrindole targeting δ opioid receptors (DORs) (Fig. 4k, l).

Our findings indicate that, in CPu, opioidergic signaling is involved in negative CBV responses to increases in neuronal activity. To corroborate these findings, we examined the direct effects of the DOR/MOR agonist[104,105], Enk analog[106], D-Ala², D-Leu⁵]-Enkephalin (DADLE) on microvessel diameter in CPu with bright-field microscopy of ex vivo mouse brain slices (Fig. 5). In our preparation, we first added thromboxane A₂ (TXA2) to the perfusion of oxygenated artificial cerebral spinal fluid (aCSF) to establish a more physiological vascular tension and select viable vessels[107–109]. As expected, viable vessels visually constricted to TXA2, and within our sample this change in diameter from aCSF alone was significant (Fig. 5a). Notably,

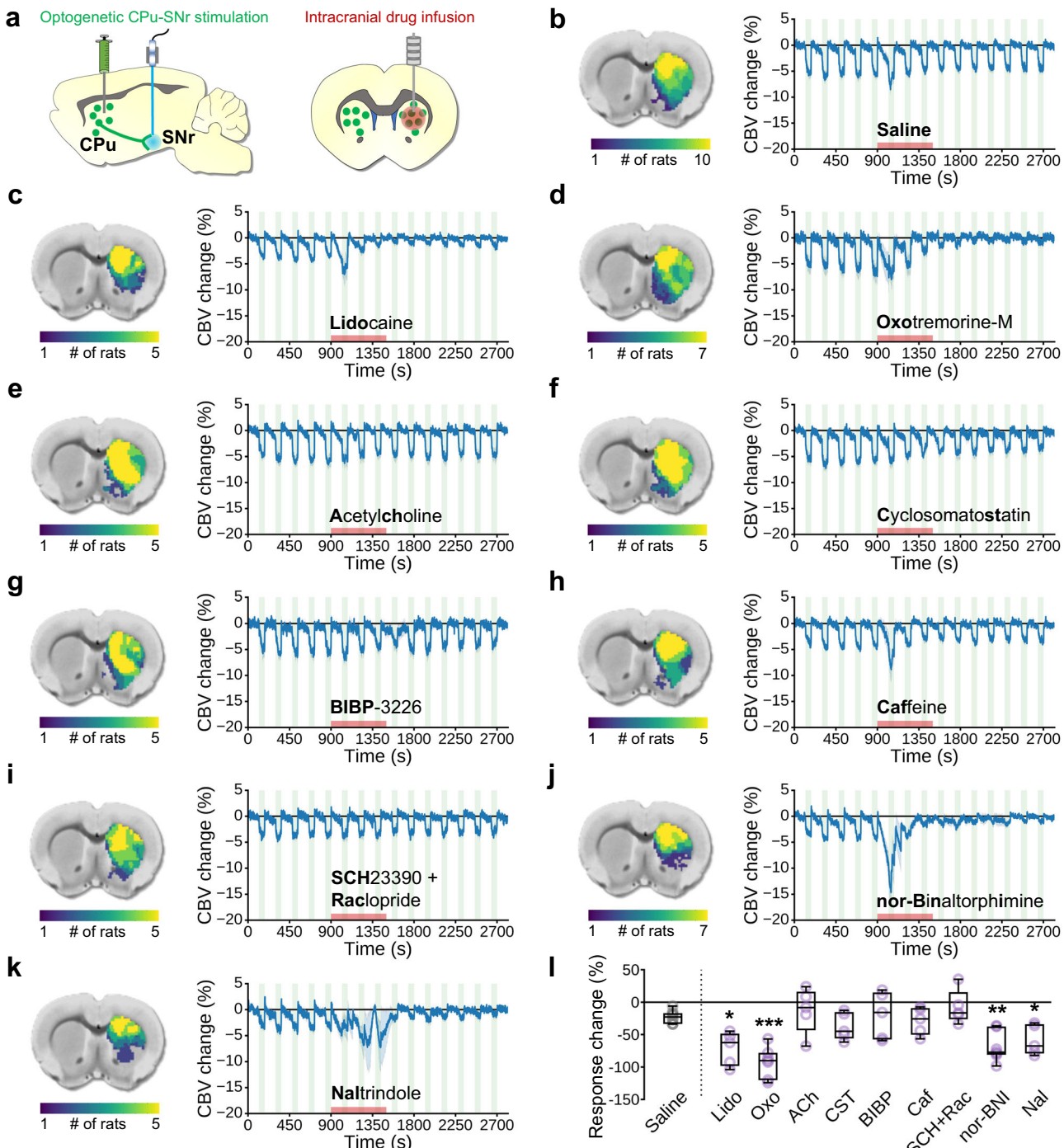

**Fig. 4 | The impact of intra-CPu drug infusions on evoked negative CPu fMRI responses. a** Bilateral optogenetic stimulation of MSN terminals in SNr. A unilateral infusion cannula was used for acute drug delivery into the CPu. **b–k** (left) Subject-wise CBV response masks obtained from pre-drug optogenetic SNr stimulation responses within the CPu. **b–k** (right) CBV time-courses extracted from subject-wise response masks tracking optogenetic SNr stimulation epochs before, during, and after intra-CPu drug infusion. Drug name is indicated under the time-course (bold lettering for abbreviations used in (**l**)), stimulation blocks are indicated by green shaded boxes, and the drug infusion time is indicated by the red shaded box. **l** Effect of intra-CPu drug infusion on negative CPu responses, relative to the saline vehicle control effect. Values are presented as % change in averaged peak-response

CBV changes between the last 5 stimulation blocks ("post-drug") relative to the first 5 stimulation blocks ("pre-drug") for each rat. Welch's ANOVA ($W = 8.265$, DFn = 9.0, DfD = 17.14; $p = 0.001$), with planned comparisons versus Saline (Saline, $n = 10$ rats; Lido, $n = 5$ rats; Oxo, $n = 7$ rats; Ach, $n = 5$ rats; Cst, $n = 5$ rats; BIBP, $n = 5$ rats; Caf, $n = 5$ rats; SCH + Rac, $n = 5$ rat; nor-BNI, $n = 7$ rats; Nal, $n = 5$ rats), FDR corrected by the two-stage linear step-up procedure (*$q < 0.05$, **$q < 0.01$, ***$q < 0.001$). Time-course data are presented as mean ± SEM. Box plots span IQR, with median line and whiskers within bounds ±1.5 IQR, using Tukey's method. Exact $q$-values and test statistics are in Source Data. Source data are provided as a Source Data file.

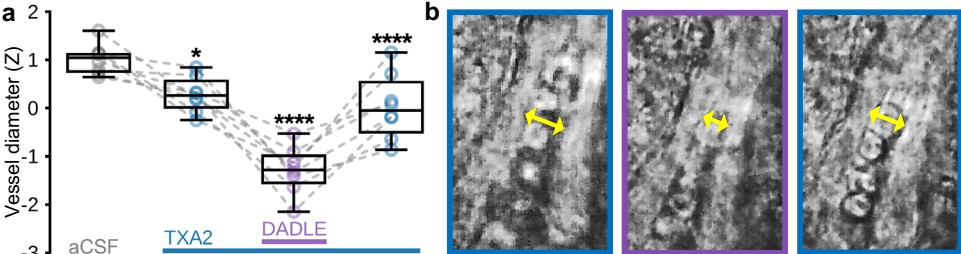

**Fig. 5 | Bright-field microscopy reveals acute constriction of CPu microvessels to Enk analog in mouse brain slices. a** Average microvessel diameter during 5 min experimental epochs (in order: aCSF (baseline), TXA2 + aCSF (pretreatment), DADLE + TXA2 + aCSF (Enk analog), TXA2 + aCSF (washout)) for 8 CPu microvessels ($n$ = 3 mice). Šídák's multiple comparisons test versus preceding experimental condition; *adjusted $p$ < 0.05, ****adjusted $p$ < 0.0001. **b** Representative microvessel showing constriction of vessel diameter (indicated by yellow arrows) to DADLE (purple frame) versus TXA2 pretreatment and washout (left and right blue frames, respectively). Box plots span IQR, with median line and whiskers within bounds ±1.5 IQR, using Tukey's method. Exact $p$-values and test statistics are in Source Data. Source data are provided as a Source Data file.

subsequent addition of DADLE significantly constricted vessels further, then washout of DADLE was accompanied by a significant relaxation back to pre-DADLE tone. These data suggest that acute feed-forward opioid signaling can transiently induce vasoconstriction in CPu.

## Noxious peripheral and targeted cortical stimulations evoke negative fMRI responses in human striatum

To determine whether our rodent model findings could be used to inform human fMRI data interpretation, we measured blood oxygen level dependent (BOLD) fMRI signals in the striatum (caudate, putamen, ventral striatum) (Fig. 6a) of awake human subjects during stimulation patterns in which increased striatal activity may occur. Noxious electrical stimulation of the rat forepaw induces a negative CPu fMRI signal (Fig. S11), which corresponds to heightened CPu neuronal activity[55,76]. We employed a similar stimulation paradigm with awake human subjects, utilizing bilateral noxious transcutaneous electrical stimulation of the median nerve. Seven healthy right-handed participants, free from acute or chronic pain and drug usage affecting pain perception or the central nervous system, were included. Stimulation intensities were individually determined based on sensory and pain thresholds for each subject. The fMRI data were acquired using a conventional gradient echo EPI sequence on a 7T system, and the experimental design during fMRI scans followed a block structure that incorporated baseline, stimulation, and rest periods. This stimulation elicited a prominent negative bilateral response in striatum (Figs. 6b, S12), similar to the response pattern in rats (Fig. S11) but more localized to the caudate subregion of the striatum. Next, we examined whether targeted activation of distinct cortical regions using transcranial magnetic stimulation (TMS) during fMRI[110] could result in negative striatal fMRI signals. As part of a previous project, with data publicly available at the NIMH Data Archives (NIMH Data Archive Collection ID: 2856), eighty-two healthy individuals without a history of psychopathology underwent TMS-fMRI scanning using a gradient echo spiral in/out pulse sequence on a 3T scanner. Prior to the scanning session, TMS stimulation sites in right anterior middle frontal gyrus (aMFG), posterior middle frontal gyrus (pMFG), and M1 were defined using high-resolution T1-weighted anatomical images, and coordinates were transformed into subject-native space. Motor threshold was determined for each participant by identifying the lowest stimulation intensity that induced a visible muscle twitch response in the contralateral abductor pollicis brevis muscle. Concurrent TMS-fMRI sessions were conducted with a custom-built TMS-compatible head-coil, covering the whole brain with 31 slices and acquiring data in a fast event-related design with heart rate and respiration monitoring. Our original analyses of these data revealed that TMS of the right M1 (Figs. 6c, S13), aMFG (Figs. 6d, S14), and pMFG

(Figs. 6e, S15) respectively produced negative BOLD responses localized to the ventral, caudate, and putamen subregions of the striatum. These findings suggest that negative hemodynamic responses in the rat CPu and at several locations within the human striatum can be observed when manipulating afferent regions, thereby indicating the possibility of shared characteristics in the vascular responses of the striatum/CPu across species.

## Discussion

Our results provide a causal demonstration that CPu vasoconstriction can be accompanied by increases in CPu neuronal activity, which may include pre- and/or post-synaptic activity changes. Direct optogenetic excitation of CPu neurons reliably elicited negative CBV responses in rats, and similar negative signals were observed in response to optogenetic stimulation of M1, GPe, and PfT, three brain regions with notable projections to CPu. The negative CBV changes (elevated T2* weighted signal in raw data) observed in our fMRI experiments are unlikely to be driven by positive BOLD signals, given our high contrast agent dosing (30 mg/kg) and validation of hemodynamic response polarity through complementary techniques. Prior studies employing sequential BOLD and CBV measurements during identical stimulation paradigms have demonstrated minimal influence from the BOLD effect. For instance, Lu et al. (2007)[111] conducted rat paw stimulation experiments at 9.4 T, employing iron contrast agent doses ranging from 5–30 mg/kg. They observed a notable BOLD influence to CBV measurement at lower iron doses (e.g., 5 mg/kg) but not at higher doses (15–30 mg/kg), indicating that as contrast agent concentration increases, the impact of fully oxygenated arterial blood on voxel homogeneity diminishes. In order to illustrate the BOLD response pattern in an identical experimental context, we replicated the PfT optogenetic stimulation experiment without administration of CBV contrast agent to obtain BOLD fMRI as the readout. The corresponding BOLD fMRI maps, time courses, and relevant discussion can be found in Fig. S16.

We observed that certain circuit manipulations, like GPe stimulation, generated CPu fMRI responses with distinct peak timing and a robust post-stimulus overshoot (Fig. 1). These differences in response timing may arise from the non-uniform propagation of firing patterns across synapses and variations in vascular responses across brain regions. The overshoot likely results from the convergence of conflicting forces: vasoconstriction due to neurotransmission and dilation from metabolic feedback mechanisms, which may operate on different timescales and intensities. Interestingly, CPu vasoconstriction was observed with optogenetic activation of brain regions that varied in their principal neurochemical phenotype (e.g., glutamatergic or GABAergic) and stimulation-induced brain-wide fMRI response patterns (Figs. 1, S2). Indeed, the most notable feature among these

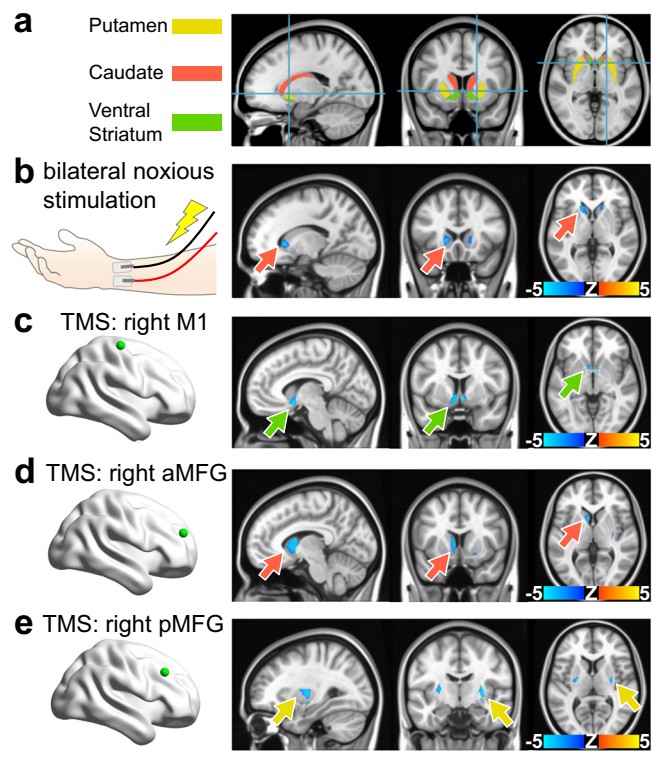

**Fig. 6 | Negative fMRI responses can be evoked in human striatum via noxious peripheral and selective afferent stimulations. a** Human striatum anatomical mask used for subsequent panels, color-coded by striatal compartment. **b** Noxious forearm stimulations given to human subjects and the corresponding BOLD fMRI response maps (n = 7 subjects, 11 scans). Noxious 5 Hz, 0.5 ms pulse width, stimulation with intensity determined by subject pain rating was delivered in 20 s intervals, repeated over 6 trials flanked by 30 s rest. These findings indicate that anticipated noxious stimulation in human subjects and unanticipated footshock in awake rats (Fig. 3) may both contribute to shaping negative striatal hemodynamic responses. **c**–**e** Brain maps indicating a striatal projection stimulated in human subjects via transcranial magnetic stimulation (TMS) (green dot), and its corresponding BOLD fMRI response maps. 68 TMS pulses were presented with a variable inter-trial interval jittered with delays of 2.4, 4.2, and 7.2 s delivered over 6 min and 41 s (167 volumes) in a fast event-related design. **c** TMS of the right M1 (n = 79 subjects). **d** TMS of the right anterior middle frontal gyrus (n = 80 subjects). **e** TMS of the right posterior middle frontal gyrus (n = 79 subjects). Arrows indicate the voxel with most significant activation and match the striatal compartment color legend used in (**a**). Unmasked versions of the human fMRI response maps displayed here as well as corresponding time courses are shown for noxious forearm stimulation, M1 TMS, aMFG TMS, and pMFG TMS in Fig. S12, Fig. S13, Fig. S14, and Fig. S15, respectively.

regions is their shared monosynaptic connectivity to CPu, suggesting that this direct input may be a crucial factor for generating activity-induced CPu vasoconstriction. The observed responses to GPe stimulation further highlight the complexity of region-dependent fMRI signal changes. GPe neurons, which are largely GABAergic and inhibitory, send direct projections to both the CPu and STN[112], leading to the expectation that GPe has an impact on the activity of both CPu and STN. Notably these changes were observed in our data, albeit contributions from poly-synaptic pathways cannot be ruled out. CPu stimulation might be expected to predominantly decrease firing rates in GPe due to the extensive, inhibitory striatopallidal output[113], while direct excitatory optogenetic stimulation of GPe is expected to increase GPe neuronal activity. Despite these likely divergent changes in GPe firing activity, both manipulations drive GPe CBV increases. We speculate that both manipulations may lead to increased metabolic demand within the GPe, potentially stemming from either local GPe

stimulation-evoked neuronal activity or synaptic input activity in GPe induced by CPu stimulation[4,114,115]. These findings in GPe align with prior studies reporting positive hemodynamic responses following the selective activation inhibitory neurons in the cerebral cortex[12,30,38–40,49,58,59]. Nevertheless, they also bring attention to the intriguing observation of negative CBV responses in CPu during the same experimental session, despite CPu neurons sharing a predominately inhibitory nature with GPe[30].

Stimulation of CPu projection terminals in SNr produced negative CPu fMRI signals, a stimulation condition which, alongside with PfT excitation, was demonstrated to drive increases in striatal MUA and LFP power at the optogenetic pulse frequency. It is worth noting that optogenetic stimulation of CPu projection terminals in SNr resulted in comparatively weaker CPu neuronal responses, possibly due to antidromic stimulation activating only approximately 50% of MSNs (i.e., the direct pathway MSNs)[30]. The involvement of local inhibitory potentials and hyperpolarization could further complicate the generation of the fMRI signal. Additionally, it should be noted that optogenetic stimulation of CPu terminals in SNr can influence CPu activity through various circuit mechanisms, including polysynaptic routes involving thalamus and cortex. Indeed, our M1 and PfT optogenetic stimulation data support the idea that corticostriatal and thalamostriatal inputs can induce CPu activity changes. We acknowledge that the slow hemodynamic signal time course and variations in neurovascular coupling among different brain regions may hinder the definitive identification of specific circuit-level mechanisms using the data collected in this study. Future investigations employing optogenetic inhibition[116] at cortical and thalamic sites, in addition to CPu terminal stimulation, may be necessary to further dissect these specific signaling pathways.

Local application of lidocaine in CPu abolished the negative CBV responses to stimulation of CPu projection terminals in SNr, providing further evidence that such vasoconstriction is dependent on CPu neuronal activity increases. Although these results were obtained in sedated rats, particularly at high stimulus frequencies, and optogenetic manipulations may drive non-physiological neural activity patterns[84], fiber-photometry recordings in awake and behaving rats suggested that such an inverse relationship of CPu neuronal activity and hemodynamic signals may also occur under more naturalistic conditions.

Although CPu activity increases can cause local vasoconstriction, it is evident that additional factors contribute to activity-dependent CPu hemodynamics. For example, we and others observed that midbrain DA cell stimulation elicits both increases in CPu neuronal activity[117] and positive CPu fMRI signals[63,72,118], demonstrating that increases in CPu firing activity alone may not be a reliable indicator of the directionality of CPu hemodynamic responses. Indeed, we show with CPu FSCV recordings that strong DA release is present for dopaminergic stimulation, but not stimulation paradigms that produced negative hemodynamic responses. In addition to their effects on neuronal cell firing, striatal neuromodulators such as DA may also exhibit vasoactive properties wherein they may more directly influence local hemodynamic signals. In CPu, dopaminergic terminals are proximal to blood vessels[119], and DA receptors are expressed on microvessels, endothelial cells, and astrocytes[32]. Direct effects of DA on brain hemodynamics have been suspected for decades[120], and the results of a pharmacological study by Choi et al. (2006)[32] indicate that D1R signaling dilates whereas D2R signaling constricts vasculature in CPu. It is thus possible that non-neuronal DA signaling, including direct vascular action, plays a major contribution for DA-dependent CPu fMRI signals.

In experiments combining optogenetic fMRI with intra-CPu pharmacology, we probed the roles of various striatal neuromodulator systems on CPu activity-dependent vasoconstriction. Many of the drugs used in our study could alter neuronal activity and

neurochemical concentrations within the CPu, including dopaminergic tone (e.g., via disruptions in opioidergic and/or cholinergic control of striatal dopamine release)[121–123], as well as possibly more direct actions on vascular signaling. Our results showed significant attenuation of negative hemodynamic responses by pharmacological inhibition of neuronal firing and blockade of opioidergic receptors. We also observed vasoconstriction of CPu blood vessels in mouse brain slices to the Enk analog, DADLE, suggesting that opioidergic involvement in CPu vasoconstriction may include direct action on the vasculature. To date, little is known about vasoactive signaling pathways for endogenous opioids released in the brain. There is limited evidence that opioids can directly signal to perivascular cell types that control vessel tone through vasomotor action or via vascular signaling cascades. One study in piglets found dilation in pial arteries by Enk and Dyn under normal conditions but constriction by Dyn under hypotension[124], and these effects were related to the concentration of prostanoids which can be released by astrocytes or endothelial cells and are known to regulate the dilation and constriction of blood vessels by signaling to vascular smooth muscle cells[98,125]. Another study found that Dyn can induce strong vasoconstriction of isolated rat cerebral arteries that is partially dependent on KOR binding[126]; although the mechanism is unknown, this effect could be mediated by the aforementioned endothelial or astrocyte signaling pathways as both express KORs[127,128], or through direct opioidergic constriction of smooth muscle cells as seen in the gastrointestinal system[129]. In addition, activation of DORs expressed on rat aortic vascular smooth muscle cells has been found to evoke vasoconstriction[130], and DORs and MORs can be expressed on endothelial cells and astrocytes[127,131–133], suggesting that Enk signaling to perivascular cell types could also regulate vascular tone directly or via vascular signaling cascades.

Our intra-CPu pharmacology results also included attenuation of negative hemodynamic responses by activation of muscarinic, but not all, ACh receptors. Several potential mechanisms could be responsible for these intriguing results. (1) Our choice of muscarinic ACh agonist, oxotremorine-M, has been shown to decrease MSN activity in vitro, which could occur due to preferential activation of inhibitory M4 receptors on striatonigral MSNs[134–136] and/or by inhibition of glutamatergic MSN input via M2 receptors[137,138]. Thus, like lidocaine, oxotremorine-M could have attenuated the negative hemodynamic response by reducing the direct neuronal effects of the optogenetic stimulation paradigm. (2) Muscarinic ACh signaling directly to the vascular unit or to interneurons that release vasoactive neurochemicals has been demonstrated to elicit vasodilation in cortex[139,140]. Therefore, an alternative explanation for our results is that vasodilation via activation of muscarinic ACh receptors in CPu may have counteracted the observed negative hemodynamic response; these processes may very well be independent. (3) Unlike the effects of oxotremorine-M, the local effects of acute ACh may have been too short-lived for detection by our pre- and post-drug experimental design due to rapid break-down of ACh by the relatively high levels of acetylchinesterase found in striatum[141]. While these scenarios do not point to ACh signaling as a key link between neuronal activation and negative hemodynamic responses in CPu, investigating the roles of specific ACh receptors in this context remains an important topic for future studies.

Lastly, to assess whether striatal activity-induced vasoconstriction may occur in the human brain, we took advantage of experimental stimulation conditions in which increases in striatal neuronal activity may be inferred. In rodents, noxious stimulation often leads to negative BOLD responses and vasoconstriction in CPu[142–149], which have been demonstrated to coincide with increased neuronal activity[55,76]. Remarkably, we observed a similar striatal negative BOLD signal in human participants undergoing transcutaneous, noxious electrical stimulation of the median nerve. Additionally, original analysis of data from a previous study, publicly available at the NIMH Data Archives

(NIMH Data Archive Collection ID: 2856), revealed that TMS directed to multiple regions of frontal cortex elicited negative striatal fMRI signals in human participants. Although striatal neuronal activity was not directly measured in these experiments, and TMS may in some cases involve a prominent inhibitory component[150,151], our findings provide early translational evidence that striatal fMRI signals may not reliably capture neuronal activity changes in the human brain.

We observed negative hemodynamic responses in CPu as a result of optogenetic stimulation, noxious stimuli, or large peaks in spontaneous local neuronal activity. These conditions feature synchronous, high-frequency neuronal activation (e.g., 40 Hz) which is known to be favorable for peptide release in the peripheral and central nervous systems[152–156]. Given that we also observed attenuation of negative CBV responses in CPu by opioid antagonists and acute vasoconstriction of CPu microvessels to the Enk analog, DADLE, it is plausible that the negative CPu hemodynamic responses observed here were produced under conditions featuring activity-dependent opioid transmission. Further, these activity-dependent negative CBV responses may be unique to the CPu and not a characteristic of GABAergic brain regions in general, as we observed positive fMRI responses in the GPe and SNr under the same setting. Importantly, our findings do not preclude the involvement of neurometabolic feedback mechanisms widely assumed to contribute to fMRI responses, but rather highlight additional neurochemical feed-forward mechanisms that can play an outsized role in determining fMRI response polarity; as such, other stimulation or behavioral conditions may not produce negative hemodynamic responses, especially those that are less likely to engage CPu opioid transmission. Indeed, our findings demonstrate that recruitment of different CPu input activity (i.e., dopaminergic inputs) may be sufficient for positive fMRI responses in CPu.

The propensity for negative, activity-induced hemodynamic responses in CPu could be related to the particular vulnerability of CPu neurons to ischemia and hypoxia[157]. Negative BOLD fMRI has been observed in situations where metabolic demand outpaces local changes in cerebral blood flow, a mismatch that is indicative of hypoxia[158]. However, our hemodynamic signal measurements in rodents, including CBV-fMRI, total hemoglobin absorption-derived CBV, and tissue-oxygen, are insensitive to this mechanism for negative BOLD contrast, and previous reports show no prominent increase in cerebral metabolic rate of oxygen and glucose consumption with negative CPu hemodynamic responses to pain[159]. Compared to naturalistic stimuli that drive transient sensory-motor activity, optogenetic manipulations elicit more synchronized activity in the striatum, whereas noxious stimuli recruit opioidergic signaling. These conditions may be necessary to induce decreases in striatal CBV. Evaluating the biological function of striatal activity-induced vasoconstriction, including possible nonmetabolic and neuroprotective roles[160], will be an important goal for future studies using multimodal recording tools[50,51,93–96,161–167]. Further, the strengths of our results lay in validating the observed negative hemodynamic phenomenon using multiple complementary techniques in awake and anesthetized conditions and in multiple species. Nonetheless, it's important to acknowledge the presence of certain limitations in our study. Although we did include female subjects in our experiments, it's essential to clarify that our study was not specifically tailored or adequately powered to delve into potential sex-related differences. It is worth noting that variations in pain perception and the efficacy of opioidergic treatment could indeed exist among different sexes, as indicated by previous research[168,169].

In summary, our results demonstrate that hemodynamic response polarity in the CPu cannot be solely determined by neuronal activity, but rather requires additional, circuit-specific, neurochemical context. Within the CPu, neuromodulators such as DA and opioids may play important contributing roles for the presence and polarity of hemodynamic responses. This information may be critical for interpreting fMRI data, especially in disorders where DA and/or opioid signaling are

dysregulated or altered as treatment. These disorders include, but are not limited to: Parkinson's disease[170,171], schizophrenia[172], attention deficit hyperactivity disorder[173], substance use disorder[174], depression[175,176], and chronic pain[177,178]. As an increasing number of fMRI studies reveal atypical or unstable hemodynamic responses, it is becoming evident that a more comprehensive understanding of factors beyond mere neuronal activity, such as subsequent neurochemical signaling, is essential for advancing fMRI as a tool to study brain function and dysfunction.

## Methods

Our investigations utilized established procedures and neuroscience tools including: optogenetic CBV-fMRI, electrophysiology, intracranial drug delivery, spectral-fiber-photometry, and FSCV, in wildtype male Sprague-Dawley and transgenic male tyrosine hydroxylase (TH)-Cre Long-Evans rats. Where applicable, our CBV-fMRI, electrophysiology, fiber-photometry, and FSCV analyses focused on individual response trials from each subject in order to characterize, with the most detail possible, the shape, sign, and reliability of responses from several stimulation paradigms over each of these modalities. Nonetheless we also present subject-level, rather than trial-level, analyses in order highlight the inter-animal variability within each experiment (Fig. S17). In addition, we performed bright-field vascular imaging on brain slices from C57BL/6J mice. We also conducted transcutaneous electrical nerve stimulation (TENS) during human fMRI and performed original analyses on data from human fMRI with concurrent TMS, collected as part of a previous project, and publicly available at the NIMH Data Archives (NIMH Data Archive Collection ID: 2856). Stock AAV vectors were purchased directly from the University of North Carolina at Chapel Hill (UNC) Vector Core, University of Pennsylvania Vector Core, or Addgene when available, other vectors were obtained as plasmids from Addgene and prepared by UNC Vector Core.

### Rodent subjects

Male Sprague-Dawley ($n = 139$, 350–600 g, postnatal day 80–300) and TH-Cre transgenic Long-Evans ($n = 28$; 400–600 g, postnatal day 120–300) rats were acquired, respectively, from the Charles River Laboratories (Wilmington, MA, USA) and the Rat Resource and Research Center (P40OD011062; Columbia, MO, USA) and pair-housed prior to surgical preparation for fiber-photometry or optogenetic stimulation during fMRI, electrophysiology, or FSCV. Additionally, Wistar rats ($n = 4$, ~350 g, three males and one female, postnatal day 80–200), born and reared at the University of North Carolina-Chapel Hill, were used for an orbitofrontal cortex (OFC) optogenetic fMRI study shown in the supplementary information. Rats were chosen as the primary subjects for the majority of our experiments due to their larger brain size, which allows for higher respective imaging resolution compared to mice[87,88,179–181]. Additionally, rats are frequently employed in pharmacological studies because many drug metabolic pathways have been well-characterized and/or are similar to humans[182,183], and they are well-suited for behavioral neuroscience research thanks to their superior ability to learn complex tasks compared to mice[184]. Animals were given food and water ad libitum and kept on a non-reversed, 12 h light/dark cycle. Animal care and handling followed the National Institutes of Health Guide for the Care and Use of Laboratory Animals (Department of Health and Human Services, NIH publication No. 86–23, revised 1985), and all animal protocols were approved by the Institutional Animal Care and Use Committee at UNC. In addition, brain tissue from young (postnatal day 21–28) C57BL/6J mice of both sexes ($n = 3$, 1 male), bred in-house, was acquired for use on site for bright-field vascular imaging experiments at the University of Sussex, and animal procedures were carried out in accordance with the guidelines of the UK Animals (Scientific Procedures) Act 1986, the Danish National Ethics Committee and European Directive 2010/63/EU. Mice were group-housed in a vivarium maintained at $21 \pm 2\,°C$ and 40–70% humidity, with a non-reversed, 12 h light/dark cycle, and given food and water ad libitum.

### Rat surgical procedures

Rats were deeply anesthetized with 4% isoflurane, which was lowered to and maintained at 2% during surgery via nose cone. Depth of anesthesia was monitored with toe pinch. A circulating warm water pad was used to maintain body temperatures at $37.0 \pm 0.5\,°C$. Animals were secured in a stereotaxic frame for viral injection surgeries, optical fiber implant surgeries, voltammetry recordings, and electrophysiological recordings (Kopf Instruments). All target locations in the brain were located stereotaxically at coordinates relative to skull at bregma according to the Paxinos and Watson Rat Brain Atlas, 6th edition[185].

### Optogenetic virus injections

Burr holes were drilled above the target neuron populations at the following viral injection coordinates (mm): M1 (4 injections): +3.0 AP, ±2.5 ML, −2.2 DV, +3.0 AP, ±1.5 ML, −2.2 DV, +2.0 AP, ±3.0 ML, −2.2 DV, and +2.0 AP, ±2.0 ML, −2.2 DV; PfT (1 injection): −4.2 AP, ±1.3 ML, −2.2 DV; SNc (two injections): −5.3 AP, ±2.7 ML, −7.3 DV and −6.0 AP, ±1.7 ML, −7.9 DV; CPu (two injections, for local or SNr stimulation): +1.5 AP, ±2.8 ML, −5.1 DV and 0.0 AP, ±3.0 ML, −5.2 DV; GPe (one injection): −1.0 AP, ±2.9 ML, −6.2 DV; Anterior Insular Cortex (AI, one injection): +2.75 AP, ±3.75 ML, −6.0 DV; Orbitofrontal Cortex (OFC, one injection per hemisphere): +3.8 AP, ±2.0 ML, −4.2 DV. The viral vectors used for optogenetic stimulation of each target were AAV5-CaMKIIa-hChR2-eYFP and AAV5-CaMKIIa-eYFP for M1, PfT, CPu, and OFC, and AAV5-hSyn-hChR2-eYFP and AAV5-hSyn-eYFP for GPe, and AAV5-hSyn-DIO-hChR2-eYFP and AA5-hSyn-DIO-eYFP for SNc. In addition, AAV5-hSyn-Chronos-GFP and AAV5-hSyn-eYFP were used for AI. Lastly, AAV9-CaMKIIa-GCaMP6F was used for CPu fiber-photometry. Injections were of 1 μL (-$10^{12}$ viral genomes/mL) viral vector, except for PfT and OFC optogenetics and CPu fiber-photometry which were 0.5 μL instead. For each injection, a 30-gauge injection needle was lowered to the target coordinate, raised 0.1 mm, pushed to micro-inject the total volume of viral vector over 10 min, kept in place for an additional 10 min to allow for diffusion, and then finally removed.

### Intracranial implantations

After at least 6 weeks of virus expression, animals were re-anesthetized for implantation surgeries. Burr holes were drilled for each optical fiber (optogenetic stimulation or fiber-photometry), microelectrode (FSCV or electrophysiology), and/or MRI-compatible guide cannula (intracranial pharmacology) needed for the experiment. Optical fibers were lowered to a depth of no more than 0.3 mm above the intended stimulation locations: M1: +3.0 AP, ±2.5 ML, −2.0 DV; PfT: −4.2 AP, ±1.3 ML, −5.8 DV; SNr: −5.5 AP, ±2.2 ML, −7.6 DV; SNc: −5.2 AP, ±1.8 ML, −7.9 DV; GPe: −1.0 AP, ±2.9 ML, −5.9 DV; CPu: +0.0 AP, ±3.0 ML, −4.9 DV; AI +2.75 AP, ±3.75 ML, −5.7 DV; OFC +3.8 AP, ±2.0 ML, −3.7 DV. Guide cannula were implanted to CPu at +0.0 AP, ±3.0 ML, −4.2 DV, 1 mm above the intended infusion area to allow infusion cannula to extend 1 mm into fresh tissue. Optical fibers, guide cannula, and/or reference wires in cerebellum for electrodes were secured to the skull via a dental cement headcap, and 3–4 brass screws (Item #94070A031, McMaster Carr, Atlanta, GA) were affixed at a 45° angle near the ridgelines of the skull to provide structural support for the headcap. Animals were allowed to recover for 1–2 weeks before being scanned and/or recorded. For acute recordings, electrodes were lowered into CPu at 0.0 AP, ±3.0 ML, with a target DV of −5.2 for voltammetry electrodes, and a range of DVs within the anatomical boundaries of CPu for electrophysiology electrodes within each recording session, described below and shown in Fig. S7. Surgical implantation accuracy (for virus injections and optical fibers) was verified with histological analysis and anatomical MRI scans (vide infra), respectively[186,187].

## Optogenetic stimulation

Optogenetic stimulation paradigms were kept consistent between different modality experiments. For each, blue laser light from a 473 nm diode laser (Shanghai Laser and Optics Century, BL473-200FC) was delivered to the targeted region(s) by way of fiber optic patch cables (200 um core, .22 NA, 7 m length; Thorlabs, Newton, NJ) connected to the previously implanted optical fibers (200 μm core, .22 NA; Thorlabs, Newton, NJ). The following target-specific stimulation parameters were used for all experiments as described in the following order: (1) Stimulation frequency, (2) 473 nm laser power, (3) pulse width, (4) stimulation-on block duration, and (5) stimulation-off block duration. M1: 40 Hz, 20 mW, 10 ms, 10 s, 30 s. PfT: 40 Hz, 10 mW, 5 ms, 10 s, 30 s. CPu: 40 Hz, 20 mW, 10 ms, 10 s, 30 s. SNr: 40 Hz, 20 mW, 10 ms, 60 s, 120 s. GPe: 40 Hz, 20 mW, 10 ms, 10 s, 30 s. SNc: 40 Hz, 10 mW, 5 ms, 10 s, 30 s. AI: 20 Hz, 10 mW, 5 ms, 20 s, 80 s. OFC: 10 and 20 Hz, 10 mW, 5 ms, 15 s, 60 s. Because these circuits of interest have not yet been comprehensively studied with fMRI, we used long stimulation durations and high power to evoke the most robust, reliable, and reproducible fMRI signal changes possible. All experiments were accompanied with rigorous eYFP controls to consider potential non-specific effects and heating confounds (Figs. 1, S2, S4, S5, S16). Laser power was manually adjusted using a photodiode power sensor (S120VC, Thorlabs, Newton, NJ) and digital optical power and energy meter (PM100D, Thorlabs, Newton, NJ). All other stimulation parameters were controlled, and synchronized with fMRI when applicable, by a data acquisition board and homemade software interface, as described in the previous section.

## Electrical forepaw stimulation

Bilateral electrical forepaw stimulation was delivered to the left and right forepaw of anesthetized rats undergoing fMRI. To prepare each paw for stimulation, one 27 G electrode needle was inserted between the first and second digits, and another between the third and fourth digits, then both were fixed in place by surgical tape. Electrode patency and correct positioning was confirmed by low power (1 mA) stimulation-induced observable "digit twitching". Electrical current was delivered to each set of electrodes by a constant current stimulus isolator (A385RC, World Precision Instruments, Sarasota, FL), triggered in synchronization with fMRI volume acquisitions by a high-speed analog/ digital input/output data acquisition board (1208Hs-2AO, Measurement Computing Corp., Nortan, MA) controlled by a homemade software program. Noxious electrical forepaw stimulation parameters used during fMRI data collection were based on previous studies[76,144,145,159], and set to 10 mA power, 9 Hz frequency, 10 ms square wave pulse duration, 60 s stimulation-on block duration, and 120 s stimulation-off block duration.

## Rat fMRI

**Anesthesia and animal physiology.** Rats were anesthetized with 2% isoflurane (4% for induction) in medical air approximately 30-min before scanning, during which they were prepared for mechanical ventilation (CWE Inc., MRI-1, Ardmore, PA) by orotracheal intubation, and for intravenous sedative, paralytic, and contrast agent administration with a tail-vein catheter. Subsequently, rats were secured in a custom MRI cradle for scanning, and were switched to a well-established light sedation protocol combining low-dose (0.5%) isoflurane with a cocktail of intravenous dexmedetomidine (0.05 mg/kg/hr) and pancuronium bromide (0.5 mg/kg/hr)[62,86]. Rats were allowed an additional 30-min to establish stable physiological parameters before fMRI scanning. Throughout scanning, ventilation rates and volumes were optimized to maintain $3.0 \pm 0.3\%$ end-tidal $CO_2$ levels (EtCO₂, measured by a capnometer (Surgivet v9004; Smith Medical, Waukesha, WI), heart rate and oxygen saturation (SpO₂) were continuously monitored by a non-invasive, MR-compatible MouseOx Plus System (STARR Life Science Corp.,

Oakmont PA), and maintained to be 250–320 bpm and >87%, respectively, and core temperature was measured by an MR-compatible rectal probe and maintained at $37.5 \pm 1 °C$ by a circulating warm water pad (Thermo Scientific, Waltham, MA). EtCO₂ values from the capnometry system were previously calibrated against invasive sampling of arterial blood gas, reflecting a partial pressure of carbon dioxide level of 30–40 mm Hg[188,189].

**Image acquisition.** fMRI studies were performed on a Bruker 9.4-Tesla/30-cm scanner with a BGA9-S gradient insert in the Center for Animal Magnetic Resonance Imaging at the UNC. A homemade surface coil (1.6 cm inner diameter) served as an RF transceiver[76,159,186]. After positioning rats in the MRI bore, deviations in magnetic field homogeneity were corrected by global shim followed by first- and second-order shims using the standard FASTMAP protocol. MRI images were acquired in anisotropic resolution with 12, 1 mm thick, coronal slices spaced at 1 mm intervals and aligned so that the fifth slice from the front of each animal's head was aligned to the anterior commissure in the mid-sagittal plane (0.36 mm AP relative to bregma); M1 optogenetic stimulation scans however were 8 slices with the 4th from the front corresponding to the AC. Anatomical images were acquired before functional scans, using a T2-weighted RARE sequence with the following parameters: spectral width = 47 kHz, TR/TE = 2500/33 ms, FOV = 2.56 × 2.56 mm², matrix size = 256 × 256, RARE factor = 8, averages = 8.

Prior to all fMRI experiments, an initial 300 s scan was performed during which the contrast agent, Feraheme (ferumoxytol, 30 mg/kg intravenous), was administered for measuring CBV changes[62,96,190–193]. A minimum of 60 s was allowed to elapse before and after Feraheme infusion in order to acquire pre- and post-Feraheme image intensities, respectively. All fMRI scans were acquired with a single shot, gradient echo EPI sequence designed for the Feraheme CBV contrast agent with the following parameters: spectral width = 300 kHz, TR/TE = 1000/8 ms, FOV = 2.56 × 2.56 cm², matrix size = 80 × 80.

**Data preprocessing.** All MR images were preprocessed using the Analysis of Functional NeuroImages software suite (AFNI)[194] and in-house written Python (version 2.7) scripts based on the standard afni_proc.py process stream. Briefly, for each subject scanning session, mean images were generated for the first and last 20 s of the Feraheme infusion scan, corresponding to pre- and post-CBV contrast, respectively. Next, the pre-CBV images and anatomical T2-weighted images were skull-stripped with a hand-drawn brain mask, then the pre-CBV images linearly coregistered to the anatomical in order to derive an affine transformation matrix. The subsequent fMRI scans in each scan session were slice-timing corrected, motion corrected, and then skull-stripped and coregistered to the anatomical scan using the brain mask and transformation matrix applied to the pre-CBV images. Next, the anatomical images were linearly coregistered to the Tohoku rat brain template[195] space and the same transformation matrix was used to convert all fMRI scans to template space. Finally, a 0.5 mm full-width-half-maximum (FWHM) Gaussian kernel was applied to the fMRI scans for spatial smoothing.

**Optogenetic fMRI analysis.** AFNI and Python (version 2.7) were used for analysis of preprocessed fMRI data. Subject-level optogenetic stimulation related fMRI activity was estimated with a general linear model (GLM) between voxel-level timeseries and a convolution of the AFNI "BLOCK" function with stimulation-on and stimulation-off block timestamps, with linear mixed-effects modeling with restricted maximum likelihood (REML) estimates applied to correct for temporal autocorrelations. REML coefficients for fit were used to generate group-level z-score response maps via 1-sample t-tests, which were thresholded to p = 0.001, then corrected for family-wise error to adjust for multiple comparisons of fMRI maps using bi-sided cluster-size

thresholds ($\alpha < 0.01$) determined for each comparison by the 3dClustSim AFNI program. For display purposes only, the polarity of the group-level, fMRI, stimulation response maps were inverted so that CBV-weighted signal changes would reflect activation-related signal changes[63,190].

Qualitative comparisons between ChR2 and eYFP group optogenetic stimulation responses were conducted on subject-level timeseries data extracted from regions of interest (ROIs) corresponding to the stimulation site and CPu (or GPe if Cpu was stimulated locally). For each stimulation site, ROIs were set as the intersection of the group-level stimulation-response map and hand-drawn anatomical areas of the target regions according to the Paxinos and Watson 6th edition rat brain atlas[185] coregistered to template space. Timeseries were converted to units of percent CBV change[63] using the following formulas: Baseline $\Delta R_2^* = -1/TE \ln(S_{bl}/S_{pre\text{-}CBV})$, where $S_{bl}$ and $S_{pre\text{-}CBV}$ represent the mean signal intensity of the first 20 timepoints (before stimulation) and the pre-CBV timepoints from the Feraheme injection scan (described above), respectively. Stimulus-evoked $\Delta R_2^* = -1/TE \ln(S_{stim}/S_{bl})$, where $S_{stim}$ is the signal intensity at each timepoint of the stimulation scan. CBV change timeseries were calculated by dividing the Stimulus-evoked $\Delta R_2^*$ timeseries by the Baseline $\Delta R_2^*$ value. Next, the subject-level timeseries were detrended by simple linear regression and realigned at baseline by subtracting the group-level average from all timepoints before onset of the first stimulation. The group-level mean and standard error of the mean (SEM) timeseries relative to each 2-stimulation epoch were plotted with the Seaborn Python library.

Peak stimulation responses were extracted from subject-level timeseries for quantitative comparisons between ChR2 and eYFP groups, and between ROIs within each group. For each stimulation paradigm, a single peak timepoint was set for all stimulation responses within a given ROI for both groups. The peak timepoint was determined by the absolute maximum mean value of timeseries from the ChR2 group for each individual stimulation, aligned to stimulation onset. Peak stimulation responses were then taken as the average of 5 timepoints centered on the peak timepoint. Finally, using GraphPad Prism 9 (GraphPad Software, San Diego, California), peak stimulation response differences between groups were assessed with Welch's t-tests and plotted as individual data points overlaid on box plots (Tukey's), and the correlations of responses between regions within each group were determined by linear regression.

## Electrophysiology

**Anesthesia.** Rats were anesthetized with 2% isoflurane (4% for induction) during acute electrode implantation as described above. Approximately 30-min before the onset of experiments, rats were switched to a light sedation protocol, similar to that used for fMRI scanning, of 0.5% isoflurane and dexmedetomidine (0.05 mg/kg) administered subcutaneously as a single bolus each hour. Body temperature was maintained at $37.5 \pm 1\,°C$ by a circulating warm water pad (Thermo Scientific, Waltham, MA).

**Acquisition.** Electrophysiological recordings were performed in a homemade Faraday cage with a 16-channel Blackrock Cerebus System (Blackrock Microsystems, Salt Lake City, UT) and NeuroNexus A1x16-10-100-703-OA16LP electrodes (NeuroNexus Ann Arbor, MI). The electrodes feature 16 recording channels spread over a vertical distance of approximately 1.2 mm (i.e., −5.5 mm DV implantation depth recorded from channels as dorsal as −4.3 mm DV), and a stainless-steel reference wire which was implanted in cerebellum. Broadband data were collected at a 30 kHz sampling rate with adaptive line-noise cancellation, and separated into local field potential (LFP) and multi-unit spiking activity (MUA) by bandpass filtering between 10–250 Hz and 250 Hz to 5 kHz, respectively. Digital timestamps were also collected corresponding to each optogenetic stimulation-on pulse during the recordings. CPu recordings were taken across multiple DV coordinates between −5.2 and −7.4 mm relative to skull at bregma (zeroed at the electrode tip), as detailed in Fig. S5. Multiple 2-stimulation-block epochs were given during each recording, separated and preceded by at least 120 s of quiescent baseline.

**Preprocessing.** MUA and LFP electrophysiology data were preprocessed with Python scripts within the Neuroexplorer 5 data analysis software package (Nex Technologies, Colorado Springs, CO). Briefly, for each channel from each recording, the MUA data was aligned to the onset of individual stimulation blocks, then spike counts over 100 ms bins (spikes/100 ms) were totaled and extracted for group-level analyses. Similarly, MUA data was also aligned to individual stimulation pulse timestamps, and spike probability (spikes/bin)/pulses over 100 μs bins was extracted for group-level analyses of single-pulse responses. Because stimulation effects can carry over into the relatively short inter-pulse-intervals, baseline comparisons for single-pulse analyses were performed against the same data aligned instead to individual stimulation pulse timestamps that had been shifted earlier by the stimulation-on block duration (e.g., 10 s or 60 s), thereby corresponding to stimulation-off blocks. The LFP data for each channel from each recording was aligned to the onset of individual stimulation blocks, then the average spectrogram and fast Fourier transform (FFT) was extracted for each to be used in group-level analyses. Spectrograms were generated with the following parameters: frequency range = 0–100 Hz, frequency values = 8192 (frequency bins = 52), windowing function = Welch, windowing shift (absolute) = 100 ms, time bandwidth product = 2, tapers = 2, units = decibels (dB).

**Analysis.** Group-level electrophysiology analyses were performed in Python (version 2.7). Recording-channel-level preprocessed MUA spike counts aligned to stimulation blocks (described above) were Z-normalized to the baseline period corresponding to all timepoints before the first stimulation, then averaged within ChR2 and eYFP groups to create histograms of the stimulation response (with SEM) for each stimulation target with the Seaborn Python library. Next, to quantify the group-level MUA stimulation response for each stimulation target, the mean spike count from across the stimulation block timepoints was plotted for each recording channel as an individual data point and overlaid on a box plot (Tukey's) with GraphPad Prism 9 (GraphPad Software, San Diego, California), and the stimulation response difference between ChR2 and eYFP groups was assessed with a Welch's t-test. For group-level analyses of the recording-channel-level preprocessed MUA spike probabilities for single-pulses of stimulation, the mean of the corresponding baseline-shifted spike probabilities was subtracted from each, such that the data reflected spike probability change. The single-pulse data was then averaged within ChR2 and eYFP groups to create single-pulse spike probability change histograms and SEM for each stimulation target and using the Seaborn Python library.

Recording-channel-level preprocessed LFP spectrogram data aligned to stimulation blocks (described above) were averaged within ChR2 and eYFP groups to create group-level average spectrograms for each stimulation target using the Matplotlib library in Python. To quantify the group-level LFP stimulation response for each stimulation target, the recording-channel-level spectrograms were averaged across timepoints from the stimulation block and the preceding baseline block, creating stimulation and baseline condition power spectral densities (PSDs), respectively. Because FFT is prone to error when the sampling window includes data that is not stationary or periodic, such as the onset and offset of stimulation, the first and last 6 timepoints of the baseline and stimulation blocks, corresponding to FFT window length (546 ms), were excluded from averaging. Next, recording-channel-level baseline PSDs were subtracted from the corresponding stimulation PSDs, representing the dB power change as a result of stimulation. Power change PSDs were then averaged within

ChR2 and eYFP groups to create group-level average power change PSDs with SEM for each stimulation target using the Seaborn Python library. In addition, for each recording-channel-level PSD, averages of the power changes from the 3 frequency bins (~5.5 Hz range) around the stimulation frequency (40 Hz) were extracted, and for each stimulation target these were plotted as individual data points overlaid by a box plot (Tukey's), and the response differences between ChR2 and eYFP groups were evaluated by Welch's $t$ tests in GraphPad Prism 9 (GraphPad Software, San Diego, California).

## Fast-scan cyclic voltammetry

**Electrode fabrication.** Borosilicate glass capillary or polyimide/fused silica carbon-fiber microelectrodes were fabricated according to existing methods[118,196,197]. Briefly, a 7 μm-diameter carbon-fiber (Thornel T-650) was threaded through a glass or fused silica/polyimide capillary (#1068,150,381; Polymicro Technologies Inc., Phoenix, AZ, USA) and sealed with heat or clear epoxy, respectively. Approximately 100 μm of carbon-fiber protruded from one end of the capillary to serve as the active surface area. Glass capillary microelectrodes were backfilled with a salt solution to serve as a conductive medium for a stainless-steel connection wire, and fused silica capillary microelectrodes were backfilled with a salt solution. Carbon fibers were adhered to a silver connection wire via consecutive layers of silver epoxy, silver paint, and clear epoxy.

**Acquisition.** FSCV experiments were performed within a homemade Faraday cage using the same sedation and anesthetic protocol used for fMRI scans, but without a paralytic or artificial ventilation. Data were acquired and analyzed using High-Definition Cyclic Voltammetry and Analysis software, respectively (UNC Electronics Facility, University of North Carolina at Chapel Hill, Chapel Hill, NC). Custom instrumentation from the UNC Electronics Facility was used. Microelectrodes were acutely implanted to between −3.8 and −5.0 DV at CPu coordinates, and a freshly chloridized Ag/AgCl reference wire was implanted near the contralateral cerebellum. Microelectrode recording depth and location was kept constant for each stimulation site and was not optimized to obtain maximal DA release. An oxygen-sensitive voltage waveform[83] was scanned at the electrode surface at 400 V/s and a rate of 10 Hz. The waveform first scans from 0 V to +0.8 V to oxidize DA, then down to −1.4 V to reduce molecular tissue-oxygen and DA orthoquinone, and finally returns to 0 V. Optogenetic stimulations were given according to the appropriate group-specific paradigm described above. Each stimulation was repeated at least three times.

**Analysis.** FSCV data were filtered using a digital 4th order Bessel low-pass filter (2 kHz cutoff frequency). Current time-courses taken at the reduction or oxidation currents of the analytes of interest were analyzed with principal component analysis (PCA) and converted to concentration changes using fiber length-normalized in vitro calibration factors acquired previously (4.8 nA/uM and −0.19 nA/uM for DA and oxygen, respectively, at fused silica capillary fiber microelectrodes and 2.5 nA/uM and −0.35 nA/uM for DA and oxygen, respectively, at glass capillary microelectrodes). If DA was not obviously present (e.g., in eYFP controls), then DA-specific PCA training sets could not be obtained[198]. In lieu of using a DA training set, the reported DA concentrations are representations of the raw current of the DA oxidation potential after oxygen and pH PCA components at the same potential are subtracted (i.e., "DA" is estimated using residual current). Time-courses were background subtracted immediately prior to stimulation epochs for consistent baselines between files. Concentration time-courses were plotted in GraphPad Prism 9 (GraphPad Software, San Diego, California) and averaged into groups according to recording hemisphere.

## Spectral fiber-photometry

**Apparatus.** Fiber-photometry recordings were collected from awake, freely moving rats in a commercially-available behavior chamber (Med Associates, Fairfax, VT). Footshocks were administered with a shocker-scrambler (ENV-414S, Med Associates, Fairfax, VT) via the metal grid floor in the behavioral chamber. Shock duration was set to 1 s, and amplitude to 0.7 mA. The spectral fiber-photometry recording system used herein is similar to that described previously in refs. 51,93,95. Briefly, laser light from a 488 nm continuous wave (CW) laser (OBIS 488 LS-60, Coherent, Santa Clara CA) and 405 nm CW laser (OBIS 405 LS-50, Coherent, Santa Clara CA) is aligned and combined by broadband dialectric mirrors (BB1-E02, Thorlabs, Newton, NJ) and a long-pass dichroic mirror (ZT488rdc, Chroma Technology Corp, Bellows Falls, VT), then sent into to a fluorescence cube (DFM1, Thorlabs, Newton, NJ). The fluorescence cube reflects and send the combined light into a multi-mode optical fiber patch cable (105 μm core; Thorlabs, Newton, NJ), which is connected to the optical fiber implanted to in the CPu of the animal being recorded. Both excitation light delivered by the lasers, and emission light from GCaMP6f is passed through the implanted optical fiber and the patch cable. The emission fluorescence is redirected at the fluorescence cube, and passed through an emission filter (ZET488/561, Chroma Technology Corp, Bellows Falls, VT), before traveling through another multi-mode patch cable (200 um core; M200L02S-A, Thorlabs, Newton, NJ) into a spectrometer (QE Pro-FL, Ocean Optics, Largo, FL). Spectral data acquisition by the spectrometer and TTL-synchronization to other hardware is controlled by the OceanView software package (Ocean Insight, Orlando, FL).

**Acquisition.** To prepare for each recording session, the behavioral chamber and recording room were darkened as much as possible, the output power of the lasers adjusted to balance spectral amplitudes (the maximum power was always less than 100 uW), then a background spectrum was acquired and automatically subtracted during recordings. Spectral fiber-photometry recordings were acquired at 20 Hz, with 405 and 488 nm laser excitation interleaved such that the effective sampling rate for both emission spectra was 10 Hz. Importantly, while the resulting emission from 488 nm GCaMP excitation is calcium dependent and related to neuronal activity, the emission from 405 nm GCaMP excitation is calcium independent[51,199] and can serve as a baseline measurement for non-neuronal signal changes[200,201]. Recordings of spontaneous activity were acquired from each animal in a single 600 s session while habituating to the behavior chamber, and were followed by a single 600 s footshock session. A total of nine footshocks were delivered at pseudorandom intervals between 30 and 90 s.

**Preprocessing.** The emission spectrum for 405 nm GCaMP at each recording timepoint was used to derive total blood hemoglobin concentration (HbT) as a surrogate for local CBV signal and correct the corresponding 488 nm GCaMP emission for blood hemoglobin absorption using an established workflow in MATLAB (R2019b, MathWorks)[51,93]. Briefly, using known molar extinction coefficients for oxyhemoglobin (HbO) and deoxyhemoglobin (HbR), photon-traveling pathlengths from a Monte Carlo simulation along with the 405 nm GCaMP spectral data over the timepoints in each recording, we solved for HbO and HbR molar concentration changes for each timepoint using the generalized method of moments (GMM). These hemoglobin changes were then used to calculate the absorption effect at each timepoint of the corresponding GCaMP signal using the GMM, and this effect was removed, leaving the hemogloblin-corrected GCaMP signal for group-level analyses. In addition, HbO and HbR changes were simply added together to get HbT changes for use in group-level analyses. Corrected GCaMP signals and HbT changes were Z-normalized for further analyses.

**Analysis.** Group-level spectral fiber-photometry data analysis was performed in Python (version 2.7). For the analysis of spontaneous activity, the timestamps corresponding to strong GCaMP peaks were isolated from each recording. To isolate activity peaks from slow-wave background oscillations, subject-level corrected GCaMP signals (described above) were copied, then the copies were detrended by simple linear regression, Z-normalized, and finally filtered with a 3rd order, 10 Hz, butterworth highpass filter. Peaks were detected by thresholding the resulting timeseries at the 97.7% percentile; to avoid overlap, if more than 1 peak was detected within a 30 s window, only the highest amplitude peak was kept. Windows from −15 to 30 s aligned to each peak timestamp were used to extract the corresponding original GCaMP and HbT timeseries data. The mean and SEM across all GCaMP and HbT timeseries data aligned to strong GCaMP peaks were then plotted with the Seaborn library in Python. In addition, to quantify the spontaneous activity, we collected values in each GCaMP-peak-aligned GCaMP and HbT timeseries around the timestamp corresponding to the peak magnitude within the mean GCaMP and HbT timeseries, respectively. The average of the five values centered on these timestamps (500 ms total) for each individual timeseries were plotted as individual data points and overlaid on box plots (Tukey's), then evaluated by 1-sample t-tests in GraphPad Prism 9 (GraphPad Software, San Diego, California).

Footshock-induced GCaMP activity and HbT changes were collected in temporal windows (−10 to 30 s) around each footshock timestamp from each recording. Each footshock-aligned timeseries was then detrended by simple linear regression and z-normalized to the 10 s baseline period preceding the footshock. The mean and SEM across all GCaMP and HbT timeseries data aligned to footshocks were then plotted with the Seaborn library in Python. To quantify the footshock induced activity, we collected the values in each footshock-aligned GCaMP and HbT timeseries around the timestamps corresponding to the peak magnitudes within the mean GCaMP and HbT timeseries, respectively. Because we observed 2 discrete peaks in the GCaMP and HbT timeseries, we collected the peak timestamp for the largest magnitude change and a second peak timestamp for the largest magnitude change after the interpolated right intersection point of the first peak. The average of the 5 values centered on these timestamps (500 ms total) for each individual timeseries were plotted as individual data points and overlaid on box plots (Tukey's), then evaluated by 1-sample t-tests in GraphPad Prism 9 (GraphPad Software, San Diego, California).

### Intracranial pharmacology

**Drug preparation.** Drug solutions were freshly mixed via sonication prior to scanning using the following concentrations in 0.9% saline vehicle: Lido (lidocaine hydrochloride monohydrate, L5647, Sigma-Aldrich, St. Louis, MO), 69 mM; Ach (acetylecholine chloride; A6625, Sigma-Aldrich, St. Louis, MO), 6.6 mM; BIBP (BIBP-3226 tifluoroacetate; 27-071-0, Tocris Bioscience, Bristol UK), 9.5 mM; SCH+Rac (R(+)-SCH-23390 hydrochloride, D054, Sigma-Aldrich, St. Louis, MO; S(-)-raclopride l-tartrate, R121, Sigma-Aldrich, St. Louis, MO), 2 mM and 1.7 mM, respectively; Nal (naloxone hydrochloride, 0599, Tocris Bioscience, Bristol UK), 6 mM; Oxo (oxotremorine m, O100, Sigma-Aldrich, St. Louis, MO), 10 μM; CST (cyclo(7-aminoheptanoyl-phe-d-trp-lys-thr[bzl]), C4801, Sigma-Aldrich, St. Louis, MO), 64 mM; Caf (caffeine, 2793, Tocris Bioscience, Bristol UK), 100 mM; nor-BNI (nor-binaltorphimine dihydrochloride, 0347, Tocris Bioscience, Bristol UK), 10 mM. A separate batch of each drug was freshly prepared to pH test the solutions in triplicate, and these values were later found to have no correlation with the corresponding drug effects on stimulus-evoked fMRI responses.

**Optogenetic stimulation and drug delivery.** An MRI-compatible plastic infusion cannula (C315I/PK/SPC, Plastics One, Roanoke, VA) loaded with drug, was inserted into the CPu via the previously implanted guide cannula (C315G/PK/SPC, Plastics One, Roanoke, VA) and secured in place with a custom-bored dummy cannula cap (C315DCN/SPC, Plastics One, Roanoke, VA). Drug infusions were controlled and synchronized with fMRI volume acquisition by a programmable syringe pump (Fusion 720, Chemyx Inc., Stafford, TX). Infusion pressure was applied to the infusion cannula via polyethelene 50 tubing connected to a 10 μL syringe (1701RN, Hamilton, Reno, NV), backfilled with 0.9% saline – a small air bubble was created at the drug-saline interface to avoid drug dilution and marked to confirm drug delivery after each scan. Optogenetic stimulation parameters for SNr (vide supra) were used, and 15 sequential stimulation epochs were presented for each scan. Stimulation-on and -off blocks were performed before, during, and after unilateral drug infusion for statistical comparisons and verifying drug delivery. After 5 "pre-drug" stimulation epochs, drug was intracranially delivered to CPu via the cannula as a 0.5 μL aliquot over a 10 min period.

**Analysis.** Individual subject responses in CPu were obtained from the pre-drug stimulation blocks (AFNI 3dREMLfit, $p < 0.001$, minimum cluster-size = 40 voxels) and used as masks to extract individual CPu response timeseries for the entire scan sessions. To accurately compare pre-drug to post-drug stimulation response amplitudes, vehicle-induced baseline shift was subtracted from timeseries via asymmetric least squares fitting[202] and (90%) quantile regression[203]. Drug effects were quantified as percent changes in CPu stimulation-evoked, peak-response amplitudes in post-drug stimulation blocks compared to pre-drug stimulation blocks. Response peaks were timestamped at the location corresponding to the largest signal change following stimulation onset across pre- and post-drug periods, and the average of the five values centered on these timestamps (500 ms total) for each individual timeseries were used to calculate percentage change in response amplitude as (pre-drug - post-drug)/pre-drug values. Using GraphPad Prism 9 (GraphPad Software, San Diego, California), percentage changes were plotted at subject-level data points and overlaid on box plots (Tukey's), then evaluated by Welch's ANOVA with planned comparisons versus the saline vehicle group, FDR corrected with the two-stage linear step-up procedure.

### Rat histology

At the conclusion of the aforementioned experiments, rats were euthanized with sodium pentobarbital (120 mg/kg i.p.) after confirming depth of anesthesia. Subjects were transcardially perfused with saline, followed by 4% paraformaldehyde (PFA), then their brains were post-fixed overnight in 4% PFA, rinsed in 1X PBS, and placed in 30% sucrose at 4 °C for 48 h. Subsequently, 40 μm brain-sections were cut with a slicing microtome (HM 450, Thermo Fisher Scientific, Waltham, MA) and mounted and cover slipped onto Superfrost Plus slides (Thermo Fisher Scientific, Waltham, MA) with Fluoro-Gel II with DAPI (#17985-50, EMS, Hatfield, PA) mounting medium. Slides were imaged at 10x magnification with a fluorescence microscope (DM500, Leica Microsystems, Wetzler Germany) for histological verification of fiber placements for those subjects that did not undergo MRI, and confirmation of virus expression as indicated by co-expressed fluorophores.

In addition, brains from naïve rats prepared with control virus, fixed as described above, were sent to the UNC Pathology Services Core (PSC) for sectioning and staining, where applicable, with primary antibodies against DARPP-32 (#611520, BD Biosciences, Haryana, India; 1:200 concentration, 30 min incubation), ChAT (#50-265, ProSci, Poway, CA; 1:200 concentration, 30 min incubation), and PV (ab13970, Abcam, Cambridge UK; 1:2000 concentration, 30 min incubation) were used as selective markers for MSNs, ChAT+ interneurons, and PV+ interneurons, respectively. Triple immunofluorescence (IF) was performed on paraffin-embedded tissues that

were sectioned at 5μm. This IF assay was carried out on the Bond Rx fully automated slide staining system (Leica Microsystems, Wetzler Germany) using the Bond Research Detection kit (DS9455). Slides were deparaffinized in Leica Bond Dewax solution (AR9222), hydrated in Bond Wash solution (AR9590) and sequentially stained. Heat induced antigen retrieval was performed at 100 °C in Bond-Epitope Retrieval solution 1 pH-6.0 (AR9961). The antigen retrieval was followed with a 5 min Bond peroxide blocking step (DS9800). After pretreatment, slides were incubated with the antibody followed with the appropriate host-dependent secondaries (DARPP-32: EnVision+ System- HRP, labeled polymer, anti-mouse; ready-to-use, 30 min incubation; K4001, Dako, Santa Clara, CA. ChAT: ImmPRESS HRP anti-goat IG; ready-to-use, 20 min incubation; MP-7405, VectorLabs, Newark, CA. PV: EnVision+ System- HRP, labeled polymer, anti-rabbit; ready-to-use, 30 min incubation; K4003, Dako, Santa Clara, CA)). Secondaries were detected with tyramide or Alexa fluorophores. Nuclei were stained with Hoechst 33258 (Invitrogen, Waltham, MA). The stained slides were mounted with ProLong Gold antifade reagent (P36930, Life Technologies, Carlsbad, CA). Positive and negative controls (no primary antibody) were included in each run.

Additional brains from naïve rats prepared with control virus, were fixed, cut into 35 μm sections, and mounted in-house as described above, then stained for TH (AB152, MilliporeSigma, Burlington, MA; 1:2000 concentration, 16 h incubation). Goat anti-rabbit secondary antibody conjugated to Alexa568 (A11036, Invitrogen, Waltham, MA; 1:1000 concentration, 2 h incubation) was used indicate TH staining, and slides were coversliped with Fluoro-Gel II with DAPI (17985-50, EMS, Hatfield, PA) mounting medium.

Confocal images of all immunohistochemically prepared brain slices were obtained with an LSM780 microscope (Zeiss, Oberkochen Germany). The quantification method for determining co-localized neurons of either eYFP and TH, or eYFP, DARRP32, and either ChAT or PV is as follows. We used a full-tissue thickness, z-stack of the CPu at 40x magnification. Tissue thickness was selected such that whole individual cells could be clearly analyzed within the viewing depth and the tissue was sampled at least 120 μm apart to avoid counting a single neuron multiple times. The images were viewed and cells were counted with the Image J (Fiji) open-source software package[204] using the "plug-in" Analyzer/Cell Counting. Only brightness and contrast were adjusted during counting.

## Bright-field vascular imaging

Brain slices were collected, imaged, and analyzed using the methods and equipment described in Hall et. al. (2014)[205]. In our preparation, we first added the TXA2 analog U46619 (100 nM; Cayman Chemicals, Ann Arbor, MI) to the perfusion of oxygenated (95% O2, 5% CO2) aCSF to establish a more physiological vascular tension[107–109]. In addition, vessel constriction to TXA2 served as visual confirmation of vessel viability, and the approximate time between TXA2 administration and peak TXA2 constriction was used to offset all other timestamps for each recording to account for differences in flow rate or slice perfusion rate. Next, DADLE (1 μM; E7131, Sigma-Aldrich, St. Louis, MO) was added to the perfusion, followed by washout of the perfusion back to only TXA2 in aCSF. The timing between experimental conditions was at least 10 min. Vessel diameter was manually measured offline, at 4-s temporal resolution, by an experimenter blinded to experimental timings and conditions using the Image J (Fiji) open-source software package[204]. Vessel diameter measurements were averaged over a 5-min window beginning at the start time of each condition (offset relative to TXA2 effect timing, as described above) for statistical analysis. Significant alterations in vessel diameter were determined by repeated-measures one-way ANOVA with Šídák's multiple comparisons test versus the preceding experimental condition in GraphPad Prism 9 (GraphPad Software, San Diego, California).

## Human fMRI

**Transcutaneous electrical nerve stimulation.** Seven healthy right-handed subjects (six males and one female, $32.1 \pm 5.5$ years old, mean ± SD) participated in the study. None of the subjects had any form of acute or chronic pain, or took drugs that affect pain sensations or the central nervous system. An informed consent form was obtained from all subjects prior the study, and the study procedures were approved by the Institutional Review Board at the UNC.

Somatosensory stimuli were evoked by TENS of the bilateral median nerves. Electrodes were placed on the wrist and forearm to stimulate the median nerve. The median nerve is responsible for carrying sensation from the thumb, the complete first and second fingers, and the lateral half of the third finger to the spinal cord. A standard TENS stimulator (HANS-200A, HANS Therapy Co., Ltd. Wuxi, China) was connected to the electrodes. The nerve stimulator outputs a square wave at 5 Hz with a pulse width of 0.2 ms. Because the sensory and pain thresholds vary in different subjects, the stimulation intensities of each subject were determined before the subject entered the scanner. Two different intensities were applied: non-painful stimulation is of minimal amplitude and just above sensory threshold, ranging from 3 to 5 mA ($3.9 \pm 0.7$ mA, mean ± SD); and painful stimulation is moderate but tolerable, rating around 60 on the numerical scale 0–100 (0 means no pain and 100 means the most intense pain imaginable), ranging from 17 to 35 mA ($25.4 \pm 6.8$ mA, mean ± SD). For each stimulation session during the fMRI scan, a block design (20 s baseline, then 20 s ON and 30 s OFF, repeated 6 times, totaling 5 min and 20 s) was used.

MRI data were acquired on a 7 T system (Siemens Magnetom, Erlangen Germany) with a 32-channel head array coil at the UNC, Biomedical Research Imaging Center. During each fMRI scan, 56 axial slices ($2.973 \times 2.973 \times 3$ mm³, in-plane matrix $74 \times 72$) were acquired sequentially (inferior to superior) with a gradient echo EPI sequence (TR/TE = 2000/25 ms, flip angle = 79°). High-resolution T1-weighted sagittal anatomical images (TR/TE = 2200/2.86 ms; in-plane matrix $256 \times 248$; $0.859 \times 0.859 \times 1$ mm³) were also acquired.

**Transcranial magnetic stimulation.** Eighty-two healthy individuals (i.e., no current psychopathology or history of psychopathology) underwent a concurrent TMS-fMRI scanning session and were included for fMRI data analyses. All participants' demographic information is presented in Table 1. The scanning sessions were conducted at Stanford University with a 3T scanner (Signa Excite, GE, Chicago, IL) according to established protocols[206] and approved by the Stanford Institutional Review Board. These data were collected as part of a previous project and are publicly available at the NIMH Data Archives (NIMH Data Archive Collection ID: 2856).

A high-resolution T1-weighted anatomical image collected at a separate scanning session was used with Visor2 neuronavigation software (ANT Neuro, Enschede NL) to define TMS stimulation sites for each participant prior to the TMS-fMRI scanning session. The TMS sites included right aMFG, pMFG, M1. The coordinates of these stimulation

**Table 1 | TMS fMRI participants' demographic information and coordinates**

| TMS sites | Total number | Female | Male | Age (s.d.) | Years of Education (s.d.) | MNI coordinates for stimulation | | |
|---|---|---|---|---|---|---|---|---|
| | | | | | | x | y | z |
| Right aMFG | 80 | 46 | 34 | 31.9 (10.7) | 16.2 (2.1) | 30 | 50 | 26 |
| Right pMFG | 79 | 47 | 32 | 31.7 (10.7) | 16.1 (2.1) | 46 | 26 | 38 |
| Right M1 | 79 | 48 | 31 | 31.7 (10.6) | 16.2 (2.1) | 40 | −18 | 64 |

sites (Table 1) were defined in MNI-152 anatomical atlas space and were then warped into subject native space using the FMRIB Software Library applywarp module[207,208]. The TMS target locations on the scalp of each subject were identified within each subject's native space anatomical image after standardizing head position using nasion and bilateral tragi as fiducial markers in Visor2. These stimulation spots were then marked on a Lycra swim cap (Speedo USA, Cypress, CA) worn by the subject to enter the scanner.

Once sites were marked, motor threshold (MT) was identified for each participant in the scanner room. MT was defined as the lowest possible stimulation intensity at a site along motor cortex that induced a visible "twitch" response in the contralateral abductor pollicis brevis (thumb) muscle during 50% of 10 single-pulse stimulations, a common within-subject metric for individualization of TMS intensity. Single-pulse TMS was then delivered during concurrent MRI scanning at 120% of motor threshold for each participant. For subjects whose motor threshold response was not identified after 20 min of testing, stimulation was conducted at 100% intensity.

Concurrent TMS-fMRI sessions were acquired with a custom-built TMS-compatible head-coil. Thirty-one slices (4.0 mm thick, 1 mm gap) covered the whole brain via a gradient echo spiral in/out pulse sequence (TR = 2400 ms, TE = 30 ms, flip angle = 85°, 1 interleave, FOV = 22 cm, pixel size = 3.4 mm, 64 × 64 matrix). Data sampling was limited to 2100 ms to leave a 300 ms gap between volumes for TMS single-pulse delivery to avoid corruption of BOLD signal. The TMS pulses were delivered via an MR-compatible figure-eight TMS coil (MRI-B91, MagVenture, Alpharetta, GA) held in place by a custom-built MRI coil holder, triggered by a stimulator (X100, MagVenture, Alpharetta, GA) located outside the scanner room and connected to the coil via the penetration panel. The TMS sites were repositioned for each participant by sliding the participant out of the magnet bore, adjusting the TMS coil position, and returning the participant into the bore. At each site, 68 TMS pulses were presented with a variable inter-trial interval jittered with a range from 1–6 TRs over 6 min and 41 s (167 volumes) in a fast event-related design. Heart rate (pulse oximeter clipped to the distal phalanx) and respiration (mid-thoracic strain gauge; both GE Healthcare, Chicago, IL) were acquired at 50 Hz during each TMS-fMRI run.

Heart rate and respiration during each run were processed offline using empirically defined thresholds for R-spike and respiration peak detection along with minimum intervals between peaks and were regressed during data reconstruction in which P-files were converted to NIFTI files. The first three volumes of TMS runs were discarded for further preprocessing analyses.

**Preprocessing and analysis.** The images collected from the TENS and TMS studies were analyzed using AFNI. For anatomical localization, T1-weighted high-resolution images from each subject were transformed to standard MNI template space[209,210] for inter-subject comparisons. All EPI images were slice-timing and motion corrected, then aligned to the corresponding anatomical T1-weighted images, and subsequently normalized to the standard MNI space. Gaussian smoothing (FWHM = 4 mm) was performed on the functional images before conducting GLM fitting with the AFNI 3dDeconvolve function. The GLM model was applied only to fMRI data confined to the striatum by an anatomical mask (Fig. 6A). To test the significance of pain-evoked or TMS responses at the group-level, we used a parametric one-sample $t$ test implemented in AFNI and reported $Z$ values that were converted from $t$ values. The false discovery rate correction was used to adjust for multiple comparisons of the fMRI maps ($p < 0.05$). For display purposes, group response maps were centered on the most significant striatum response voxel.

**Reporting summary**
Further information on research design is available in the Nature Portfolio Reporting Summary linked to this article.

## Data availability
Processed data used for discrete hypothesis testing, bar, box, and scatter plots, and the exact test statistics derived from these data are provided in the Source Data file. All other data associated with this work, including those used for fMRI response maps, time-course plots, and peri-event spectrograms, are publicly available on Zenodo: all rodent data and human TENS fMRI data, https://doi.org/10.5281/zenodo.8417144[211]; human right aMFG TMS fMRI data, https://doi.org/10.5281/zenodo.10484834[212]; human right M1 TMS fMRI data, https://doi.org/10.5281/zenodo.10486110[213]; human right pMFG TMS fMRI data, https://doi.org/10.5281/zenodo.10486587[214]. The human TMS fMRI data, which was collected as part of a previous project, is also publicly available on the NIMH Data Archives (NIMH Data Archive Collection ID: 2856): https://nda.nih.gov/edit_collection.html?id=2856. Source data are provided with this paper.

## Code availability
The code used to perform the analyses in this study are publicly available on Zenodo: https://doi.org/10.5281/zenodo.7683340[215].

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

## Acknowledgements

We thank Nathalie Van Den Berge (currently affiliated with the Aarhus University), Esteban Oyarzabal (currently affiliated with the Charles River Laboratories), Martin Mackinnon (currently affiliated with the National Institute of Neurological Disorders and Stroke), and Manasmita Das (currently affiliated with the Presidency University) for their support and constructive feedback during their tenure at the Center for Animal MRI at the UNC. We appreciate Thomas Kash and Gregory Scherrer at the UNC and Warren Grill at the Duke University for their insightful discussions. We also thank Kira Shaw, Orla Bonnar, and Laura Bell at the University of Sussex for technical assistance with bright-field microscopy, and Yongjuan Xia in the Pathology Services Core at the UNC for technical assistance with tissue sectioning and imaging (supported in part by P30CA016080). The human TMS-fMRI data used in the preparation of this manuscript were obtained from authors A.E. and J.J. These data were previously made available by A.E., J.J. and other Submitters at the National Institute of Mental Health (NIMH) Data Archive (NDA). NDA is a collaborative informatics system created by the National Institutes of Health to provide a national resource to support and accelerate research in mental health. Dataset identifier: NIMH Data Archive Collection ID #2856. This manuscript reflects the views of the authors and may not reflect the opinions or views of the NIH or of the Submitters submitting original data to NDA. This work was supported in part by the National Institute of Mental Health (RF1MH117053, R01MH126518, R01MH111429, S10MH124745 to Y.-Y.I.S., F32MH115439 to L.R.W., DP1MH116406 to A.E.), National Institute of Neurological Disorders and Stroke (R01NS091236 to Y.-Y.I.S., R21NS133913 to W.Z. and Y.-Y.I.S.), National Institute on Alcohol Abuse and Alcoholism (P60AA011605 and U01AA020023 to Y.-Y.I.S. and S.H.L., T32AA007573 to D.H.C.), National Institute of Biomedical Imaging and Bioengineering (R01EB033790 to Y.-Y.I.S.), National Institute of Drug Abuse (R21DA057503 to Y.-Y.I.S.), National Institute of Child Health and Human Development (P50HD103573 to Y.-Y.I.S. and S.H.L.), and National Institute of Health Office of the Director (S10OD026796 to Y.-Y.I.S.).

## Author contributions

Conceptualization, D.H.C., D.L.A., Y.-Y.I.S.; Methodology, D.H.C., D.L.A., L.R.W., B.K., T.-W.W., T.-H.H.C., W.Z., S.-H.L., A.E., C.N.H., G.S., Y.-Y.I.S.; Investigation, D.H.C., D.L.A., L.R.W., B.K., T.-W.W., T.-H.H.C., R.J.N., J.J.; Resources, W.Z., J.J., S.-H.L., A.E., C.N.H., G.S., Y.-Y.I.S.; Data Curation, D.H.C., L.R.W., B.K., T.-W.W., T.-H.H.C., W.Z., J.J.; Writing – Original Draft, D.H.C., D.L.A., L.R.W., Y.-Y.I.S.; Visualization, D.H.C.; Supervision, D.H.C., A.E., C.N.H., Y.-Y.I.S.; Project Administration, D.H.C., A.E., C.N.H., Y.-Y.I.S.; Funding Acquisition, D.H.C., A.E., C.N.H., Y.-Y.I.S.

## Competing interests

The authors declare no competing interests.

## Additional information

[1]Center for Animal MRI, the University of North Carolina at Chapel Hill, Chapel Hill, NC, USA. [2]Biomedical Research Imaging Center, the University of North Carolina at Chapel Hill, Chapel Hill, NC, USA. [3]Department of Neurology, the University of North Carolina at Chapel Hill, Chapel Hill, NC, USA. [4]Department of Biological Sciences, Carnegie Mellon University, Pittsburgh, PA, USA. [5]Department of Psychiatry and Behavioral Sciences, Stanford University, Stanford, CA, USA. [6]Wu Tsai Neurosciences Institute, Stanford University, Stanford, CA, USA. [7]Department of Pediatrics, University of Iowa Carver College of Medicine, Iowa City, IA, USA. [8]Department of Psychiatry, University of Iowa Carver College of Medicine, Iowa City, IA, USA. [9]Iowa Neuroscience Institute, University of Iowa Carver College of Medicine, Iowa City, IA, USA. [10]Alto Neuroscience, Los Altos, CA, USA. [11]Sussex Neuroscience, University of Sussex, Falmer, United Kingdom. [12]School of Psychology, University of Sussex, Falmer, United Kingdom. [13]Center for Neurobiology of Addiction, Pain, and Emotion, University of Washington, Seattle, WA, USA. [14]Department of Anesthesiology and Pain Medicine, University of Washington, Seattle, WA, USA. [15]Department of Pharmacology, University of Washington, Seattle, WA, USA. [16]These authors contributed equally: Domenic H. Cerri, Daniel L. Albaugh, Lindsay R. Walton. ✉e-mail: shihy@unc.edu

