## [Peer Review File · Nature Communications]

Distinct neurochemical influences on fMRI response polarity in the striatumREVIEWER COMMENTS

Reviewer #1 (Remarks to the Author):

Overall: Cerri et al. investigated the source of negative hemodynamic signals in the caudate putamen (CPu) with optogenetic fMRI, pharmacological interventions, and electrophysiology. Negative BOLD response was often observed in the CPu, but its signal source was unknown. In this manuscript, systemic investigations were thoroughly performed with anesthetized and awake rats, and humans. The important finding is that opioid neuropeptide is a contributor of vasoconstriction, which is novel. This exciting finding will make a major impact on neurovascular coupling and fMRI field by offering interpretation of the negative fMRI signal in neurodegenerative patients or pain.

Major Comments:

1. To read results easily, it is necessary to justify a choice of 40 Hz and to specify whether optogenetic stimulation was done unilaterally or bilaterally (found in the Supplementary materials). Authors mentioned high frequency favors peptide release (page 22 line 418), but it will be useful to provide frequency-dependent optogenetic fMRI as preliminary data for justification. Note that optogenetic stimulation of M1 with 5Hz induced positive BOLD response in the CPu (10.1016/j.neuroimage.2022.119640). This reviewer is wondering whether different stimulation frequency changes polarity of fMRI responses.
2. CPu stimulation induces negative hemodynamic signals in this study, while others reported positive responses (10.1016/j.neuron.2016.06.010; 10.1016/j.celrep.2021.110161). Could the authors explain why results are different? Whether the use of CaMKII promoter differentially targets D1 or D2-MSNs.
3. The relationship between the fMRI response at the stimulation site and CPu response (Fig. 1a-c) is negatively correlated. Careful examination shows a temporal mismatch between the stimulation site and CPu time course. For example, the time course of CPu responding to GPe stimulation (Fig 1c) appears fast with large post-stimulus overshoot, compared to GPe time course. Can the authors comment on this observation?
4. Stimulating GPe (Fig 1c, Fig S2c) (part of the indirect pathway) results in cortex activation, while stimulation of CPu (Fig 1d) leads to strong GPe activity, but no cortex activity. Although the basal ganglia network is not the purpose of this study, explanation for difference in network response may help interpretation of downstream effects. If possible, time courses in downstream targets of GPe such as STN, GPi, and SNr could help interpret whether GPe activation mainly affects CPu.
5. The authors chose to focus on CPu terminals at SNr stimulation (SNr-CPu) for further investigation of feed-forward mechanisms via pharmacological CBV measurements. However, one major concern is whether antidromic activation of CPu (10.1093/cercor/bhaa164) is the main reason for the negative BOLD observed. Could authors mention if the negative CBV observed in CPu could be antidromic or from PFT disinhibition? If this is antidromic activity, I do not believe this invalidates the study. However, the authors later attempt to claim feed-forward mechanism as cause of vasoconstriction in the CPu. If this is antidromic, this is a false claim, and rather should be changed to local CPu mechanisms.
6. A longer stimulation duration with high power (20 mW) and high duty cycle (10 ms/25 ms) was utilized for SNr-CPu with no mention to why. High power stimulation induced heating artifacts (e.g., Fig. S2e) in control YFP rats. Please comment on this.
7. In Figure 2b, very weak CPu neural activity is detected from the SNr-CPu stimulation. This could be due to the weaker downstream disinhibition of the thalamus and cortex which sends efferents to CPu, or due to antidromic activation. CBV time course from M1 or PFT from SNr-CPu can help solve this issue.
8. Drug application studies (Fig 4) are better positioned right after Fig. 2. Then, awake rat studies can link with human fMRI studies easily.
9. In awake rat data (Fig. 3), foot shock-evoked responses were quite noisy. Can the authors comment on motion artifacts?
10. In human studies, it will be good to add time courses of subsections of the striatum for identifying functional selectivity.
11. In the CPu, a positive hemodynamic change is commonly observed for natural stimuli such as somatomotor stimulation, while the negative change was observed for noxious stimuli in rats and humans. Does optogenetic stimulation of M1 and thalamus mimic the noxious stimulus?

Minor comments:

- Page 10 line 181, supplementary figure labeling of MRN terminal in SNr and PFT are switched.
- Authors attempt to correlate negative CBV in CPu in awake rat footshock, anesthetized rat noxious stimulation, and noxious stimulation in humans for translational evidence. Human study consists of anticipated noxious stimulation while the awake rat footshock suggests unanticipated. Could this cause different vascular response within CPu?

Reviewer #2 (Remarks to the Author):

Cerri et al. present a remarkable analysis of the complexity of neurovascular coupling in the rat dorsal striatum. By combining optogenetic methods with fMRI in rats, the authors first show that negative CBV responses in dorsal striatum are elicited by multiple different circuit interventions, each of which induces expected positive responses near stimulus targets. They further demonstrate that analogous stimuli cause CPu neural activity increases detectable by electrode recordings, thus providing a dissociation between neuroimaging and neurophysiology readouts. To try to understand the apparent inverse relationship between dorsal striatal fMRI and electrophysiology results, the authors next perform a series of pharmacological interventions, finding that a muscarinic agonist and two opioid antagonists significantly suppress negative CBV responses evoked by optogenetic stimulation of medium spiny neural projections to the dopaminergic midbrain. Finally, the authors show that noxious peripheral stimuli can induce negative BOLD responses in human striatum, suggesting that mechanisms behind the human result may reflect phenomena observed in rodents.

This paper represents a phenomenal effort with an impressive number of experimental modalities and specific measurement conditions organized into a convincing demonstration that neural activity and hemodynamic responses of opposite polarity may be produced in dorsal striatum. The study is in the tradition of over two decades of studies that have progressively revealed how the biology of neurovascular coupling varies among brain regions and experimental preparations. The authors have done well to try to understand hemodynamic signals in the striatum, and I think that their study will be appropriate for publication in Nature Communications after a small amount of refinement. To help facilitate that, I offer a few comments and questions:

1. The paper currently does relatively little to inform the reader how hemodynamic signals in the dorsal striatum should generally be interpreted in light of the data presented. Instead, the study seems to function more as an extended existence proof that striatal neural activity and hemodynamics can be inversely related. I think the authors could strengthen the paper by going beyond the existence proof idea and trying to indicate more precisely under what conditions a positive striatal signal might reflect increased neural activity and how to distinguish, e.g., positive responses arising from dopamine release from positive responses resulting from suppression of other inputs.
2. Most of the neuroimaging results presented in the paper focus narrowly on the striatum and whatever region is being stimulated, despite the fact that brain-wide fMRI responses were obtained for most experiments. Related to the previous point, I wonder if more holistic consideration of the distributed activity patterns reported in Figure S2 could foster interpretation of striatal fMRI signals with respect to distinct inputs and pathways involved.
3. Despite the extensive use of circuit-specific neurostimulation paradigms in this paper, there is comparatively little discussion of broader circuit-level contributions to the results presented. Many of the stimuli applied in Figures 1-2 could have both direct and indirect postsynaptic consequences, as well as antidromic effects. Some of these could even act in opposition to the driving stimulus, given the prevalence of inhibitory signaling in the basal ganglia. Similarly, pharmacological stimuli such as the opioid antagonists of Figure 4 could have downstream effects, such as reducing dopamine concentration. I think the authors could strengthen their discussion by considering how factors such as these contribute to their observations.

4. The central result of the paper is the observation of negative striatal CBV changes in response to a variety of excitatory stimuli in Figure 1. Although the responses do indeed appear to be primarily negative, however, they have additional features that deserve discussion and perhaps explanation. Two features common to most of the conditions caught my attention: (1) the pronounced post-stimulus "overshoot" visible especially with M1 and GPe stimuli; and (2) the extremely sharp return to baseline seen across the board—much sharper than the response onsets or typical (cortical) hemodynamic offset times. Could the authors comment on the origin of these features and try to reconcile them with the signaling pathways they believe to be involved?

5. The data of Figure 3 are presented as a demonstration that inverse relationships between hemodynamic and neural activity signatures can be observed under naturalistic conditions in awake rats. The relationship between the GCaMP and CBV signals reported seems complex, however. In particular, in Figure 3b, there are clearly features shared between the GCaMP and CBV traces, such as the peak observed after stimulus offset ($t = \sim 5$ s) and the increasing signal from about $t = 15$ -30 s. Is it really fair to say that the two recordings are broadly opposite in sign? Also, at a technical level, have the authors completely ruled out cross-talk between the GCaMP and CBV channels in this experiment?

6. Do the authors have any data about the dependence of fMRI results on stimulus parameters such as intensity and frequency? I don't want them to do further experiments, but it would be very interesting to learn more about what might have been tried already. On a related point, can they comment on the behavioral consequences of stimuli such as those used for Figure 1.

7. The electrode recordings of Figure 2 provide the most direct evidence for increased CPU neural activity under conditions that elicited the negative fMRI responses in Figure 1. Do these results really demonstrate that the net result of the stimuli was to induce neurophysiological activation, however? The experiment of Figure 2b in particular barely elicits a spiking response. Could it be that the stimulus here also induces inhibition in the form of inhibitory potentials and hyperpolarization in neurons that don't contribute to MUA? Even if such inhibitory activity is accompanied by increased spiking by some neurons, the net result could be consistent with the hemodynamic results. Perhaps some increased discussion of this kind of possibility would be merited.

8. As a minor corollary to the previous point, I don't think that the CBV decreases can be necessarily be "attributed to large increases in CPU activity" (lines 343-44). Ambiguity about net activity changes in CPU and the possibility of circuit mechanisms that complicate relationships between the driving stimuli and the responses (point 3) both complicate the picture. I think it would be more accurate to say that the striatal vasoconstriction is "accompanied by increases in CPU activity." This would also be more in keeping with the relatively restrained language of the abstract.

9. The human experiments of Figure 5 are a nice touch to this paper. In describing the TMS procedures on p. 17, I think the authors should say a bit more in the main text about the experimental design, rather than forcing readers to skip down to the methods section. It would also be good to use more cautious language on lines 328-9 regarding what the TMS is accomplishing at a neural level. This should be consistent with the ambiguities highlighted correctly on lines 412-3. Finally, in Figure 5, were the deactivations observed in CPU really the only consequences of the various stimuli? The TMS targets are all regions that connect to many other regions throughout the brain, so it would be surprising if the striatum was the only area where responses were observed, even to the exclusion of local responses. Some more clarification and/or discussion of this issue would be useful.

10. A minor point: on line 56, the text should be edited to denote that the CPU is only the dorsal portion of the striatum, as conventionally defined.

11. I believe that the rightmost graph in Fig. 1d is mislabeled, since it shows negative ChR2 responses in GPe but not CPU.

12. I have the impression that most of the fMRI statistics in the paper count each response in each animal as an independent measurement. This doesn't seem entirely fair to me, since variability due to injections, fiber placement, physiological peculiarities, etc. vary by animal and not by stimulus presentation. Perhaps the authors could find some way to represent inter-animal variability more clearly than I think they have done here.

Reviewer #3 (Remarks to the Author):

Overall Review:

The paper presents a comprehensive assessment of CPU fMRI responses to optogenetic stimulation of CPU neurons or afferents in rats, investigating the relationship between neuronal and hemodynamic activity in the striatum. The authors demonstrate that striatal hemodynamic responses are not solely dependent on neuronal activity and identify opioids as playing a critical role in generating negative fMRI signals. The paper also provides evidence that negative fMRI responses may occur in the human striatum, prompting consideration of local cellular and neurochemical environments in fMRI signal interpretation. The research questions and the results are interesting. The text is generally clearly presented. However, in my detailed description below, I raise some concerns that need to be addressed.

Major:

1. The authors keep referring to decreased CBV as "negative fMRI responses" which is kind of confusing. Is it the decreased T2*-weighted signal or decreased blood volume? It would be recommended to clarify this in the manuscript.
2. It is novel to see decreased CBV, i.e., increased T2* weighted signal in the targeted region. The authors claim it is due to vessel constriction (reduced iron concentration per voxel/ improved homogeneity), but there is another well-known effect of the improved homogeneity by increasing the deoxyhemoglobin/oxyhemoglobin, i.e., positive BOLD signal. Should the authors clarify whether there are still BOLD signals which will contribute to the increased T2* weighted signal. This is a crucial control experiment for the authors to make the statement that vessel constriction is the pure contribution to the increased T2* weighted signals. Iron particle dose-dependent experiment is a potential way to exclude BOLD contribution.
3. It is great to detect potential similarities in striatum/CPU vascular response mechanisms between species. Since CBV was used in rodents and BOLD was used in humans, it would be nice to discuss why the authors didn't perform the BOLD in the same setup in the animal models to further strengthen the similarity statement.
4. The authors claim that decreased BOLD in humans and decreased CBV in rodents share potential similarities. As these two phenomena do not necessarily concurrent, it is better to discuss more detail.
5. Could the authors clarify the rationale for using different species (Rats and mice) in the experiment?
6. In Figure 1, the authors find negative CBV response amplitude in CPU was anti-correlated with the positive responses, what does this finding indicate?
7. Subjects were all male rodents. The authors should comment briefly on this as a limitation.
8. The authors stimulate each of the brain target regions with a 40 Hz pulse, it would be good to explain why 40 Hz was chosen.

REVIEWER COMMENTS

We appreciate the valuable feedback provided by the Reviewers on our manuscript. In the event that our manuscript is accepted for publication in Nature Communications, we intend to make this Response-to-Reviewers document publicly accessible, allowing readers to benefit from these insightful comments and our point-by-point responses. We extend our gratitude to the Reviewers for dedicating their expertise and time to the evaluation process.

- Point-by-point responses: Labeled in green.
- Changes in the revised manuscript: Labeled in blue.

Reviewer #1 (Remarks to the Author):

Overall: Cerri et al. investigated the source of negative hemodynamic signals in the caudate putamen (CPu) with optogenetic fMRI, pharmacological interventions, and electrophysiology. Negative BOLD response was often observed in the CPu, but its signal source was unknown. In this manuscript, systemic investigations were thoroughly performed with anesthetized and awake rats, and humans. The important finding is that opioid neuropeptide is a contributor of vasoconstriction, which is novel. This exciting finding will make a major impact on neurovascular coupling and fMRI field by offering interpretation of the negative fMRI signal in neurodegenerative patients or pain.

[Response] We thank the Reviewer for the kind words summarizing our work. We appreciate the Reviewer's constructive comments and suggestions, which have greatly helped us to improve the manuscript.

Major Comments:

1. To read results easily, it is necessary to justify a choice of 40 Hz and to specify whether optogenetic stimulation was done unilaterally or bilaterally (found in the Supplementary materials). Authors mentioned high frequency favors peptide release (page 22 line 418), but it will be useful to provide frequency-dependent optogenetic fMRI as preliminary data for justification. Note that optogenetic stimulation of M1 with 5 Hz induced positive BOLD response in the CPu (10.1016/j.neuroimage.2022.119640). This reviewer is wondering whether different stimulation frequency changes polarity of fMRI responses.

[Response] We selected the 40 Hz stimulus frequency based on a classical study (Shmuel and Leopold, Human Brain Mapping, 2008; PMID: 18465799), which highlighted gamma-range local field potentials (LFP) and spiking as primary contributors to the BOLD signal. Additionally, prior work in our lab, such as optogenetic stimulation of VTA dopaminergic neurons (10-40 Hz; Decot et al., 2017, Neuropsychopharmacology; PMID: 27515791) and electrical deep brain stimulation of striatal output nuclei (10-400 Hz; Van den berg et al., 2017, Neuroimage; PMID: 27825979), consistently demonstrated the effectiveness of this frequency in eliciting strong striatal fMRI responses. We've clarified this choice in our revised manuscript's Results section, shown below.

To further illustrate that negative striatal CBV changes can occur at lower stimulus frequencies and through various cortical inputs, we conducted optogenetic stimulations at 20 Hz in the anterior insular cortex (AI) and 10 Hz and 20 Hz in the orbitofrontal cortex (OFC). In addition, the AI experiments were executed by using a different duty cycle (20 sec on, 80 sec off), lower pulse-width than some stimulations (5 ms), lower power (10 mW versus 20 mW), and a different virus (Chronos vs Chr2),

whereas the OFC experiments were done in a different rat strain (Wistar). These experiments revealed robust unilateral positive fMRI responses at the stimulation site and stimulus-evoked negative CBV responses within the CPu, as shown in Fig. S4 and S5.

These new Supplementary Figures are referred to in the manuscript Results section as follows, where we conservatively stated that negative CBV responses “could” be evoked under these conditions:

“Optogenetic stimulation was applied to each of these target regions using a 40 Hz pulse train, chosen not only for its alignment with low gamma frequency and its established contribution to the BOLD signal⁶¹ but also based on our prior studies showing pronounced modulation of fMRI signals within the CPu when employing this particular frequency.^{62, 63} This stimulation paradigm yielded local positive CBV changes at the stimulation site while consistently eliciting negative CBV responses within the CPu across all cases (Fig. 1a-c, Fig. S2a-c, and Fig. S3a-c). Furthermore, we extended our findings by demonstrating that negative CBV responses within the CPu could also be evoked at lower stimulus frequencies in other cortical regions projecting to the CPu (Fig. S4, Fig. S5).”

Further, these new supplementary figures are presented in the Supplementary Information accordingly:

Fig. S4: 20 Hz optogenetic stimulation of anterior insular cortex (AI) during CBV fMRI. a fMRI response maps acquired from eYFP control and Chronos subjects following optogenetic stimulation of the right AI. Response maps thresholded to $p < 0.001$, FWE corrected to $\alpha < 0.01$. **b** ROIs used for timeseries extraction were collected from the intersection of the stimulus evoked response map (red) and anatomical boundary of AI (yellow) or CPu (blue), corresponding to the orange and purple regions, respectively. **a-b** Left-to-right corresponds to 1 mm steps in the posterior-to-anterior direction, with the 5th slice from the right located at the anterior commissure (approximately -0.36 mm AP). **c** Stimulation schematics (top left), where AI viral expression and projections to CPu are indicated in green (not all projections shown) and optogenetically stimulated area of AI is indicated in blue; locations of optical fiber tips for AI optogenetic

stimulation (top right); representative tissue cross-section indicating spread of eYFP for from AI injection site (bottom), with 3-D models of the brain to the right indicating the location of the histological slice. **d** CBV time-courses from AI response maps (left) and CPu response maps (right) aligned to stimulation epochs (green bars indicate 20 Hz optogenetic stimulation blocks; data are presented as mean \pm SEM), with corresponding quantified peak amplitude changes (Chronos vs. eYFP Welch's t-test, **** $p < 0.0001$). **a-d** AI stimulation (20 s on, 80 s rest; 20 Hz, 10 mW power at fiber tip, 5 ms pulse-width) during CBV fMRI data: (Chronos $n = 9$ rats, 40 epochs, 80 peaks; eYFP $n = 6$ rats, 18 epochs, 36 peaks).

Fig. S5: 10 and 20 Hz optogenetic stimulation of orbitofrontal cortex (OFC) during CBV fMRI. fMRI response maps were acquired from unilateral optogenetic stimulation of the OFC (5 ms pulse width, 473 nm wavelength, 10 mW power), using a paradigm of 30 s stimulation off, followed by 15 s stimulation on then 60 s off repeated a total of four times, in either the ChR2-containing (left) hemisphere, **a**, at 10 Hz (top) and 20 Hz (bottom), or the eYFP-containing control (right) hemisphere, **b**, at 10 Hz (top) and 20 Hz (bottom), of four rats (3 male, 1 female). **a-b** Response maps were obtained from one-sample t-tests using the AFNI4 3dttest++ command and thresholded to $p < 0.05$ without multiple comparison correction due to the limited statistical power of this proof-of-concept experiment. Instead, a cluster voxel size > 40 was applied. Left-to-right images correspond to 1 mm steps in the posterior-to-anterior direction, with the 5th slice from the right located at the anterior commissure (approximately -0.36 mm AP). Before group-level analysis, data were preprocessed for slice timing and motion corrections with AFNI and spatially aligned to the Tohoku rat brain template² using ITK-SNAP.³ Functional images were smoothed with a 0.5 mm FWHM Gaussian kernel and first-level GLM was conducted via AFNI 3dDeconvolve against the MION hemodynamic response function convolved with the stimulation paradigm, which also included a second-order polynomial curve and motion parameters. **c** Stimulation schematics (left), where OFC viral expression and projections to CPu are indicated in green (not all projections shown) and optogenetically stimulated area of OFC is indicated in blue; locations of optical fiber tips for OFC optogenetic stimulation (right), implanted 0.5 mm above the viral infusion site. **d** CBV time-course data for 10 Hz stimulation (top)

and 20 Hz stimulation (bottom) was extracted from experimenter defined ROIs corresponding to the generated evoked contrast maps, trimmed to the anatomical boundaries of the OFC (left) and CPu (right) regions (green bars indicate optogenetic stimulation blocks; data are presented as mean \pm SEM; n = 4 rats). All other surgical procedures, animal preparation for scanning and scanning protocol, and the percent CBV change for time-course data calculation was as described in the main Methods.

Despite the new data above, we cannot rule out the possibility that certain stimulus frequencies or pulse paradigms might elicit distinct fMRI responses. In the Introduction, we have incorporated additional references (including the one suggested by the reviewer) that discuss these nuances and emphasize variations in observations:

“Previous studies have revealed that heightened neuronal activity does not consistently lead to positive hemodynamic responses in the striatum. In rodent experiments, both positive⁴²⁻⁵⁰ and negative⁵¹⁻⁵⁷ hemodynamic responses in the CPu have been observed where increased neuronal activity was either directly measured or inferred by selective manipulations (i.e., optogenetics). Intriguingly, the occurrence of negative hemodynamic responses in the CPu cannot be easily attributed to the activation of inhibitory neurons, as several studies have reported positive local hemodynamic responses following the selective activation of cortical inhibitory neurons.^{12, 30, 38-40, 49, 58, 59} Consequently, it becomes apparent that the variable hemodynamic responses observed in the CPu may depend on several factors, including the choice of anesthetics, the physiological condition of the subject, the prevailing brain state, as well as the frequency and amplitude of the targeted modulations.”

In addition, we have raised cautions regarding the use of 40 Hz optogenetics in Results:

“While optogenetic stimulation enabled us to precisely control neural circuits capable of driving negative hemodynamic responses in CPu, the synchronized activity induced by such artificial stimulations is unlikely to occur naturally, and at certain stimulation targets may drive neuronal spiking rates outside of physiological ranges.⁸⁴ Further, although the light anesthesia protocol employed for the aforementioned recordings is well established in rodents to facilitate reproducible neuronal and vascular responses and stable physiology,⁸⁵⁻⁸⁹ it is widely known that anesthetics can alter these metrics.⁸⁹⁻⁹² To examine the relationship between CPu hemodynamics and neuronal firing under more naturalistic conditions, we employed spectral fiber-photometry to simultaneously measure neuronal and vascular activity^{51, 93-96} in the CPu of freely-moving awake rats (Fig. 3a).”

Furthermore, we have taken a conservative approach in the Discussion when addressing the use of 40 Hz and the interpretation of respective findings:

“Although these results were obtained in sedated rats, particularly at high stimulus frequencies, and optogenetic manipulations may drive non-physiological neural activity patterns,⁸⁴ fiber photometry recordings in awake and behaving rats suggested that such an inverse relationship of CPu neuronal activity and hemodynamic signals may also occur under more naturalistic conditions.”

“We observed negative hemodynamic responses in CPu as a result of optogenetic stimulation, noxious stimuli, or large peaks in spontaneous local neuronal activity. These conditions feature synchronous, high-frequency neuronal activation (e.g., 40 Hz) which is known to be favorable for peptide release in the peripheral and central nervous systems.¹⁵⁴⁻¹⁵⁸”

Finally, in regards to the Reviewer’s request to clarify in the main manuscript which stimulations were performed bilaterally, we have added “bilateral” before the description of the SNr and SNc targeted stimulations in both the Results section and figure legend for Fig. 1.

2. CPU stimulation induces negative hemodynamic signals in this study, while others reported positive responses (10.1016/j.neuron.2016.06.010; 10.1016/j.celrep.2021.110161). Could the authors explain why results are different? Whether the use of CaMKII promoter differentially targets D1 or D2-MSNs.

[Response] In our Introduction, we highlight the variable hemodynamic responses that have been observed in rodent striatum with increases in neuronal activity, citing the studies brought up by the Reviewer. We have also indicated the possibility of stimulus frequency, physiology, and brain state dependence, as seen in our responses to R1.1 above. Specifically, it is not directly evident why some groups have reported vasoconstriction with CPU stimulation, while other teams have reported vasodilation. Given the high levels of methodological variation in the relevant studies (e.g., subjects, anesthesia/sedation protocols, experimental procedures, etc.) we do not wish to extensively speculate. However, our finding of striatal activity-dependent vasoconstriction is bolstered by its reliable presence across arousal state (sedated and awake), species (rat and human), and circuit stimulation approaches.

We conducted histological examinations of virally-mediated ChR2 expression with a CaMKII-promoter AAV (Fig. S1), demonstrating that ChR2 was strongly expressed in GPe and SNr, highlighting expression in the indirect and direct pathways, respectively (see also Klug et al., 2012, PLoS ONE; PMID: 23028932). In contrast, we did not detect ChR2 expression in striatal interneurons. Although it is possible that CPU neurons of either pathway differentially contributed to the observed striatal fMRI signals, our axon terminal stimulation experiments of striatonigral afferents were intended to isolate the direct pathway. Direct pathway stimulation at the level of striatonigral terminals elicited robust striatal vasoconstriction, suggesting that stimulating a mixed population of striatal direct and indirect pathway neurons, or the direct pathway more selectively, both generate CPU vasoconstriction under our experimental condition.

We now emphasize in the main manuscript Results section our supplemental histology figure demonstrating the CaMKII expression profile:

“Using a CaMKII α promoter to selectively target ChR2 to striatal MSNs (Fig. S1d-g), we indeed observed striatal CBV decreases in response to optogenetic stimulation of MSN cell bodies (Fig. 1d, Fig. S2d, and Fig. S3d) or bilateral stimulation of the striatal direct pathway terminals within the substantia nigra pars reticulata (SNr) (Fig. 1e, Fig. S2e, and Fig. S3e).”

3. The relationship between the fMRI response at the stimulation site and CPU response (Fig. 1a-c) is negatively correlated. Careful examination shows a temporal mismatch between the stimulation site and CPU time course. For example, the time course of CPU responding to GPe stimulation (Fig 1c) appears fast with large post-stimulus overshoot, compared to GPe time course. Can the authors comment on this observation?

[Response] We deliberately chose fMRI to study evoked hemodynamic responses while preserving the integrity of brain circuits. With this approach (as compared to experiments such as patch clamp), we expect more distinct temporal activation profiles between stimulation areas and their downstream targets. Specifically, we anticipate that firing patterns will not propagate uniformly across synapses, and the vascular responses elicited by neuronal activity may exhibit variations across different brain regions. As the Reviewer pointed out, these were indeed observed in our data.

The post-stimulus overshoot is likely the convergence of two conflicting forces: vasoconstriction from active neurotransmission and dilation from metabolic feedback mechanisms. These forces may not act on the same time scale or occur with the same intensity, as each force may be driven by different cell types and/or neurotransmitters that could contribute to or oppose vasodilation driven by energy consumption. That the overshoot occurs rapidly once the stimulation ceases may indicate that our

stimulus is driving a strong vasoconstrictive process, consistent with our hypothesis that driving MSN activity in CPU evokes opioid-induced vasoconstriction. Biphasic responses to stimulation due to evoked forces acting on different time scales have been observed in other experimental paradigms, including a recent paper describing ultra-slow vasodilation via Substance P in rats while under anesthesia (Vo et al., 2023, PNAS; PMID: 37098063). Speculation beyond that these forces are involved cannot be supported without experiments that go beyond the scope of this work. We now draw attention to these response features in the main manuscript.

In Fig. 1 legend:

“Note that multiple circuits show a post-stimulus “overshoot” or faster offset times than others. The result for each circuit likely reflects an accumulated confluence of vasoconstrictive and dilative forces operating at different time scales over the course of the stimuli, including metabolism, activated synapses, and vasoactive neurotransmission.”

In Discussion:

“We observed that certain circuit manipulations, like GPe stimulation, generated CPU fMRI responses with distinct peak timing and a robust post-stimulus overshoot (Fig. 1). These differences in response timing may arise from the non-uniform propagation of firing patterns across synapses and variations in vascular responses across brain regions. The overshoot likely results from the convergence of conflicting forces: vasoconstriction due to neurotransmission and dilation from metabolic feedback mechanisms, which may operate on different timescales and intensities.”

4. Stimulating GPe (Fig 1c, Fig S2c) (part of the indirect pathway) results in cortex activation, while stimulation of CPU (Fig 1d) leads to strong GPe activity, but no cortex activity. Although the basal ganglia network is not the purpose of this study, explanation for difference in network response may help interpretation of downstream effects. If possible, time courses in downstream targets of GPe such as STN, GPi, and SNr could help interpret whether GPe activation mainly affects CPU.

[Response] Indeed, we observed positive CBV responses in the frontal cortex during optogenetic GPe stimulation, which aligns with an earlier study where similar frontal cortical activation was reported with GPe electrical stimulation (Van den berg et al., 2017, Neuroimage; PMID: 27825979). Given the existence of direct projections from GPe to frontal cortex (Saunders et al., 2015, Nature; PMID: 25739505), our data suggest that increased cortical metabolic demand, resulting from the activation of pallidocortical fibers, could contribute to the observed positive CBV responses in the cortex with GPe stimulation, a pattern distinct from CPU optogenetic stimulation. While the intricacies of GPe-induced downstream time courses warrant further investigation, we refrain from making definitive statements without a comprehensive dissection of the circuit dynamics, which would require additional opto-fMRI and electrophysiology experiments beyond the scope of the present study. Nevertheless, taking GPe manipulation as an example to highlight the complexity of activity patterns, we have added the following text to the Discussion to foster interpretation of relevant fMRI signals, including observed frontal cortical activation with GPe stimulation:

“Interestingly, CPU vasoconstriction was observed with optogenetic activation of brain regions that varied in their principal neurochemical phenotype (e.g., glutamatergic or GABAergic) and stimulation-induced brain-wide fMRI response patterns (Fig. 1 and Fig. S2). Indeed, the most notable feature among these regions is their shared monosynaptic connectivity to CPU, suggesting that this direct input may be a crucial factor for generating activity-induced CPU vasoconstriction. The observed responses to GPe stimulation further highlight the complexity of region-dependent fMRI signal changes. GPe neurons, which are largely GABAergic and inhibitory, send direct projections to both the CPU and STN,¹¹⁴ leading

to the expectation that GPe has an impact on the activity of both CPu and STN. Notably these changes were observed in our data, albeit contributions from poly-synaptic pathways cannot be ruled out. CPu stimulation might be expected to predominantly decrease firing rates in GPe due to the extensive, inhibitory striatopallidal output,¹¹⁵ while direct excitatory optogenetic stimulation of GPe is expected to increase GPe neuronal activity. Despite these likely divergent changes in GPe firing activity, both manipulations drive GPe CBV increases. We speculate that both manipulations may lead to increased metabolic demand within the GPe, potentially stemming from either local GPe stimulation-evoked neuronal activity or synaptic input activity in GPe induced by CPu stimulation.¹¹⁶⁻¹¹⁸ These findings in GPe align with prior studies reporting positive hemodynamic responses following the selective activation inhibitory neurons in the cerebral cortex.^{12, 30, 38-40, 49, 58, 59} Nevertheless, they also bring attention to the intriguing observation of negative CBV responses in CPu during the same experimental session, despite CPu neurons sharing a predominately inhibitory nature with GPe.³⁰

5. The authors chose to focus on CPu terminals at SNr stimulation (SNr-CPu) for further investigation of feed-forward mechanisms via pharmacological CBV measurements. However, one major concern is whether antidromic activation of CPu (10.1093/cercor/bhaa164) is the main reason for the negative BOLD observed. Could authors mention if the negative CBV observed in CPu could be antidromic or from PFT disinhibition? If this is antidromic activity, I do not believe this invalidates the study. However, the authors later attempt to claim feed-forward mechanism as cause of vasoconstriction in the CPu. If this is antidromic, this is a false claim, and rather should be changed to local CPu mechanisms.

[Response] We appreciate the Reviewer's comment and apologize for any lack of clarity. We would like to clarify that the striatal vasoconstriction is indeed induced by local feed-forward microcircuitry involving MSN activation, opioid release, and subsequent vasoconstriction within the CPu. In our study, we employed an antidromic optogenetic stimulation approach to facilitate MSN activation, such that stimulation fibers can be placed further away from the striatum. As the Reviewer pointed out, the specific pathway by which this stimulation reaches the striatum should not invalidate our conclusion because we observed increased striatal spiking and LFP changes through electrophysiological measurements.

To clarify the Reviewer's concern, we have provided clarification and referenced the suggested citation in the manuscript.

In Results:

“Specifically, stimulation of CPu terminals at the SNr can evoke antidromic activities in direct pathway MSNs,⁶⁶⁻⁶⁸ while allowing the optical fiber to be placed farther away from the CPu region of interest.”

“It is generally accepted that positive changes in neuronal activity and fMRI signal are coupled, as regional blood flow increases to provide sufficient local energy supplies to satisfy metabolic demand; this is known as neurovascular coupling.⁹⁹ However, this is insufficient to explain the neurovascular relationship in CPu, as shown under our experimental conditions. Although it was originally believed that neurovascular coupling was dictated by metabolic feedback mechanisms, increasing experimental evidence illustrates the dominant role of activity-dependent, feed-forward mechanisms instead.^{100, 101} Neurochemicals released via neuronal activity can alter vascular tone either directly by interacting with vascular smooth muscle cells or pericytes or indirectly through astrocytes, endothelial cells, and/or interneurons.^{100, 102, 103} To evaluate whether activity-dependent, feed-forward neurochemical release could be responsible for the

vasoconstriction we have observed in CPu, we performed intra-CPu pharmacology experiments with optogenetic fMRI by stimulating CPu terminals in SNr.”

“These data suggest that acute feed-forward opioid signaling can transiently induce vasoconstriction in CPu.”

In Discussion:

“Importantly, our findings do not preclude the involvement of neurometabolic feedback mechanisms widely assumed to contribute to fMRI responses, but rather highlight additional neurochemical feed-forward mechanisms that can play an outsized role in determining fMRI response polarity; as such, other stimulation or behavioral conditions may not produce negative hemodynamic responses, especially those that are less likely to engage CPu opioid transmission.”

6. A longer stimulation duration with high power (20 mW) and high duty cycle (10 ms/25 ms) was utilized for SNr-CPu with no mention to why. High power stimulation induced heating artifacts (e.g., Fig. S2e) in control YFP rats. Please comment on this.

[Response] The CPu-related circuits of interest have not yet been comprehensively studied with fMRI, we used long stimulation durations and high power to evoke the most robust, reliable, and reproducible fMRI signal changes possible; however, we acknowledge that these stimulations are not optimized, and that the intensity of this stimulation may not be necessary for future studies of this circuit. As shown in Fig. 1 and Fig. S2, we performed these experiments with rigorous eYFP controls. Additionally, we are not concerned about heating artifacts reaching the CPu from SNr stimulations due to the physical distance between these brain regions. In line with this, we used lower power for other circuit stimulations where stimulation sites and the CPu were closer together. Further, any heating effects on the fMRI signal would affect experimental and control subjects equally and all statistical comparisons were made against controls. We have added the following to the Methods.

“Because these circuits of interest have not yet been comprehensively studied with fMRI, we used long stimulation durations and high power to evoke the most robust, reliable, and reproducible fMRI signal changes possible. All experiments were accompanied with rigorous eYFP controls to consider potential nonspecific effects and heating confounds (Fig. 1, Fig. S2, Fig. S4, Fig. S5, and Fig. S16).”

7. In Figure 2b, very weak CPu neural activity is detected from the SNr-CPu stimulation. This could be due to the weaker downstream disinhibition of the thalamus and cortex which sends efferents to CPu, or due to antidromic activation. CBV time course from M1 or PFT from SNr-CPu can help solve this issue.

[Response] We concur with the Reviewer that optogenetic stimulation of striatonigral projection terminals could influence striatal neural activity and hemodynamic signals through various putative mechanisms. These mechanisms encompass antidromic spiking as well as several polysynaptic routes, such as thalamus->striatum and thalamus->cortex->striatum pathways. Notably, our data demonstrate that optogenetic stimulation of both the motor cortex and parafascicular thalamus induces striatal negative fMRI signals, indicating that corticostriatal and thalamostriatal inputs can both elicit such responses.

However, we acknowledge that the slow time course of hemodynamic signals and potential variations in neurovascular coupling among these cortical and thalamic regions may limit the ability of comparative

CBV time course analyses to definitively pinpoint the specific circuit-level mechanism(s) governing striatal activity within the context of our present experiments (as discussed in our responses in R1.3). While we report that such stimulation elicits electrophysiological activity changes (Fig. 2), determining whether these changes predominantly arise from antidromic or feed-forward thalamic or cortical activities would require additional optogenetic inhibition at cortical and thalamic sites, in addition to the CPU terminal manipulation.

In light of these complexities, we respectfully propose the inclusion of an additional statement in the manuscript Discussion that highlights some potential circuit-level mechanisms of action, including those considerations raised by the Reviewer.

“It is worth noting that optogenetic stimulation of CPU projection terminals in SNr resulted in comparatively weaker CPU neuronal responses, possibly due to antidromic stimulation activating only approximately 50% of MSNs (i.e., the direct pathway MSNs).³⁰ The involvement of local inhibitory potentials and hyperpolarization could further complicate the generation of the fMRI signal. Additionally, it should be noted that optogenetic stimulation of CPU terminals in SNr can influence CPU activity through various circuit mechanisms, including polysynaptic routes involving thalamus and cortex. Indeed, our M1 and PfT optogenetic stimulation data support the idea that corticostriatal and thalamostriatal inputs can induce CPU activity changes. We acknowledge that the slow hemodynamic signal time course and variations in neurovascular coupling among different brain regions may hinder the definitive identification of specific circuit-level mechanisms using the data collected in this study. Future investigations employing optogenetic inhibition¹¹⁹ at cortical and thalamic sites, in addition to CPU terminal stimulation, may be necessary to further dissect these specific signaling pathways.”

8. Drug application studies (Fig 4) are better positioned right after Fig. 2. Then, awake rat studies can link with human fMRI studies easily.

[Response] We respectfully choose to keep the original figure order. The electrophysiology and oxygen time-course results obtained in anesthetized rats for Fig. 2 are validated by the calcium signaling and cerebral blood volume results, respectively, obtained in awake rats for Fig. 3. Validating that evoked negative fMRI responses also occur in awake rats assures the reader that this phenomenon is relevant to evoked hemodynamic responses outside of anesthetized conditions and optogenetic stimuli. The current figure order presents all data that validate the reproducibility of the negative hemodynamic signals in animals before presenting data that attempt to probe underlying causes of the evoked negative signal (i.e., the drug application studies in Fig. 4). We thank the Reviewer for understanding the rationale of this arrangement.

9. In awake rat data (Fig. 3), foot shock-evoked responses were quite noisy. Can the authors comment on motion artifacts?

[Response] We thank the Reviewer for this comment. We have added the following to the Fig. 3 legend.

“Additionally, while motion confounds may be present in our footshock data, motion typically induces same directional changes in both the green and red spectra,⁹⁸ so the generally opposing polarity of GCaMP and CBV signals observed here, particularly at stimulation onset, argues against motion as a major contributing factor in these signals.”

10. In human studies, it will be good to add time courses of subsections of the striatum for identifying functional selectivity.

[Response] Time courses from various striatal subsections in human fMRI experiments, encompassing data from both the transcutaneous electrical nerve stimulation dataset obtained at UNC and the transcranial magnetic stimulation dataset acquired at Stanford, have been added to the supplementary materials. These datasets have been made publicly accessible to the research community for further causal analyses (the UNC dataset through this study and the Stanford dataset provided by the Atkins lab via NIMH). We would like to clarify that we did not to include time series from the TMS stimulation sites due to the presence of fMRI susceptibility artifacts. In addition, TMS pulses were delivered with a variable inter-trial interval, and these intervals were jittered with various delays as part of a fast event-related design, thus deconstructing the average timeseries response for a single pulse would require additional modeling and assumptions that are beyond the scope of this study. Nonetheless, the overall TMS pulse timing paradigm was identical between subjects so we were able to present an averaged time course for the entire TMS session.

These new supplementary figures, which also include unmasked response maps, are now referenced in the main manuscript Results, and referred to in greater detail in the Fig. 6 legend:

“Unmasked versions of the human fMRI response maps displayed here as well as corresponding time courses are shown for noxious forearm stimulation, M1 TMS, aMFG TMS, and pMFG TMS in Fig. S12, Fig. S13, Fig. S14, and Fig. S15, respectively.”

These new figures and their legends are included in the following format in Supplementary Information:

Fig. S12: Unmasked human response maps and striatum timeseries from noxious peripheral stimulation in Fig. 6b. **a** Unmasked human BOLD fMRI response maps corresponding to the masked response maps presented in Fig. 6b. **b** Human striatum anatomical ROI mask used for time-course extraction, color-coded by striatal compartment. **c** Stimulation aligned (green bars indicate stimulation ON), BOLD fMRI time-courses from each bilateral striatal compartment ($n = 7$ subjects, 11 scans; data are shown as subject mean \pm SEM).

Fig. S13: Unmasked human response maps and striatum timeseries from right M1 TMS in Fig. 6c. **a** Unmasked human BOLD fMRI response maps corresponding to the masked response maps presented in Fig. 6c. **b** Human striatum anatomical ROI mask used for time-course extraction, color-coded by hemisphere and striatal compartment. **c** BOLD fMRI time-courses from each unilateral striatal compartment during fast event-related delivery of TMS pulses (pulse delivery was consistent between subjects and pulses are indicated by green bars; $n = 79$ subjects; data are shown as subject mean \pm SEM).

Fig. S14: Unmasked human response maps and striatum timeseries from right aMFG TMS in Fig. 6d. **a** Unmasked human BOLD fMRI response maps corresponding to the masked response maps presented in Fig. 6d. **b** Human striatum anatomical ROI mask used for time-course extraction, color-coded by hemisphere and striatal compartment. **c** BOLD fMRI time-courses from each unilateral striatal

compartment during fast event-related delivery of TMS pulses (pulse delivery was consistent between subjects and pulses are indicated by green bars; n = 80 subjects; data are shown as subject mean \pm SEM).

Fig. S15: Unmasked human response maps and striatum timeseries from right pMFG TMS in Fig 6e. a Unmasked human BOLD fMRI response maps corresponding to the masked response maps presented in Fig. 6e. **b** Human striatum anatomical ROI mask used for time-course extraction, color-coded by hemisphere and striatal compartment. **c** BOLD fMRI time-courses from each unilateral striatal compartment during fast event-related delivery of TMS pulses (pulse delivery was consistent between subjects and pulses are indicated by green bars; n = 79 subjects; data are shown as subject mean \pm SEM).

11. In the CPU, a positive hemodynamic change is commonly observed for natural stimuli such as somatomotor stimulation, while the negative change was observed for noxious stimuli in rats and humans. Does optogenetic stimulation of M1 and thalamus mimic the noxious stimulus?

[Response] The Reviewer raises an interesting question of whether optogenetic circuit manipulations that drive CPU vasoconstriction, including optogenetic stimulation of M1 or parafascicular thalamus, are noxious. To our knowledge, there have been no reports of optogenetic stimulation of the rodent M1 or parafascicular thalamus serving as noxious stimuli. However, the intralaminar thalamic nuclei, including the parafascicular thalamus, have been widely implicated in the affective processing of pain (Mercer-Lindsay et al., 2021, Science Translational Medicine; PMID: 34757810). A careful evaluation of how striatal vasoconstriction relates to pain processing, including the potential relevance of this fMRI signal as a biomarker for pain processing, is an exciting area for future research. To highlight the conditions that yielded negative CBV changes, we have added the following in the Discussion:

“Compared to naturalistic stimuli that drive transient sensory-motor activity, optogenetic manipulations elicit more synchronized activity in the striatum, whereas noxious stimuli recruit opioidergic signaling. These conditions may be necessary to induce decreases in striatal CBV.”

Minor comments:

- Page 10 line 181, supplementary figure labeling of MSN terminal in SNr and PfT are switched.

[Response] We thank the Reviewer for bringing this to our attention. The figure caption has been fixed to now refer to the correct labels.

- Authors attempt to correlate negative CBV in CPu in awake rat footshock, anesthetized rat noxious stimulation, and noxious stimulation in humans for translational evidence. Human study consists of anticipated noxious stimulation while the awake rat footshock suggests unanticipated. Could this cause different vascular response within CPu?

[Response] We appreciate the Reviewer for bringing this to our attention. Upon reflection, we recognize that our choice of wording may have led to confusion, and we have subsequently made the following correction:

“To determine whether our rodent model findings could be used to inform human fMRI data interpretation, we measured blood oxygen level dependent (BOLD) fMRI signals in the striatum (caudate, putamen, ventral striatum) (Fig. 6a) of awake human subjects during stimulation patterns in which increased striatal activity may occur.”

In addition, the anticipation of pain is indeed a documented contributor to pain-related fMRI signals (e.g., Ploghaus et al., 2000, PNAS; PMID: 10908676). It is therefore notable that striatal vasoconstriction was a shared response in our rodent and human imaging data. This may suggest that striatal vasoconstriction *per se* does not signal whether a noxious stimulus was anticipated or not. Future studies are needed to address the differences between the responses to anticipated and unanticipated stimuli. The following has been added to the Fig. 6 legend.

“These findings indicate that anticipated noxious stimulation in human subjects and unanticipated footshock in awake rats (Fig. 3) may both contribute to shaping negative striatal hemodynamic responses.”

Reviewer #2 (Remarks to the Author):

Cerri et al. present a remarkable analysis of the complexity of neurovascular coupling in the rat dorsal striatum. By combining optogenetic methods with fMRI in rats, the authors first show that negative CBV responses in dorsal striatum are elicited by multiple different circuit interventions, each of which induces expected positive responses near stimulus targets. They further demonstrate that analogous stimuli cause CPU neural activity increases detectable by electrode recordings, thus providing a dissociation between neuroimaging and neurophysiology readouts. To try to understand the apparent inverse relationship between dorsal striatal fMRI and electrophysiology results, the authors next perform a series of pharmacological interventions, finding that a muscarinic agonist and two opioid antagonists significantly suppress negative CBV responses evoked by optogenetic stimulation of medium spiny neural projections to the dopaminergic midbrain. Finally, the authors show that noxious peripheral stimuli can induce negative BOLD responses in human striatum, suggesting that mechanisms behind the human result may reflect phenomena observed in rodents.

This paper represents a phenomenal effort with an impressive number of experimental modalities and specific measurement conditions organized into a convincing demonstration that neural activity and hemodynamic responses of opposite polarity may be produced in dorsal striatum. The study is in the tradition of over two decades of studies that have progressively revealed how the biology of neurovascular coupling varies among brain regions and experimental preparations. The authors have done well to try to understand hemodynamic signals in the striatum, and I think that their study will be appropriate for publication in Nature Communications after a small amount of refinement. To help facilitate that, I offer a few comments and questions:

[Response] We are pleased that the reviewer was impressed by our efforts and found our results to be convincing. The reviewer's specific comments were insightful and addressing them in detail, as noted below, has helped us further improve our manuscript.

1. The paper currently does relatively little to inform the reader how hemodynamic signals in the dorsal striatum should generally be interpreted in light of the data presented. Instead, the study seems to function more as an extended existence proof that striatal neural activity and hemodynamics can be inversely related. I think the authors could strengthen the paper by going beyond the existence proof idea and trying to indicate more precisely under what conditions a positive striatal signal might reflect increased neural activity and how to distinguish, e.g., positive responses arising from dopamine release from positive responses resulting from suppression of other inputs.

[Response] Our experimental dissection on the bases of dorsal striatal fMRI signals suggests numerous contributing factors, most notably metabolic demand and neurochemical signaling. Based on our findings, we posit that increases in local striatal activity drive vasoconstriction and negative fMRI signals in an opioid signaling-dependent manner. In contrast, dopaminergic signaling appears to be a significant driver of striatal vasodilation and positive fMRI signals. None of these factors exist in isolation, and dorsal striatal fMRI signals likely reflect their competition and/or summation. However, some of these factors may hold larger weighting than others. For example, optogenetic stimulation of SNc dopamine neurons evoked both increased striatal dopamine concentrations and neuronal firing rate increases (Fig. 2c, e). This stimulus evoked positive striatal fMRI signals suggests that dopaminergic signaling may play an outsized role relative to local energy demand in the formation of striatal fMRI signals.

The following sections in the Discussion now summarize how hemodynamic signals in the dorsal striatum should generally be interpreted:

“We observed negative hemodynamic responses in CPU as a result of optogenetic stimulation, noxious stimuli, or large peaks in spontaneous local neuronal activity. These conditions feature synchronous, high-frequency neuronal activation (e.g., 40 Hz) which is known to be favorable for peptide release in the peripheral and central nervous systems.¹⁵⁵⁻¹⁵⁹ Given that we also observed attenuation of negative CBV responses in CPU by opioid antagonists and acute vasoconstriction of CPU microvessels to the Enk analog, DADLE, it is plausible that the negative CPU hemodynamic responses observed here were produced under conditions featuring activity-dependent opioid transmission. Further, these activity-dependent negative CBV responses may be unique to the CPU and not a characteristic of GABAergic brain regions in general, as we observed positive fMRI responses in the GPe and SNr under the same setting.”

“In summary, our results demonstrate that hemodynamic response polarity in the CPU cannot be solely determined by neuronal activity, but rather requires additional, circuit-specific, neurochemical context. Within the CPU, neuromodulators such as DA and opioids may play important contributing roles for the presence and polarity of hemodynamic responses. This information may be critical for interpreting fMRI data, especially in disorders where DA and/or opioid signaling are dysregulated or altered as treatment.”

2. Most of the neuroimaging results presented in the paper focus narrowly on the striatum and whatever region is being stimulated, despite the fact that brain-wide fMRI responses were obtained for most experiments. Related to the previous point, I wonder if more holistic consideration of the distributed activity patterns reported in Figure S2 could foster interpretation of striatal fMRI signals with respect to distinct inputs and pathways involved.

[Response] The 12-slice fMRI maps in Fig. S2 highlight a remarkable degree of heterogeneity in the brain-wide responses to circuit manipulations that have the shared feature of eliciting striatal vasoconstriction. These manipulations involved either direct stimulation of striatal neurons, or regions that send dense monosynaptic projections to striatum. In examining the resulting brain-wide responses, we do not observe extra-striatal territories with consistent responsivity to the same manipulations, suggesting that their convergent connectivity to striatum, rather than any secondary functional (e.g., glutamatergic or GABAergic synapses) or anatomical features, is a critical element in their capacity to elicit striatal vasoconstriction. Nevertheless, taking GPe manipulation as an example to highlight the complexity of activity patterns, we have added the following text to the Discussion to foster interpretation of relevant fMRI signals.

“Interestingly, CPU vasoconstriction was observed with optogenetic activation of brain regions that varied in their principal neurochemical phenotype (e.g., glutamatergic or GABAergic) and stimulation-induced brain-wide fMRI response patterns (Fig. 1 and Fig. S2). Indeed, the most notable feature among these regions is their shared monosynaptic connectivity to CPU, suggesting that this direct input may be a crucial factor for generating activity-induced CPU vasoconstriction. The observed responses to GPe stimulation further highlight the complexity of region-dependent fMRI signal changes. GPe neurons, which are largely GABAergic and inhibitory, send direct projections to both the CPU and STN,¹¹⁴ leading to the expectation that GPe has an impact on the activity of both CPU and STN. Notably these changes were observed in our data, albeit contributions from poly-synaptic pathways cannot be ruled out. CPU stimulation might be expected to predominantly decrease firing rates in GPe due to the extensive, inhibitory striatopallidal output,¹¹⁵ while direct excitatory optogenetic stimulation of GPe is expected to

increase GPe neuronal activity. Despite these likely divergent changes in GPe firing activity, both manipulations drive GPe CBV increases. We speculate that both manipulations may lead to increased metabolic demand within the GPe, potentially stemming from either local GPe stimulation-evoked neuronal activity or synaptic input activity in GPe induced by CPU stimulation.¹¹⁶⁻¹¹⁸ These findings in GPe align with prior studies reporting positive hemodynamic responses following the selective activation inhibitory neurons in the cerebral cortex.^{12, 30, 38-40, 49, 58, 59} Nevertheless, they also bring attention to the intriguing observation of negative CBV responses in CPU during the same experimental session, despite CPU neurons sharing a predominately inhibitory nature with GPe.³⁰

3. Despite the extensive use of circuit-specific neurostimulation paradigms in this paper, there is comparatively little discussion of broader circuit-level contributions to the results presented. Many of the stimuli applied in Figures 1-2 could have both direct and indirect postsynaptic consequences, as well as antidromic effects. Some of these could even act in opposition to the driving stimulus, given the prevalence of inhibitory signaling in the basal ganglia. Similarly, pharmacological stimuli such as the opioid antagonists of Figure 4 could have downstream effects, such as reducing dopamine concentration. I think the authors could strengthen their discussion by considering how factors such as these contribute to their observations.

[Response] We greatly value the insights provided by the reviewer. In response, we have expanded our discussion of GPe stimulation (as detailed in R2.2 above) and introduced additional discussion regarding CPU terminal stimulation below. These two experiments represent the chosen circuits likely to result in poly-synaptic influences on CPU signals, in contrast to other potentially more direct targets such as M1, PFT, and SNc. Given the protracted time course of hemodynamic signals and the potential variability in neurovascular coupling across diverse cortical and thalamic regions, precisely identifying the specific circuit-level mechanisms governing striatal activity within the confines of our current experiments is a complex task. To gain a deeper understanding of the specific signaling cascades, it would require further optogenetic inhibition at relay brain regions beyond the primary manipulation sites explored in our study. In light of this, we have included in the Discussion an elaboration of potential circuit-level mechanisms that may underlie our observed results, incorporating the insightful points raised by the Reviewer.

“It is worth noting that optogenetic stimulation of CPU projection terminals in SNr resulted in comparatively weaker CPU neuronal responses, possibly due to antidromic stimulation activating only approximately 50% of MSNs (i.e., the direct pathway MSNs).³⁰ The involvement of local inhibitory potentials and hyperpolarization could further complicate the generation of the fMRI signal. Additionally, it should be noted that optogenetic stimulation of CPU terminals in SNr can influence CPU activity through various circuit mechanisms, including polysynaptic routes involving thalamus and cortex. Indeed, our M1 and PFT optogenetic stimulation data support the idea that corticostriatal and thalamostriatal inputs can induce CPU activity changes. We acknowledge that the slow hemodynamic signal time course and variations in neurovascular coupling among different brain regions may hinder the definitive identification of specific circuit-level mechanisms using the data collected in this study. Future investigations employing optogenetic inhibition¹¹⁹ at cortical and thalamic sites, in addition to CPU terminal stimulation, may be necessary to further dissect these specific signaling pathways.”

The reviewer also raises the important point that our pharmacologic interventions may have diverse actions within the striatum that contribute to the observed drug-induced changes in hemodynamic responsivity to striatal stimulation. Given the likely complexities of these actions, which could possibly include synergistic and/or competing influences on hemodynamic signal generation, only loose

speculation is possible. We have now added a statement in our Discussion that highlights the likely diverse actions of many of the drugs used in our pharmacologic experiments, including actions on the dopamine system.

“Many of the drugs used in our study could alter neuronal activity and neurochemical concentrations within the CPU, including dopaminergic tone (e.g., via disruptions in opioidergic and/or cholinergic control of striatal dopamine release),¹²⁴⁻¹²⁶ as well as possibly more direct actions on vascular signaling.”

4. The central result of the paper is the observation of negative striatal CBV changes in response to a variety of excitatory stimuli in Figure 1. Although the responses do indeed appear to be primarily negative, however, they have additional features that deserve discussion and perhaps explanation. Two features common to most of the conditions caught my attention: (1) the pronounced post-stimulus “overshoot” visible especially with M1 and GPe stimuli; and (2) the extremely sharp return to baseline seen across the board—much sharper than the response onsets or typical (cortical) hemodynamic offset times. Could the authors comment on the origin of these features and try to reconcile them with the signaling pathways they believe to be involved?

[Response] We thank the reviewer for this question and have used the direction provided by these observations to add additional discussions on the post-stimulus overshoot and variability in offset times observed in our fMRI results. We have added the following to the Fig. 1 legend:

“Note that multiple circuits show a post-stimulus “overshoot” or faster offset times than others. The result for each circuit likely reflects an accumulated confluence of vasoconstrictive and dilative forces operating at different time scales over the course of the stimuli, including metabolism, activated synapses, and vasoactive neurotransmission.”

In addition, we have expanded our Discussion regarding the overshoot. We reiterate that the post-stimulus overshoot likely results from the convergence of two opposing forces: vasoconstriction triggered by active neurotransmission and dilation driven by metabolic processes. These forces may not operate on the same time scale or with the same intensity, as they can be influenced by different cell types and neurotransmitters that either promote or counteract vasodilation associated with energy consumption. The rapid return to baseline following stimulation cessation suggests that our stimulus induces robust vasoconstriction, causing blood vessels to release and rebound from intense constriction. This aligns with our hypothesis that CPU MSN activity provokes opioid-induced vasoconstriction, supported by evidence that opioid agonists can induce strong vasoconstriction in pre-constricted vessels (Fig. 5). Additionally, biphasic responses to stimulation, resulting from forces acting on different time scales, have been documented in other experimental paradigms, such as the recent study on ultra-slow vasodilation via Substance P in anesthetized rats (Vo et al., 2023, PNAS; PMID: 37098063). We have included these points in our Discussion section. However, it's important to note that further experimentation beyond the scope of this study would be necessary to provide definitive evidence regarding the involvement of these forces.

“We observed that certain circuit manipulations, like GPe stimulation, generated CPU fMRI responses with distinct peak timing and a robust post-stimulus overshoot (Fig. 1). These differences in response timing may arise from the non-uniform propagation of firing patterns across synapses and variations in vascular responses across brain regions. The overshoot likely results from the convergence of conflicting forces: vasoconstriction due to neurotransmission and dilation from metabolic feedback mechanisms, which may operate on different timescales and intensities.”

5. The data of Figure 3 are presented as a demonstration that inverse relationships between hemodynamic and neural activity signatures can be observed under naturalistic conditions in awake rats. The relationship between the GCaMP and CBV signals reported seems complex, however. In particular, in Figure 3b, there are clearly features shared between the GCaMP and CBV traces, such as the peak observed after stimulus offset ($t \sim 5$ s) and the increasing signal from about $t = 15-30$ s. Is it really fair to say that the two recordings are broadly opposite in sign? Also, at a technical level, have the authors completely ruled out cross-talk between the GCaMP and CBV channels in this experiment?

[Response] We appreciate the reviewer's detailed observation. Indeed, there are shared features between the GCaMP and CBV traces, as suggested. While we cannot completely dismiss the possibility of cross-talk between the GCaMP and CBV channels, we want to clarify that the data were carefully corrected for cross-talk using the approach we recently published (Zhang et al., 2022, Cell Reports Methods; PMID: 35880016; and Zhang et al., 2022, STAR Protocols; PMID: 35776651). Further, the results presented in Fig. 3c provide additional supporting evidence for the claimed relationship. Moreover, the statement regarding the opposite polarity was based on a comparison of the data against the baseline period, considering a substantial number of trials (79 footshock trials and 66 spontaneous GCaMP activation peaks). These analyses consistently showed that CBV values hardly exceed zero, while GCaMP values rarely drop below zero following the baseline period. However, we acknowledge the need for caution and have included a clarification in the Fig. 3 legend:

“The footshock and spontaneous peak activation data, after correcting for sensor cross-talk,^{51,97} generally exhibit opposing response polarities in GCaMP (mostly above zero) and CBV (mostly below zero) signals, suggesting atypical neurovascular coupling. Nevertheless, there are also notable non-opposing features between foot shock GCaMP and CBV responses (e.g., $t \sim 5$ s and $\sim 15-30$ s). This observation underscores the intricate coupling relationship between neuronal and vascular activities in more naturalistic conditions.”

6. Do the authors have any data about the dependence of fMRI results on stimulus parameters such as intensity and frequency? I don't want them to do further experiments, but it would be very interesting to learn more about what might have been tried already. On a related point, can they comment on the behavioral consequences of stimuli such as those used for Figure 1.

[Response] We selected the 40 Hz stimulus frequency based on a classical study highlighting gamma-range LFP and spiking as primary contributors to the BOLD signal (Shmuel and Leopold, Human Brain Mapping, 2008; PMID: 18465799), and prior work in our lab consistently demonstrating the effectiveness of this frequency in driving strong striatal fMRI responses (Decot et al., 2017, Neuropsychopharmacology; PMID: 27515791; and Van den berg et al., 2017, Neuroimage; PMID: 27825979). We've clarified this choice in our revised manuscript's Results section and to further illustrate that negative striatal CBV changes can occur at lower stimulus frequencies and through various cortical inputs, we added two additional supplemental figures, one with optogenetic stimulation at 20 Hz (anterior insular cortex, AI) and the other at 10 and 20 Hz (orbitofrontal cortex, OFC). In addition, the AI experiments were executed by using a different duty cycle (20 sec on, 80 sec off), lower pulse width than some stimulations (5 ms), lower power (10 mW versus 20 mW), and a different virus (Chronos vs ChR2), whereas the OFC experiments were done in a different rat strain (Wistar). These experiments revealed robust unilateral positive fMRI responses at the stimulation site and stimulus-evoked negative CBV responses within the CPU.

These new supplementary Figures are referred to in the manuscript as follows:

“Optogenetic stimulation was applied to each of these target regions using a 40 Hz pulse train, chosen not only for its alignment with low gamma frequency and its established contribution to the BOLD signal⁶¹ but also based on our prior studies showing pronounced modulation of fMRI signals within the CPU when employing this particular frequency.^{62, 63} This stimulation paradigm yielded local positive CBV changes at the stimulation site while consistently eliciting negative CBV responses within the CPU across all cases (Fig. 1a-c, Fig. S2a-c, and Fig. S3a-c). Furthermore, we extended our findings by demonstrating that negative CBV responses within the CPU could also be evoked at lower stimulus frequencies in other cortical regions projecting to the CPU (Fig. S4, Fig. S5).”

These new supplementary figures are presented in the Supplementary Information accordingly:

Fig. S4: 20 Hz optogenetic stimulation of anterior insular cortex (AI) during CBV fMRI. a fMRI response maps acquired from eYFP control and Chronos subjects following optogenetic stimulation of the right AI. Response maps thresholded to $p < 0.001$, FWE corrected to $\alpha < 0.01$. **b** ROIs used for timeseries extraction were collected from the intersection of the stimulus evoked response map (red) and anatomical boundary of AI (yellow) or CPU (blue), corresponding to the orange and purple regions, respectively. **a-b** Left-to-right corresponds to 1 mm steps in the posterior-to-anterior direction, with the 5th slice from the right located at the anterior commissure (approximately -0.36 mm AP). **c** Stimulation schematics (top left), where AI viral expression and projections to CPU are indicated in green (not all projections shown) and optogenetically stimulated area of AI is indicated in blue; locations of optical fiber tips for AI optogenetic stimulation (top right); representative tissue cross-section indicating spread of eYFP from AI injection site (bottom), with 3-D models of the brain to the right indicating the location of the histological slice. **d** CBV time-courses from AI response maps (left) and CPU response maps (right) aligned to stimulation epochs (green bars indicate 20 Hz optogenetic stimulation blocks; data are presented as mean \pm SEM), with corresponding quantified peak amplitude changes (Chronos vs. eYFP Welch’s t-test, **** $p < 0.0001$).

a-d AI stimulation (20 s on, 80 s rest; 20 Hz, 10 mW power at fiber tip, 5 ms pulse-width) during CBV fMRI data: (Chronos n = 9 rats, 40 epochs, 80 peaks; eYFP n = 6 rats, 18 epochs, 36 peaks).

Fig. S5: 10 and 20 Hz optogenetic stimulation of orbitofrontal cortex (OFC) during CBV fMRI. fMRI response maps were acquired from unilateral optogenetic stimulation of the OFC (5 ms pulse width, 473 nm wavelength, 10 mW power), using a paradigm of 30 s stimulation off, followed by 15 s stimulation on then 60 s off repeated a total of four times, in either the ChR2-containing (left) hemisphere, **a**, at 10 Hz (top) and 20 Hz (bottom), or the eYFP-containing control (right) hemisphere, **b**, at 10 Hz (top) and 20 Hz (bottom), of four rats (3 male, 1 female). **a-b** Response maps were obtained from one-sample t-tests using the AFNI4 3dttest++ command and thresholded to $p < 0.05$ without multiple comparison correction due to the limited statistical power of this proof-of-concept experiment. Instead, a cluster voxel size > 40 was applied. Left-to-right images correspond to 1 mm steps in the posterior-to-anterior direction, with the 5th slice from the right located at the anterior commissure (approximately -0.36 mm AP). Before group-level analysis, data were preprocessed for slice timing and motion corrections with AFNI and spatially aligned to the Tohoku rat brain template² using ITK-SNAP.³ Functional images were smoothed with a 0.5 mm FWHM Gaussian kernel and first-level GLM was conducted via AFNI 3dDeconvolve against the MION hemodynamic response function convolved with the stimulation paradigm, which also included a second-order polynomial curve and motion parameters. **c** Stimulation schematics (left), where OFC viral expression and projections to CPu are indicated in green (not all projections shown) and optogenetically stimulated area of OFC is indicated in blue; locations of optical fiber tips for OFC optogenetic stimulation (right), implanted 0.5 mm above the viral infusion site. **d** CBV time-course data for 10 Hz stimulation (top) and 20 Hz stimulation (bottom) was extracted from experimenter defined ROIs corresponding to the generated evoked contrast maps, trimmed to the anatomical boundaries of the OFC (left) and CPu (right) regions (green bars indicate optogenetic stimulation blocks; data are presented as mean \pm SEM; $n = 4$ rats). All other surgical procedures, animal preparation for scanning and scanning protocol, and the percent CBV change for time-course data calculation was as described in the main Methods.

Despite the new data above, we cannot rule out the possibility that certain stimulus frequencies or pulse paradigms might elicit distinct fMRI responses. In the Introduction, we have incorporated additional references that discuss these nuances and emphasize variations in observations:

“Previous studies have revealed that heightened neuronal activity does not consistently lead to positive hemodynamic responses in the striatum. In rodent experiments, both positive⁴²⁻⁵⁰ and negative⁵¹⁻⁵⁷ hemodynamic responses in the CPU have been observed where increased neuronal activity was either directly measured or inferred by selective manipulations (i.e., optogenetics). Intriguingly, the occurrence of negative hemodynamic responses in the CPU cannot be easily attributed to the activation of inhibitory neurons, as several studies have reported positive local hemodynamic responses following the selective activation of cortical inhibitory neurons.^{12, 30, 38-40, 49, 58, 59} Consequently, it becomes apparent that the variable hemodynamic responses observed in the CPU may depend on several factors, including the choice of anesthetics, the physiological condition of the subject, the prevailing brain state, as well as the frequency and amplitude of the targeted modulations.”

In addition, we have raised cautions regarding the use of 40 Hz optogenetics in Results:

“While optogenetic stimulation enabled us to precisely control neural circuits capable of driving negative hemodynamic responses in CPU, the synchronized activity induced by such artificial stimulations is unlikely to occur naturally, and at certain stimulation targets may drive neuronal spiking rates outside of physiological ranges.⁸⁴ Further, although the light anesthesia protocol employed for the aforementioned recordings is well established in rodents to facilitate reproducible neuronal and vascular responses and stable physiology,⁸⁵⁻⁸⁹ it is widely known that anesthetics can alter these metrics.⁸⁹⁻⁹² To examine the relationship between CPU hemodynamics and neuronal firing under more naturalistic conditions, we employed spectral fiber-photometry to simultaneously measure neuronal and vascular activity^{51, 93-96} in the CPU of freely-moving awake rats (Fig. 3a).”

Furthermore, we have taken a conservative approach in the Discussion when addressing the use of 40 Hz and the interpretation of respective findings:

“Although these results were obtained in sedated rats, particularly at high stimulus frequencies, and optogenetic manipulations may drive non-physiological neural activity patterns,⁸⁴ fiber photometry recordings in awake and behaving rats suggested that such an inverse relationship of CPU neuronal activity and hemodynamic signals may also occur under more naturalistic conditions.”

“We observed negative hemodynamic responses in CPU as a result of optogenetic stimulation, noxious stimuli, or large peaks in spontaneous local neuronal activity. These conditions feature synchronous, high-frequency neuronal activation (e.g., 40 Hz) which is known to be favorable for peptide release in the peripheral and central nervous systems.^{154-158”}

We appreciate the reviewer's comments regarding the potential behavioral consequences of the manipulations presented in Fig. 1. Since we did not conduct specific behavioral studies in these settings, we exercise caution in making speculative claims. However, we do emphasize the conditions that resulted in negative CBV changes in the CPU and have included the following statement in the Discussion:

“Compared to naturalistic stimuli that drive transient sensory-motor activity, optogenetic manipulations elicit more synchronized activity in the striatum, whereas noxious stimuli recruit opioidergic signaling. These conditions may be necessary to induce decreases in striatal CBV.”

7. The electrode recordings of Figure 2 provide the most direct evidence for increased CPU neural activity under conditions that elicited the negative fMRI responses in Figure 1. Do these results really demonstrate that the net result of the stimuli was to induce neurophysiological activation, however? The experiment of Figure 2b in particular barely elicits a spiking response. Could it be that the stimulus here also induces inhibition in the form of inhibitory potentials and hyperpolarization in neurons that don't contribute to MUA? Even if such inhibitory activity is accompanied by increased spiking by some neurons, the net result could be consistent with the hemodynamic results. Perhaps some increased discussion of this kind of possibility would be merited.

[Response] We appreciate the insightful comment from the reviewer. While various scenarios are possible, we do believe that the net result of the stimuli was to induce neurophysiological activation. This is supported by the fact that local CPU injection of lidocaine effectively suppressed the negative fMRI response, providing evidence that such manipulation increased CPU neuronal activity, as illustrated in Fig. 4c and discussed in the main manuscript. However, as the reviewer rightly points out, it is also plausible that CPU terminal stimulation in SNr may induce local inhibitory potentials and hyperpolarization in CPU, potentially contributing to the fMRI signal. Furthermore, it's possible that the stimuli selectively activate specific subsets of neurons, such as direct pathway MSNs, which could result in weaker electrophysiological responses. In addition, optogenetic stimulation of striatonigral projection terminals may influence striatal neural activity and hemodynamic signals through various putative mechanisms. These mechanisms include antidromic spiking as well as several polysynaptic routes, such as thalamus->striatum and thalamus->cortex->striatum pathways. To address these complexities, as well as the related points brought up by the Reviewer in R2.3, we have included the following text in the Discussion:

“It is worth noting that optogenetic stimulation of CPU projection terminals in SNr resulted in comparatively weaker CPU neuronal responses, possibly due to antidromic stimulation activating only approximately 50% of MSNs (i.e., the direct pathway MSNs).³⁰ The involvement of local inhibitory potentials and hyperpolarization could further complicate the generation of the fMRI signal. Additionally, it should be noted that optogenetic stimulation of CPU terminals in SNr can influence CPU activity through various circuit mechanisms, including polysynaptic routes involving thalamus and cortex. Indeed, our M1 and PfT optogenetic stimulation data support the idea that corticostriatal and thalamostriatal inputs can induce CPU activity changes. We acknowledge that the slow hemodynamic signal time course and variations in neurovascular coupling among different brain regions may hinder the definitive identification of specific circuit-level mechanisms using the data collected in this study. Future investigations employing optogenetic inhibition¹¹⁹ at cortical and thalamic sites, in addition to CPU terminal stimulation, may be necessary to further dissect these specific signaling pathways.”

We further acknowledge the potential limitation of the stimulation protocol used in Fig. 2b. In a complementary experimental approach, we have included a more naturalistic stimulation condition using photometry (Fig. 3). In this approach, the overall outcome demonstrates neuronal activation and vasoconstriction within the CPU. It's important to note that photometry records neuronal activity over a relatively large area of the striatum, offering a broader perspective compared to the limited number of cells recorded in electrophysiology.

8. As a minor corollary to the previous point, I don't think that the CBV decreases can be necessarily be “attributed to large increases in CPU activity” (lines 343-44). Ambiguity about net activity changes in CPU and the possibility of circuit mechanisms that complicate relationships between the driving

stimuli and the responses (point 3) both complicate the picture. I think it would be more accurate to say that the striatal vasoconstriction is “accompanied by increases in CPU activity.” This would also be more in keeping with the relatively restrained language of the abstract.

[Response] We appreciate the Reviewer's comment. We agree with the Reviewer and have revised our language accordingly. The sentence now states:

“Our results provide a causal demonstration that CPU vasoconstriction can be accompanied by increases in CPU neuronal activity, which may include pre- and/or post-synaptic activity changes.”

The relevant sentence in the abstract now reads:

“In human studies, manipulations aimed at increasing striatal neuronal activity, such as noxious transcutaneous electrical nerve stimulation or transcranial magnetic stimulation, have likewise elicited striatal negative fMRI responses, supporting the potential translatability of our findings.”

9. The human experiments of Figure 6 are a nice touch to this paper. In describing the TMS procedures on p. 17, I think the authors should say a bit more in the main text about the experimental design, rather than forcing readers to skip down to the methods section. It would also be good to use more cautious language on lines 328-9 regarding what the TMS is accomplishing at a neural level. This should be consistent with the ambiguities highlighted correctly on lines 412-3. Finally, in Figure 6, were the deactivations observed in CPU really the only consequences of the various stimuli? The TMS targets are all regions that connect to many other regions throughout the brain, so it would be surprising if the striatum was the only area where responses were observed, even to the exclusion of local responses. Some more clarification and/or discussion of this issue would be useful.

[Response] We have incorporated a concise summary description of the TMS procedures within the Results section:

“Eighty-two healthy individuals without a history of psychopathology underwent TMS-fMRI scanning using a gradient echo spiral in/out pulse sequence on a 3T scanner. Prior to the scanning session, TMS stimulation sites in right anterior middle frontal gyrus (aMFG), posterior middle frontal gyrus (pMFG), and M1 were defined using high-resolution T1-weighted anatomical images, and coordinates were transformed into subject-native space. Motor threshold was determined for each participant by identifying the lowest stimulation intensity that induced a visible muscle twitch response in the contralateral abductor pollicis brevis muscle. Concurrent TMS-fMRI sessions were conducted with a custom-built TMS-compatible head-coil, covering the whole brain with 31 slices and acquiring data in a fast event-related design with heart rate and respiration monitoring.”

We also agree with the Reviewer's feedback regarding the language used in the original lines 328-329. The revised sentence now reads:

“These findings suggest that negative hemodynamic responses in the rat CPU and at several locations within the human striatum can be observed when manipulating afferent regions, thereby indicating the possibility of shared characteristics in the vascular responses of the striatum/CPU across species.”

Finally, we appreciate the reviewer's question regarding the apparent lack of consequences outside of the CPU. While our main focus in Fig. 6 was on the striatal responses, it's important to clarify that the negative BOLD observed in the CPU were not the only consequences of the various stimuli. We

employed a masking approach to avoid susceptibility artifacts from the TMS stimulation sites, following an established protocol from our collaborators at Stanford University. In our original submission, we had clarified the use of ROI masking in the figure legend and displayed those masks in Fig 6a. In the revised manuscript, we have provided all unmasked data in new supplemental figures, shown below. Additionally, both the UNC (bilateral noxious electrical forearm stimulation) and Stanford (TMS) datasets are publicly accessible for further analysis and transparency in our research.

These new supplementary figures, which also include unmasked response maps, are now referenced in the main manuscript Results, and referred to in greater detail in the Fig. 6 legend:

“Unmasked versions of the human fMRI response maps displayed here as well as corresponding time courses are shown for noxious forearm stimulation, M1 TMS, aMFG TMS, and pMFG TMS in Fig. S12, Fig. S13, Fig. S14, and Fig. S15, respectively.”

These new figures and their legends are included in the following format in Supplementary Information:

Fig. S12: Unmasked human response maps and striatum timeseries from noxious peripheral stimulation in Fig. 6b. **a** Unmasked human BOLD fMRI response maps corresponding to the masked response maps presented in Fig. 6b. **b** Human striatum anatomical ROI mask used for time-course extraction, color-coded by striatal compartment. **c** Stimulation aligned (green bars indicate stimulation ON), BOLD fMRI time-courses from each bilateral striatal compartment ($n = 7$ subjects, 11 scans; data are shown as subject mean \pm SEM).

Fig. S13: Unmasked human response maps and striatum timeseries from right M1 TMS in Fig. 6c. **a** Unmasked human BOLD fMRI response maps corresponding to the masked response maps presented in Fig. 6c. **b** Human striatum anatomical ROI mask used for time-course extraction, color-coded by hemisphere and striatal compartment. **c** BOLD fMRI time-courses from each unilateral striatal compartment during fast event-related delivery of TMS pulses (pulse delivery was consistent between subjects and pulses are indicated by green bars; $n = 79$ subjects; data are shown as subject mean \pm SEM).

Fig. S14: Unmasked human response maps and striatum timeseries from right aMFG TMS in Fig. 6d. **a** Unmasked human BOLD fMRI response maps corresponding to the masked response maps presented in Fig. 6d. **b** Human striatum anatomical ROI mask used for time-course extraction, color-coded by hemisphere and striatal compartment. **c** BOLD fMRI time-courses from each unilateral striatal

compartment during fast event-related delivery of TMS pulses (pulse delivery was consistent between subjects and pulses are indicated by green bars; n = 80 subjects; data are shown as subject mean \pm SEM).

Fig. S15: Unmasked human response maps and striatum timeseries from right pMFG TMS in Fig. 6e. a Unmasked human BOLD fMRI response maps corresponding to the masked response maps presented in Fig. 6e. **b** Human striatum anatomical ROI mask used for time-course extraction, color-coded by hemisphere and striatal compartment. **c** BOLD fMRI time-courses from each unilateral striatal compartment during fast event-related delivery of TMS pulses (pulse delivery was consistent between subjects and pulses are indicated by green bars; n = 79 subjects; data are shown as subject mean \pm SEM).

10. A minor point: on line 56, the text should be edited to denote that the CPu is only the dorsal portion of the striatum, as conventionally defined.

[Response] We have modified the text accordingly. This line now reads:

“The dorsal striatum (or caudate-putamen, CPu, in rodents) is the major input nucleus of the basal ganglia”

11. I believe that the rightmost graph in Fig. 1d is mislabeled, since it shows negative Chr2 responses in GPe but not CPu.

[Response] We appreciate the Reviewer for the attention to detail and have fixed the label in the figure accordingly.

12. I have the impression that most of the fMRI statistics in the paper count each response in each animal as an independent measurement. This doesn't seem entirely fair to me, since variability due to injections, fiber placement, physiological peculiarities, etc. vary by animal and not by stimulus presentation. Perhaps the authors could find some way to represent inter-animal variability more clearly than I think they have done here.

We thank the Reviewer for their suggestion. We chose to analyze responses individually rather than by subject because it is the goal of this study to show not just that responses exist in the CPU and other ROI (which would be an apt use of subject-level stats) but to characterize the shape and sign of the responses with the highest detail and demonstrate the reliability of these responses. Though we believe our approach to be a straightforward and sufficient methodology for conveying the message that stimulation responses are similar as opposed to showing that all subjects had similar responses, for full

transparency of inter-animal variability, we have included an additional Supplementary Figure that shows the data grouped by subject as opposed to by trials, where applicable.

We now include the following rationale for our approach in the main manuscript Methods:

“Where applicable, our CBV-fMRI, electrophysiology, fiber photometry, and FSCV analyses focused on individual response trials from each subject in order to characterize, with the most detail possible, the shape, sign, and reliability of responses from several stimulation paradigms over each of these modalities. Nonetheless we also present subject-level, rather than trial-level, analyses in order highlight the inter-animal variability within each experiment (Fig. S17).”

The new figure displayed below, with subject-level statistics, can now be found in Supplementary Information.

Fig. S17: Analyses reproductions with subject-level mean data. **a** (Fig. 1) CBV fMRI response peaks to optogenetic stimulation; from left to right: (Fig. 1a) M1 stimulation (Chr2 vs. eYFP Welch's t-test, *p = 0.0371, **p = 0.0081); (Fig. 1b) PfT stimulation (Chr2 vs. eYFP Welch's t-test, ^{ns}p = 0.1185, **p = 0.0082); (Fig. 1c) GPe stimulation (Chr2 vs. eYFP Welch's t-test, ^{ns}p = 0.0775, **p = 0.0047); (Fig. 1d) CPu stimulation (Chr2 vs. eYFP Welch's t-test, **p = 0.0062, ^{ns}p = 0.1322); (Fig. 1e) SNr stimulation (Chr2 vs. eYFP Welch's t-test, ****p < 0.0001); (Fig. 1f) SNc stimulation (Chr2 vs. eYFP Welch's t-test, *p = 0.0496). **b** (Fig. 2a-c) Electrophysiology 40Hz LFP power change and mean MUA spike frequency during optogenetic stimulation; from left to right: (Fig. 2a) PfT stimulation LFP (Chr2 vs. eYFP Welch's t-test, *p = 0.0388) and MUA (Chr2 vs. eYFP Welch's t-test, ^{ns}p = 0.0760); (Fig. 2b) SNr stimulation LFP (Chr2 vs. eYFP Welch's t-test, ^{ns}p = 0.0819) and MUA (Chr2 vs. eYFP Welch's t-test, ^{ns}p = 0.2572); (Fig. 2c) SNc stimulation LFP (Chr2 vs. eYFP Welch's t-test, **p = 0.0067) and MUA (Chr2 vs. eYFP Welch's t-test, *p = 0.0292). **c** (Fig. 2e-g) Voltammetry DA and O₂ response peaks to optogenetic stimulation; from left to right: (Fig. 2e) SNc stimulation DA (Chr2 vs. eYFP Welch's t-test, ^{ns}p = 0.2327) and O₂ (Chr2 vs. eYFP Welch's t-test, **p = 0.0023); (Fig. 2f) PfT stimulation DA (Chr2 vs. eYFP Welch's t-test, ^{ns}p = 0.6772) and O₂ (Chr2 vs. eYFP Welch's t-test, *p = 0.0148); (Fig. 2g) SNr stimulation DA (Chr2 vs. eYFP Welch's t-test, ^{ns}p = 0.6472) and O₂ (Chr2 vs. eYFP Welch's t-test, *p = 0.0373). **d** (Fig. 3) GCaMP and CBV photometry peaks from footshocks and relative to spontaneous GCaMP activity; from left to right: (fig. 3b) footshock response GCaMP (1st peak one-sample t-test, *p = 0.0155; 2nd peak one-sample t-test, ^{ns}p = 0.4018) and CBV (1st peak one-sample t-test, *p = 0.0217; 2nd peak one-sample t-test, ***p = 0.0006 ; (fig. 3c) spontaneous activity GCaMP (one-sample t-test, ****p < 0.0001) and CBV (one-sample t-test, ^{ns}p = 0.0911). **e** (Fig. S11) CBV fMRI response peaks to noxious forepaw stimulation (one-sample t-test, ****ps < 0.0001). **f** (Fig. S16) BOLD fMRI response peaks to optogenetic PfT stimulation (Chr2 vs. eYFP, Welch's t-test, ^{ns}p = 0.1225, **p = 0.0062). **g** (Fig. S4) CBV fMRI response peaks to optogenetic AI stimulation (Chronos vs. eYFP Welch's t-test, ***ps = 0.0003).

Reviewer #3 (Remarks to the Author):

Overall Review:

The paper presents a comprehensive assessment of CPU fMRI responses to optogenetic stimulation of CPU neurons or afferents in rats, investigating the relationship between neuronal and hemodynamic activity in the striatum. The authors demonstrate that striatal hemodynamic responses are not solely dependent on neuronal activity and identify opioids as playing a critical role in generating negative fMRI signals. The paper also provides evidence that negative fMRI responses may occur in the human striatum, prompting consideration of local cellular and neurochemical environments in fMRI signal interpretation. The research questions and the results are interesting. The text is generally clearly presented. However, in my detailed description below, I raise some concerns that need to be addressed.

[Response] We are glad that the Reviewer found our work to be comprehensive and interesting and appreciate the kind words. We value the reviewer's concerns and addressing them in detail has enabled us to further improve our manuscript.

Major:

1. The authors keep referring to decreased CBV as "negative fMRI responses" which is kind of confusing. Is it the decreased T2*-weighted signal or decreased blood volume? It would be recommended to clarify this in the manuscript.

[Response] We thank the reviewer for this comment. Our original wording suggested decreased blood volume. Except for the human BOLD fMRI studies, the "negative fMRI responses" language has been replaced with "negative CBV responses" or "vasoconstriction" throughout the manuscript.

2. It is novel to see decreased CBV, i.e., increased T2* weighted signal in the targeted region. The authors claim it is due to vessel constriction (reduced iron concentration per voxel/ improved homogeneity), but there is another well-known effect of the improved homogeneity by increasing the deoxyhemoglobin/oxyhemoglobin, i.e., positive BOLD signal. Should the authors clarify whether there are still BOLD signals which will contribute to the increased T2* weighted signal. This is a crucial control experiment for the authors to make the statement that vessel constriction is the pure contribution to the increased T2* weighted signals. Iron particle dose-dependent experiment is a potential way to exclude BOLD contribution.

[Response] We appreciate the Reviewer for this note. In our opinion, positive BOLD signals are not likely to contribute to the increased T2* weighted signal observed in our fMRI experiments, as we used high contrast agent dosing (30 mg/kg) and cross-validated the polarity of the hemodynamic responses with complementary techniques.

Existing studies that have acquired evoked BOLD and CBV changes in the same subjects using the same stimulation have observed a low impact from the BOLD effect. For example, Lu et al. (2007, Magnetic Resonance in Medicine; PMID: 17763339) performed paw stimulation in rat at 9.4 T with 5-30 mg/kg iron contrast agent dosing and observed that evoked CBV changes strongly reflected BOLD signal at low iron doses (e.g., 5 mg/kg) but not at high doses (23-30 mg/kg). The impact of fully oxygenated arterial blood on voxel homogeneity decreases as voxels are increasingly dephased through increasing contrast agent concentrations. At the chosen contrast agent dose of 30 mg/kg, CBV imaging also has superior signal to noise ratio to BOLD, which contributed to our choice of modality.

Additionally, previous work from our lab (Shih et al., 2011, Journal of Cerebral Blood Flow and Metabolism; PMID: 20940730) and others (Zhao et al., 2012, Neuroimage; PMID: 21856430) has shown

that noxious paw stimulations that decreased CBV also evoked negative BOLD in caudate putamen (using 30 mg/kg and 15 mg/kg contrast agent dosing, respectively). In the present study, we observed signal changes consistent with CBV fMRI in the caudate putamen with photometry and voltammetry (decreased CBV and decreased concentration of extracellular oxygen, respectively). Collectively, increased homogeneity from increased BOLD is unlikely to contribute to our observed T2* increases. To clarify this concern, we have added the following text to Discussion:

“The negative CBV changes (elevated T2* weighted signal in raw data) observed in our fMRI experiments are unlikely to be driven by positive BOLD signals, given our high contrast agent dosing (30 mg/kg) and validation of hemodynamic response polarity through complementary techniques. Prior studies employing sequential BOLD and CBV measurements during identical stimulation paradigms have demonstrated minimal influence from the BOLD effect. For instance, Lu et al. (2007)¹¹³ conducted rat paw stimulation experiments at 9.4 T, employing iron contrast agent doses ranging from 5-30 mg/kg. They observed a notable BOLD influence to CBV measurement at lower iron doses (e.g., 5 mg/kg) but not at higher doses (15-30 mg/kg), indicating that as contrast agent concentration increases, the impact of fully oxygenated arterial blood on voxel homogeneity diminishes.”

3. It is great to detect potential similarities in striatum/CPu vascular response mechanisms between species. Since CBV was used in rodents and BOLD was used in humans, it would be nice to discuss why the authors didn't perform the BOLD in the same setup in the animal models to further strengthen the similarity statement.

[Response] We value the Reviewer's consideration regarding the comparison of species using distinct fMRI modalities. While we did not collect BOLD data for all circuit stimulations presented in Fig. 1, we have introduced a new Supplementary Figure that replicates the PfT optogenetic stimulation using BOLD fMRI. As anticipated, positive BOLD responses were detected at the stimulation site, with an overall negative BOLD pattern in CPu. The following text has been added to the Discussion:

“In order to illustrate the BOLD response pattern in an identical experimental context, we replicated the PfT optogenetic stimulation experiment without administration of CBV contrast agent to obtain BOLD fMRI as the readout. The corresponding BOLD fMRI maps, time courses, and relevant discussion can be found in Fig. S16.”

Aside from the negative BOLD response in the CPu ROI, it is worth mentioning that we also identified a relatively small cluster exhibiting positive BOLD within the CPu, and this observation is discussed in the figure legend below.

Fig. S16: Optogenetic stimulation of PFT during BOLD fMRI. **a** BOLD fMRI response maps acquired from eYFP control and ChR2 subjects following optogenetic stimulation of the right PFT. Response maps thresholded to $p < 0.001$, FWE corrected to $\alpha < 0.01$. **b** ROIs used for timeseries extraction were collected from the intersection of the stimulus evoked response map (red) and anatomical boundary of PFT (yellow) or CPu (blue), corresponding to the orange and purple regions, respectively. **a-b** Left-to-right corresponds to 1 mm steps in the posterior-to-anterior direction, with the 5th slice from the right located at the anterior commissure (approximately -0.36 mm AP). **c** Stimulation schematics (left), where PFT viral expression and projections to CPu are indicated in green (not all projections shown) and optogenetically stimulated area of PFT is indicated in blue, and locations of optical fiber tips for PFT optogenetic stimulation (right). **d** BOLD time-courses from PFT response maps (left) and CPu response maps (right) aligned to stimulation epochs (green bars indicate 40 Hz optogenetic stimulation blocks; data are presented as mean \pm SEM), with corresponding quantified peak amplitude changes (ChR2 vs. eYFP Welch's t-test, **** $p < 0.0001$). **a-d** PFT stimulation during BOLD fMRI data: (ChR2 $n = 5$ rats, 75 epochs, 150 peaks; eYFP $n = 2$ rats, 30 epochs, 60 peaks). While the time course extracted from the CPu ROI displayed negative BOLD changes during the stimulus period, we also observed a minor cluster exhibiting positive BOLD activation within the dorsolateral CPu. Notably, in a separate experiment employing identical manipulation, only negative CBV responses were elicited in the CPu (Fig. 1b, Fig. S2b). The increases in BOLD in dorsolateral CPu, potentially accompanied by decreases in CBV, could occur when CBV decreases substantially without prominent alterations in blood oxygenation, resulting in reduced deoxyhemoglobin occupancy and susceptibility effects. Alternatively, this observation may indeed be a result of the interplay between two competing forces within the CPu, as discussed in the main text of this manuscript: feed-forward vasoconstrictive neurotransmission versus activity-driven metabolic processes. It is plausible that BOLD, being a contrast sensitive to multiple physiological factors, is capable of capturing these nuanced differences within distinct territories of the CPu that may not be discernible through CBV measurement alone.⁴ Given the multifaceted nature of the BOLD signal, which can be influenced by varying alterations in blood oxygenation, blood flow, CBV, and local oxygen consumption, it is imperative that future multimodal neuroimaging studies are conducted to systematically dissect the underlying biophysical mechanisms driving this intriguing observation.

4. The authors claim that decreased BOLD in humans and decreased CBV in rodents share potential similarities. As these two phenomena do not necessarily concurrent, it is better to discuss more detail.

[Response] We appreciate the reviewer's comment. Indeed, as described in Drew, PJ (2019, Current Opinion in Neurobiology; PMID: 31336326), BOLD and CBV responses are not necessarily concurrent and represent different processes. Our responses to R3.2 and R3.3 should have illustrated that while there is a basis to expect comparability between CBV and BOLD responses in CPu for a given stimulation paradigm, differences between these two measures may also exist. We refer the readers to this discussion from the main manuscript text, as indicated below:

“The corresponding BOLD fMRI maps, time courses, and relevant discussion can be found in Fig. S16.”

5. Could the authors clarify the rationale for using different species (Rats and mice) in the experiment?

[Response] We now provide an additional explanation for our use of rats as our primary subjects in the Methods.

“Rats were chosen as the primary subjects for the majority of our experiments due to their larger brain size, which allows for higher respective imaging resolution compared to mice.^{87, 88, 182-184} Additionally, rats are frequently employed in pharmacological studies because many drug metabolic pathways have been well-characterized and/or are similar to humans,^{185, 186} and they are well-suited for behavioral neuroscience research thanks to their superior ability to learn complex tasks compared to mice.¹⁸⁷”

Mice were only used in the bright field microscopy experiments out of necessity. The highly-specialized technique is performed by only a few laboratories in the world, who happen to work with murine subjects. We specify this specific use of mice in the Methods of the main manuscript.

“In addition, brain tissue from young (postnatal day 21–28) C57BL/6J mice of both sexes (n = 3, 1 male), bred in house, was acquired for use on site for bright-field vascular imaging experiments at the University of Sussex”

6. In Figure 1, the authors find negative CBV response amplitude in CPu was anti-correlated with the positive responses, what does this finding indicate?

[Response] We thank the reviewer for bringing this up and agree that this should be addressed more clearly. Our finding that negative CBV response amplitude was negatively correlated with positive responses is consistent with a direct, activity-dependent relationship of hemodynamic signals in the CPu and upstream regions that were targeted for stimulation. Such correlations may not be anticipated if the regions were more indirectly coupled. We have added an additional statement in the Results to reflect this interpretation:

“The observed negative correlations between CBV responses at various optogenetic stimulation targets and the CPu suggest a “negative coupling” relationship, indicating that the intensity of CPu vasoconstriction scales with the activation levels in its connected brain regions. This suggests that neurovascular coupling, manifested as a decrease in CBV when synaptic inputs to these regions or CPu cell bodies are activated, still maintains an activity-dependent relationship.”

7. Subjects were all male rodents. The authors should comment briefly on this as a limitation.

[Response] We have added the following text to the Discussion to highlight this limitation:

“Further, the strengths of our results lay in validating the observed negative hemodynamic phenomenon using multiple complementary techniques in awake and anesthetized conditions and in multiple species.

Nonetheless, it's important to acknowledge the presence of certain limitations in our study. Although we did include female subjects in our experiments, it's essential to clarify that our study was not specifically tailored or adequately powered to delve into potential sex-related differences. It is worth noting that variations in pain perception and the efficacy of opioidergic treatment could indeed exist among different sexes, as indicated by previous research.^{171, 172}

8. The authors stimulate each of the brain target regions with a 40 Hz pulse, it would be good to explain why 40 Hz was chosen.

We selected the 40 Hz stimulus frequency based on a classical study highlighting gamma-range LFP and spiking a primary contributors to the BOLD signal (Shmuel and Leopold, Human Brain Mapping, 2008; PMID: 18465799), and prior work in our lab consistently demonstrating the effectiveness of this frequency in driving strong striatal fMRI responses (Decot et al., 2017, Neuropsychopharmacology; PMID: 27515791; and Van den berg et al., 2017, Neuroimage; PMID: 27825979). We've clarified this choice in our revised manuscript's Results section and to further illustrate that negative striatal CBV changes can occur at lower stimulus frequencies and through various cortical inputs, we added two additional supplemental figures, one with optogenetic stimulation at 20 Hz (anterior insular cortex, AI) and the other at 10 and 20 Hz (orbitofrontal cortex, OFC). In addition, the AI experiments were executed by using a different duty cycle (20 sec on, 80 sec off), lower pulse width than some stimulations (5 ms), lower power (10 mW versus 20 mW), and a different virus (Chronos vs Chr2), whereas the OFC experiments were done in a different rat strain (Wistar). These experiments revealed robust unilateral positive fMRI responses at the stimulation site and stimulus-evoked negative CBV responses within the CPu.

These new supplementary Figures are referred to in the manuscript as follows:

“Optogenetic stimulation was applied to each of these target regions using a 40 Hz pulse train, chosen not only for its alignment with low gamma frequency and its established contribution to the BOLD signal⁶¹ but also based on our prior studies showing pronounced modulation of fMRI signals within the CPu when employing this particular frequency.^{62, 63} This stimulation paradigm yielded local positive CBV changes at the stimulation site while consistently eliciting negative CBV responses within the CPu across all cases (Fig. 1a-c, Fig. S2a-c, and Fig. S3a-c). Furthermore, we extended our findings by demonstrating that negative CBV responses within the CPu could also be evoked at lower stimulus frequencies in other cortical regions projecting to the CPu (Fig. S4, Fig. S5).”

Further, these new supplementary figures are presented in the Supplementary Information accordingly:

Fig. S4: 20 Hz optogenetic stimulation of anterior insular cortex (AI) during CBV fMRI. **a** fMRI response maps acquired from eYFP control and Chronos subjects following optogenetic stimulation of the right AI. Response maps thresholded to $p < 0.001$, FWE corrected to $\alpha < 0.01$. **b** ROIs used for timeseries extraction were collected from the intersection of the stimulus evoked response map (red) and anatomical boundary of AI (yellow) or CPu (blue), corresponding to the orange and purple regions, respectively. **a-b** Left-to-right corresponds to 1 mm steps in the posterior-to-anterior direction, with the 5th slice from the right located at the anterior commissure (approximately -0.36 mm AP). **c** Stimulation schematics (top left), where AI viral expression and projections to CPu are indicated in green (not all projections shown) and optogenetically stimulated area of AI is indicated in blue; locations of optical fiber tips for AI optogenetic stimulation (top right); representative tissue cross-section indicating spread of eYFP for from AI injection site (bottom), with 3-D models of the brain to the right indicating the location of the histological slice. **d** CBV time-courses from AI response maps (left) and CPu response maps (right) aligned to stimulation epochs (green bars indicate 20 Hz optogenetic stimulation blocks; data are presented as mean \pm SEM), with corresponding quantified peak amplitude changes (Chronos vs. eYFP Welch's t-test, **** $p < 0.0001$). **a-d** AI stimulation (20 s on, 80 s rest; 20 Hz, 10 mW power at fiber tip, 5 ms pulse-width) during CBV fMRI data: (Chronos $n = 9$ rats, 40 epochs, 80 peaks; eYFP $n = 6$ rats, 18 epochs, 36 peaks).

Fig. S5: 10 and 20 Hz optogenetic stimulation of orbitofrontal cortex (OFC) during CBV fMRI. fMRI response maps were acquired from unilateral optogenetic stimulation of the OFC (5 ms pulse width, 473 nm wavelength, 10 mW power), using a paradigm of 30 s stimulation off, followed by 15 s stimulation on then 60 s off repeated a total of four times, in either the ChR2-containing (left) hemisphere, **a**, at 10 Hz (top) and 20 Hz (bottom), or the eYFP-containing control (right) hemisphere, **b**, at 10 Hz (top) and 20 Hz (bottom), of four rats (3 male, 1 female). **a-b** Response maps were obtained from one-sample t-tests using the AFNI4 3dttest++ command and thresholded to $p < 0.05$ without multiple comparison correction due to the limited statistical power of this proof-of-concept experiment. Instead, a cluster voxel size > 40 was applied. Left-to-right images correspond to 1 mm steps in the posterior-to-anterior direction, with the 5th slice from the right located at the anterior commissure (approximately -0.36 mm AP). Before group-level analysis, data were preprocessed for slice timing and motion corrections with AFNI and spatially aligned to the Tohoku rat brain template² using ITK-SNAP.³ Functional images were smoothed with a 0.5 mm FWHM Gaussian kernel and first-level GLM was conducted via AFNI 3dDeconvolve against the MION hemodynamic response function convolved with the stimulation paradigm, which also included a second-order polynomial curve and motion parameters. **c** Stimulation schematics (left), where OFC viral expression and projections to CPu are indicated in green (not all projections shown) and optogenetically stimulated area of OFC is indicated in blue; locations of optical fiber tips for OFC optogenetic stimulation (right), implanted 0.5 mm above the viral infusion site. **d** CBV time-course data for 10 Hz stimulation (top) and 20 Hz stimulation (bottom) was extracted from experimenter defined ROIs corresponding to the generated evoked contrast maps, trimmed to the anatomical boundaries of the OFC (left) and CPu (right) regions (green bars indicate optogenetic stimulation blocks; data are presented as mean \pm SEM; $n = 4$ rats). All other surgical procedures, animal preparation for scanning and scanning protocol, and the percent CBV change for time-course data calculation was as described in the main Methods.

Despite the new data above, we cannot rule out the possibility that certain stimulus frequencies or pulse paradigms might elicit distinct fMRI responses. In the Introduction, we have incorporated additional references that discuss these nuances and emphasize variations in observations:

“Previous studies have revealed that heightened neuronal activity does not consistently lead to positive hemodynamic responses in the striatum. In rodent experiments, both positive⁴²⁻⁵⁰ and negative⁵¹⁻⁵⁷ hemodynamic responses in the CPU have been observed where increased neuronal activity was either directly measured or inferred by selective manipulations (i.e., optogenetics). Intriguingly, the occurrence of negative hemodynamic responses in the CPU cannot be easily attributed to the activation of inhibitory neurons, as several studies have reported positive local hemodynamic responses following the selective activation of cortical inhibitory neurons.^{12, 30, 38-40, 49, 58, 59} Consequently, it becomes apparent that the variable hemodynamic responses observed in the CPU may depend on several factors, including the choice of anesthetics, the physiological condition of the subject, the prevailing brain state, as well as the frequency and amplitude of the targeted modulations.”

In addition, we have raised cautions regarding the use of 40 Hz optogenetics in Results:

“While optogenetic stimulation enabled us to precisely control neural circuits capable of driving negative hemodynamic responses in CPU, the synchronized activity induced by such artificial stimulations is unlikely to occur naturally, and at certain stimulation targets may drive neuronal spiking rates outside of physiological ranges.⁸⁴ Further, although the light anesthesia protocol employed for the aforementioned recordings is well established in rodents to facilitate reproducible neuronal and vascular responses and stable physiology,⁸⁵⁻⁸⁹ it is widely known that anesthetics can alter these metrics.⁸⁹⁻⁹² To examine the relationship between CPU hemodynamics and neuronal firing under more naturalistic conditions, we employed spectral fiber-photometry to simultaneously measure neuronal and vascular activity^{51, 93-96} in the CPU of freely-moving awake rats (Fig. 3a).”

Furthermore, we have taken a conservative approach in the Discussion when addressing the use of 40 Hz and the interpretation of respective findings:

“Although these results were obtained in sedated rats, particularly at high stimulus frequencies, and optogenetic manipulations may drive non-physiological neural activity patterns,⁸⁴ fiber photometry recordings in awake and behaving rats suggested that such an inverse relationship of CPU neuronal activity and hemodynamic signals may also occur under more naturalistic conditions.”

“We observed negative hemodynamic responses in CPU as a result of optogenetic stimulation, noxious stimuli, or large peaks in spontaneous local neuronal activity. These conditions feature synchronous, high-frequency neuronal activation (e.g., 40 Hz) which is known to be favorable for peptide release in the peripheral and central nervous systems.^{154-158”}

REVIEWERS' COMMENTS

Reviewer #1 (Remarks to the Author):

This revised manuscript has addressed all comments satisfactory. This exciting paper will make a major impact on neurovascular coupling and fMRI fields.

Reviewer #2 (Remarks to the Author):

The authors have done a very good job responding to my comments. I congratulate them on their work.

Reviewer #3 (Remarks to the Author):

All comments have been addressed.

REVIEWERS' COMMENTS

We express our gratitude to the reviewers for their encouraging support, and we are delighted that our revised manuscript effectively addressed all of their comments. Once more, we sincerely appreciate the reviewers' time, invaluable expertise, and guidance, which have notably shaped and enhanced this work.

Reviewer #1 (Remarks to the Author):

This revised manuscript has addressed all comments satisfactory. This exciting paper will make a major impact on neurovascular coupling and fMRI fields.

Reviewer #2 (Remarks to the Author):

The authors have done a very good job responding to my comments. I congratulate them on their work.

Reviewer #3 (Remarks to the Author):

All comments have been addressed.